# Ice-nucleating particles from open-lot livestock facilities in Texas

Naruki Hiranuma[1], Brent W. Auvermann[2], Franco Belosi[3], Jack Bush[2], Kimberly M. Cory[1,4], Romy Ullrich-Fösig[5], Dimitrios G. Georgakopoulos[6], Kristina Höhler[5], Yidi Hou[1], Larissa Lacher[5], Harald Saathoff[5], Gianni Santachiara[3], Xiaoli Shen[5,7], Isabelle Steinke[5,8], Nsikanabasi S. Umo[5], Hemanth S. K. Vepuri[1], Franziska Vogel[5], Ottmar Möhler[5]

[1]Department of Life, Earth, and Environmental Sciences, West Texas A&M University, Canyon, TX 79016, USA
[2]Texas A&M AgriLife Research, Amarillo, TX 79106, USA
[3]Institute of Atmospheric Sciences and Climate, National Research Council, Bologna, 40129, Italy
[4]Department of Environmental Toxicology, Texas Tech University, Lubbock, TX 79409, USA
[5]Institute of Meteorology and Climate Research, Karlsruhe Institute of Technology, Karlsruhe, 76021, Germany
[6]Department of Crop Science, Agricultural University of Athens, Athens, 118 55, Greece
[7]Department of Earth Atmospheric and Planetary Sciences, Purdue University, West Lafayette, IN 47907, USA
[8]Atmospheric Sciences & Global Change, Pacific Northwest National Laboratory, Richland, WA 99354, USA

Correspondence: Naruki Hiranuma (nhiranuma@wtamu.edu)

Keywords. Ice, Freezing, Cloud, Cattle, Open-Lot Livestock Facility, Soil Dust

**Abstract.**
This study presents a comprehensive investigation of ice-nucleating particles (INPs) from open-lot livestock facilities (OLLFs) in Texas, USA. A three-year field survey (2017 – 2019) was conducted to understand immersion-mode INP abundance from four commercial OLLFs in the Texas Panhandle in different seasons, including summer, spring, and winter. A high concentration of INPs in air, $n_{INP}$, of $1,171.6 \pm 691.6$ L$^{-1}$ (average ± standard error) was measured at -25 °C for aerosol particles collected at the downwind edges of these OLLFs. An obvious seasonal variation in $n_{INP}$, peaking in summer, was observed at OLLFs with the maximum $n_{INP}$ at the same temperature exceeding 10,000 L$^{-1}$ on July 23, 2018. The observed high $n_{INP}$ is an order of magnitude higher than what has been found in previous studies on fertile and agricultural soil dust INPs, and we were able to detect INPs at temperatures as high as -5 °C. Interestingly, the $n_{INP}$ values from our field survey exhibited a strong correlation with measured particulate matter mass concentration (r = 0.94; $> 3 \times 10^{-7}$ g L$^{-1}$ in summer), suggesting the importance of large particles in immersion freezing for INPs from OLLF. Motivated by these extremely high INP concentrations, we have conducted a systematic laboratory study at the Aerosol Interaction and Dynamics in the Atmosphere (AIDA) cloud chamber facility to gain further insights into INP propensity and properties of surface materials from two OLLF facilities, one in the Texas Panhandle and another from McGregor, Texas, as OLLF dust proxies. Based on a modern suite of online and offline aerosol particle characterization instruments, we examined the ice nucleation (IN) efficiency of these materials in the temperature range between -7.5 °C and -29°C. Our laboratory results showed on average ≈ 50% supermicron size dominance in the INPs of both OLLF dust proxies with a high efficiency of immersion/condensation freezing, as represented by an ice nucleation active surface site density $n_{s,geo}$ ($n_{INP}$ scaled to the total geometric particle surface area) of $\geq 10^{10}$ m$^{-2}$ at -25 °C. This $n_{s,geo}$ value agrees reasonably well with estimates from our field survey. Therefore, the usage of OLLF surface materials as dust surrogates was verified in our AIDA-controlled laboratory study. Thus agricultural fields, especially OLLF, might represent important INP sources if these particles rise to sufficient height (i.e., cloud altitude) in the atmosphere. New data on the ice nucleation properties of OLLF dust at heterogeneous freezing temperatures above -29 °C were generated. Moreover, we successfully characterized physical, chemical, and biological properties of OLLF dust samples, finding that their IN properties remain unchanged after dry-heating at 100 °C but a subset of analyzed samples are influenced by boiling. Ice crystal residuals, or INPs that remain after the evaporation of water content, were rich in organics and low in salts. These findings imply the importance of heat-insensitive organics in OLLF dust INPs. Surprisingly, no known ice-nucleating microorganisms were found in our OLLF proxy samples. This negative result suggests that proteinaceous and biological ice-nucleating components are not the primary source of INPs from OLLFs. However, more systematic and careful studies are necessary to gain further insight into aerosol and INP properties (e.g., through analyses on ambient samples and ice crystal residuals from multiple seasons). In summary, we developed an INP parameterization for OLLF dust, which contributes to an improved understanding of INP emission and cloud microphysical processes in the supermicron-particle laden region. These OLLF INPs may directly influence the lifetime of supercooled clouds in a unique manner for this region. An application of our IN parameterization is crucial to explore the relationship between INP and supercooled cloud properties over such a predominant agricultural area.

## 1. Introduction

Atmospheric ice-nucleating particles (INPs) are a small subset of aerosol particles that initiate ice crystal formation in supercooled clouds (Vali, 1968; Chapter 9 of Pruppacher and Klett, 2010). While their importance, relevance, and perturbations to cloud and precipitation properties have been revealed by numerous past studies (e.g., Kanji et al., 2017 and references therein), the potential climatic impact of INPs and their representation in numerical models remain under debate (Storelvmo, 2017). One of the greatest challenges in the INP research field is the fact that INP sources are fast-changing worldwide in part due to the ongoing global

climate change (Murray et al., 2021). Thus, it is crucial to identify and characterize any perturbation sources that alter INP abundance and cloud-phase feedback.

Recently, a resurgence of "fertile-and-agricultural soil dust" (soil dust hereafter) INP research has been underway in part because of recent concerns regarding hydrological cycle alternation contributed by modern agricultural practices (Overpeck and Udall, 2020; Alter et al., 2015). Moreover, since agricultural practices represent a substantial dust emission source, accounting for up to 25% of total global dust emission (Ginoux et al., 2012), a large amount of INPs are globally anticipated from agricultural activities. Motivated by these reasons as well as earlier studies on ice nucleation (IN) of surface soil organic and biological samples

(Schnell and Vali, 1972; 1973), more recent studies utilized various online and offline instruments. In turn, scientists now have a better understanding of ambient INP concentrations ($n_{INP}$, per unit volume of air), especially through immersion freezing (i.e., the freezing propensity of INP immersed in supercooled water), from different agricultural sources (Conen et al., 2011; Hill et al., 2016; Steinke et al., 2016; Suski et al., 2018). These efforts allowed for the first-order estimates of immersion mode $n_{INP}$ from soil dust that is relevant to mixed-phase clouds (O'Sullivan et al., 2014; Tobo et al., 2014; Steinke et al., 2020). For instance, by

compiling the data from the Colorado State University continuous flow diffusion chamber and an ice spectrometer, the range of measured soil dust $n_{INP}$ at -20 °C from Suski et al. (2018) spanned from ~ 0.3 to 10 L$^{-1}$. Based on global mean aerosol particle concentrations and immersion mode IN parameterization, O'Sullivan et al. (2014) estimated the simulated $n_{INP}$ at -20 °C and 600 mb to range from ~ 0.01 to 8 L$^{-1}$. Similarly, Steinke et al. (2020) estimated that soil dust $n_{INP}$ can be as high as ≈ 40 L$^{-1}$ at -20 °C based on their laboratory-derived IN parameterization for soil dusts from Northwestern Germany and Wyoming, USA. Overall,

these measurements and approximations represent the upper bound of general field-studied $n_{INP}$ from different geographical areas summarized in Kanji et al. (2017; Fig. 1-10) in the same $T$ range, i.e. ~ 0.002 to 60 L$^{-1}$ (see **Sect. 3.6** for more detailed comparison discussion).

        Agricultural land use is in excess of 50% of total U.S. land use according to the U.S. Department of Agriculture, and there are > 26,000 "open-lot livestock facilities" (OLLFs) in the U.S. (Drouillard, 2018). The term OLLF is adapted to denote a particular

type of animal-feeding operation, in which cattle livestock is raised in outdoor confinement, as distinct from partially or totally enclosed housing, and also as distinct from pasture or free-range production systems (Auvermann et al., 2004). OLLFs are common in semi-arid and arid climates. Contrasted with the alternative production systems typical of wetter and more temperate climates, they (1) are an intensified form of livestock production, generating more marketable product per unit land area with less built infrastructure, (2) make use of the elevated evaporative demand to reduce or eliminate precipitation-generated wastewater that

must be controlled under water-quality regulations, and (3) capitalize on the nocturnal cooling characteristic of semi-arid and desert climates to avoid major investments in (and operating costs associated with) ventilation systems while still reducing the incidence and duration of livestock heat stress under most conditions.

        In particular, the Texas Panhandle (northern most counties of Texas; also known as West Texas) is a major contributor to the U.S. cattle production, accounting for 42% of fed beef cattle in the U.S. and 30% of the total cattle population in Texas (> 11

million head). Annually, these cattle produce > 5 million tons of manure, which represents a complex microbial habitat containing bacteria and other microorganisms, on an as-collected basis (Von Essen and Auvermann, 2005). Agricultural dust particles observed at OLLFs have long been known to affect regional air quality because the dust emission flux and 24-hour averaged ground-level dust concentration can be as high as 23.5 µg m$^{-2}$ s$^{-1}$ and 1,200 µg m$^{-3}$ (Bush et al., 2014; Hiranuma et al., 2011). Furthermore, our previous study revealed a presence of OLLF-derived particles at 3.5 km downwind of the facility, suggesting

their ability to be transported regionally (Hiranuma et al., 2011). Moreover, some recent studies suggest that aerosol particles emitted from agricultural activities might reach cloud heights due to wind erosion, scouring, and other relevant mechanisms (Steinke et al., 2020 and references therein; Duniway et al., 2019; Katra, 2020).

        Due to the potential to act as a prevalent point source of microbiome-enriched dust particles in the Southern High Plains region, where a convective cloud and updraft system persists (Li et al., 2017), we hypothesized that an OLLF can be a source of

100 soil dust INPs. To verify this hypothesis, IN propensities of aerosol particles from OLLFs, IN efficiencies of OLLF proxies, and their physicochemical and biological properties were studied in both field and laboratory settings. Specifically, we examined the immersion mode IN ability of ambient OLLF dust (sampled in the field and analyzed in an offline lab setting) and surface-derived material samples aerosolized in the cloud simulation chamber. We focused on the immersion mode freezing because recent modeling simulation and remote sensing studies suggest that immersion freezing is the most prominent heterogeneous IN

mechanism, accounting for 85 to 99%, through which ice crystals are formed in mixed-phase clouds (Hande and Hoose, 2017; Westbrook and Illingworth, 2011). OLLF-emitted particles are known to include substantial amounts of organic materials. Our previous work using Raman micro-spectroscopy revealed that ≈ 96% of ambient aerosol particles sampled at the downwind edge of an OLLF contain brown or black carbon, hydrophobic humic acid, water soluble organics, less soluble fatty acids and carbonaceous materials mixed with salts and minerals (Hiranuma et al., 2011). Recently, organic acids (i.e., long-chain fatty acids)

and heat stable organics were found to act as efficient INPs (DeMott et al., 2018; Perkins et al., 2020). However, our knowledge regarding what particular features of OLLF dust trigger immersion freezing at heterogeneous freezing temperatures ($T$s; i.e., size vs. composition) is still lacking. To improve our knowledge, we conducted single-particle composition analyses of different types of OLLF-derived ice crystal residual (ICR) samples. Finally, our study attempted to investigate the presence of any known biological INPs by taxonomic identification of the IN-active microbiome and by comparing the IN ability of heat-treated samples

to non-heat-treated ones. In general, cattle manure hosts a wide variety of bovine rumen bacteria (e.g., *Prevotellaceae*, *Clostridiales*), lipoprotein components of certain bacterial cell walls, and non-bacterial fauna of the rumen, such as fungal spores,

lichens, fungi, *Plantae*, *Protista*, *Protozoa*, *Chromalveolata,* and *Archaea* (Nagaraja, 2016). Hence, we examined if any IN-active cattle bovine microorganisms or associated fragments could be identified when aerosolized.

## 2. Materials and Methods

### 2.1. Field description

Four commercial OLLFs, ranging from 0.5 to 2.6 $km^2$ (< 45,000 head capacity), located in the Texas Panhandle region were used as the ambient aerosol particle sampling sites. All four sites are located within a 53 km radius of West Texas A&M University in Canyon, Texas. Our experimental layouts at each site, denoted as OLLF-1 to OLLF-4, are shown in **Fig. 1** (no further specification is provided to protect location privacy). All sites have a capacity greater than 1,000 head, which are categorized as large concentrated animal feeding operation facilities for cattle under the U.S. Environmental Protection Agency's definition. These OLLFs were selected primarily for the east-west orientation of their feeding and working alleys, which were nearly orthogonal to prevailing south to southwest winds, allowing for downwind and upwind sampling. Our sampling sites represent typical OLLFs, as more than 75% of cattle are produced in large concentrated animal feeding operation facilities in the U.S. (Drouillard, 2018).

Aerosol particles were collected to assess immersion freezing properties of "ambient" OLLF samples using offline immersion assays. These field samples were collected using 47 mm Nuclepore filters (Whatman, Track-Etched Membranes, 0.2 μm pore) through polycarbonate filter samplers. A filter holder was deployed at ~ 1.5 m above the ground. The filter sampling conditions measured locally (during individual sampling activities) are summarized in **Table 1**. Our samples were collected in different meteorological seasons, including summers in 2017 – 2019, springs in 2018 and 2019, and winter in 2019, in order to examine the seasonal variation in $n_{INP}$. In 2017, polycarbonate filter samplers were used at both upwind and downwind edges (< 80 m away from OLLF pens) of OLLF-1, 2, and 3 to understand the spatial variation in $n_{INP}$ within facilities (**Fig. 1**). Our sampling durations varied, but were up to ~ 4.5 hours, and our final IN propensity results were scaled to the sampled volume of air afterwards (**Table 1**). All filter samples were kept in sterilized tubes refrigerated at 4 °C until the immersion freezing measurements commenced (typically within 24 hours after sampling).

To complement the polycarbonate filter samplers, simultaneous 1-min time-resolved mass concentration measurements of $PM_{10}$ ($PM_x$ = particulate matter smaller than x μm) during individual sampling intervals were also carried out using DustTrak particulate monitors (TSI Inc., Model 8520) equipped with a $PM_{10}$ inlet. Additionally, tapered-element oscillating microbalances (TEOMs; Thermo Scientific Inc., Model 1400ab; Patashnick and Rupprecht, 1991) were deployed at OLLF-1 to continuously monitor $PM_{10}$ mass concentration side-by-side with a polycarbonate filter sampler and DustTrak. With an operating flow rate of 16.7 LPM, our TEOM measured < 1 g $m^{-3}$ of PM with a 5-min time resolution. Two identical TEOMs were deployed at OLLF-1: one at the upwind edge and another at downwind location of OLLF-1 (**Fig. 1**). Both TEOMs ran continuously during the entire 2016 – 2019 study period except for routine maintenance activities. The inlets of DustTrak and TEOMs were maintained at ~ 1.5 m above the ground to be consistent with our polycarbonate filter samplers. It is noteworthy that our TEOM and DustTrak $PM_{10}$ measurements agreed within ± 40% on average.

### 2.2. Surface samples for laboratory-based experiments

Two types of OLLF surface-derived materials, namely Texas-Dust-01 (TXD01) and Texas-Dust-05 (TXD05), were used as surrogates for dust particles observed at the downwind location of OLLFs in Texas. These proxy samples were used in our laboratory study at the Aerosol Interaction and Dynamics in the Atmosphere (AIDA) facility. TXD01 is a composite sample of surface soils from several OLLFs located in the Texas Panhandle. The other sample (TXD05) originates from a research feedlot in McGregor, TX. Both samples represent a raw surface material composite from feedlot pens, where cattle are fed without antibiotics or probiotics. All samples were ground and sieved for < 75 μm in grain size. Physically pulverizing the surface samples simulates the primary emission mechanism and characteristic of OLLFs (Razote et al., 2006; Bush et al., 2014; von Holdt et al., 2021).

Dry-heated samples (i.e., ~ 100 °C oven-dried for 12 hours) of each type were analyzed in this study to assess the heat tolerance of INPs. Moreover, wet-boiled samples (i.e., filter samples suspended in pure water and boiled for 20 min; Schiebel, 2017) were also examined for their INP abundance using an offline freezing technique.

A summary of our sample physical properties is provided in **Table 2**. Briefly, bulk density values of all samples were measured using a gas displacement pycnometer (Quantachrome, 1200e Ultrapyc). As seen, all measured densities are almost identical. There is no systematic difference between non-heated material densities and pre-heated ones, which may be indicative of heat-resistant features, potentially due to pre-exposure to soil $T$ on average higher than ambient $T$ even at the depth of 150 mm during summer (Cole et al., 2009). Next, geometric specific surface area (*SSA*) values were computed based on AIDA aerosol particle size distribution measurements (i.e., fraction of total surface area concentration to total mass concentration estimated from the size distribution data; see next section and **Table 3**). Additionally, nitrogen adsorption-based *SSA* values, Brunauer-Emmett-Teller (BET) *SSA*, for all samples are also shown in **Table 2**. The Autosorb iQ model 7 gas sorption system (Anton Paar, former Quantachrome Instruments) was used to measure BET *SSAs* in this study. The measured BET *SSA* values of OLLF samples are slightly higher compared to those of previously measured agricultural soil dust samples (0.74 – 2.31 $m^2$ $g^{-1}$; O'Sullivan et al., 2014), which suggests that TXD01 and TXD05 are more porous than these previous soil samples, leading to higher BET *SSA*. On average, our geometric *SSA* value (± standard error) is 4.59 ± 0.81 $m^2$ $g^{-1}$, which is higher than the BET *SSA* values. As demonstrated in our

previous studies, a small *SSA* value is often consistent with the presence of a large aerosol particle population (Hiranuma et al., 2015). Hence, the predominance of larger particles in bulk powders assessed in BET is presumably responsible for the observed differences in these two *SSA* values (**Table 2**). Indeed, the particles observed in AIDA were all ≲ 6.5 μm volume equivalent diameter, $D_{ve}$ (**Table 3**), whereas the particles evaluated by BET were up to 75 μm. Therefore, in association with large grain size involved in the BET analysis, bulk samples might have exhibited smaller *SSA* than dry dispersed ones. Furthermore, our *SSA* measurements suggest heat-tolerance in our OLLF samples. We examined BET *SSAs* using two different degassing *T*s (55 °C and 200 °C) for each sample within ± 10% accuracy of the BET instrument. Geometric *SSAs* of non-heated and heated samples also agreed within given standard errors. Further discussions on representativeness of the surface samples used in this study compared to ambient OLLF soil dust are provided in **Sect. 3**.

As demonstrated in our previous study, the surface area distribution of ambient OLLF dust peaks in mode diameter at ~ 10 μm (i.e., Fig. 5 of Hiranuma et al., 2011). This mode diameter is larger than surface-derived samples aerosolized and examined in the AIDA chamber (**Table 3**). However, it is cautiously noted that the ambient OLLF dust size distribution is not spatially uniform, and the emitting mechanism itself is not controllable as it highly depends on a unit of mobile livestock. Granting the primacy of hoof action as the decisive emissions mechanism of OLLF dust as described in Bush et al. (2014), a more controlled laboratory experiment has been desired to characterize IN ability of OLLF soil dust. The difference mentioned above and the demand for controllable investigation motivated analyzing IN properties of both bulk samples (< 75 μm-sieved) and aerosolized samples (≲ 6.5 μm). Further results and discussions about representativeness of the surface samples used in this study compared to ambient OLLF soil dust are provided in **Sect. 3**.

## 2.3. AIDA laboratory study

We used the AIDA expansion cloud-simulation chamber (Möhler et al., 2003) and a set of analytical instruments at Karlsruhe Institute of Technology to conduct a laboratory campaign named TXDUST01 in 2018. This study aimed at investigating the immersion mode ice-nucleating properties and other characteristics of OLLF dust proxies. We chose the AIDA chamber as our study platform because it simulates ice formation in mixed-phase clouds in a controlled setting with respect to both *T* (± 0.3 °C) and humidity (± 5%; Fahey et al. 2014). This chamber generates artificial clouds and activates particles in a simulated atmospheric cloud parcel via expansion cooling. The air volume adjacent to the chamber wall in the 84 m³ vessel is much smaller in comparison to the actively mixed volume of the vessel. Hence, we neglect the so-called wall effect (e.g., particle wall deposition) in the AIDA experiment. The AIDA has been applied for the analysis of both ambient and lab-generated INPs and has facilitated characterization of many INP species with the IN efficiency uncertainty of ± 39% (Steinke et al., 2020; Ullrich et al., 2017; Niemand et al., 2012; Hoose and Möhler, 2012). Note that the AIDA results provided a validation of the other INP spectrometers employed in this study.

An overall AIDA experimental schematic is shown in **Fig. 2**. Our OLLF dust proxy sample was injected into the AIDA chamber in an aerosolized form through a rotating brush disperser (PALAS, RGB1000) followed by passing through a series of inertial cyclone impactor stages to limit particle size to < 10 μm in $D_{ve}$. Subsequently, the OLLF particle size distribution in the AIDA chamber was measured prior to each simulated adiabatic expansion experiment. Specifically, a combination of a scanning mobility particle sizer (SMPS, TSI Inc., Model 3080 differential mobility analyzer and Model 3010 condensation particle counter), an aerosol particle sizer (APS, TSI Inc., Model 3321), and a condensation particle counter (CPC; TSI Inc., Model 3076) collectively measured the total number and size distribution of aerosol particles at a horizontally extended outlet of the AIDA chamber (Möhler et al., 2006). As seen in **Fig. 2**, a set of complementary filter samples of the aerosol particles directly from the AIDA chamber was also collected prior to expansion experiments for three purposes: (1) examining the condensation/immersion freezing ability of aerosol particle collected on nitrocellulose membrane filters (Millipore HABG04700, nominal porosity 0.45 μm) in the dynamic filter processing chamber (DFPC; Santachiara et al., 2010), (2) using them to perform measurements with the IN Spectrometer of the Karlsruhe Institute of Technology (INSEKT; Schiebel, 2017; Schneider et al., 2021), and (3) conducting metagenomics analyses to study biological components of the aerosolized samples. Afterwards, each particle type (i.e., TXD01 and TXD05) was individually examined for its immersion freezing ability during expansion experiments. To complement the AIDA chamber immersion results, INSEKT was used for aerosol particles collected on 47 mm Nuclepore filters (Whatman WHA10417012, pore size 0.2 μm) as well as for < 75 μm sieved-bulk samples collected. The DFPC technique was also used to measure the number concentration, ice-activated fraction, and nucleation efficiency of the INPs under different *T* conditions and for different particle sizes (i.e., PM$_1$ vs. total) collected on filters. DNA sampling for metagenomics analysis to study biological components of the OLLF bulk samples was also conducted on aerosol particles collected on the Nuclepore filters through an independent inlet.

Another motivation for using the AIDA facility is its ice-selecting pumped counterflow virtual impactor (IS-PCVI; Hiranuma et al., 2016). As detailed in **Supplemental Information (SI) Sect. S1**, IS-PCVI separates ICRs from interstitial particles, including cloud droplets, at *T*s below -20 °C. Preserving ICRs, which are leftover INPs after the evaporation of water content, by the IS-PCVI is key for elucidating physicochemical identities of INPs. ICRs were collected using TEM-grids (Ted Pella Inc., 01844N-F/01896N-F/162-100), and also compared to the total aerosol particles collected directly from the AIDA chamber on Nuclepore™ filters (Whatman, Track-Etched Membranes, 0.2 μm pore size). More detailed information about our IS-PCVI experiments in this study is provided below. Offline single particle analyses were conducted using an electron microscope (JEOL, JSM-6010LA) equipped with an energy dispersive X-ray spectroscopy function. Through this unique capability and subsequent analyses of ICR samples, we obtained detailed information on ICR composition of individual residual particles. In addition, we

used a single particle mass spectrometer to characterize aerosol particle chemical compositions of our surface samples (presented in **SI Sect.** S2). Individual details of all lab and field instruments and techniques are introduced in sections below.

## 2.4. Offline immersion freezing experiment techniques

To assess the ambient $n_{INP}$ through samples collected in the field, we used an offline droplet-freezing assay instrument, the West Texas Cryogenic Refrigerator Applied to Freezing Test system (WT-CRAFT; Vepuri et al., 2021). Briefly, WT-CRAFT enables a simulation of atmospheric immersion freezing using supercooled droplets containing aerosol particles at $T$ > -25 °C. WT-CRAFT was a replica of NIPR-CRAFT (Tobo, 2016), but the two systems currently possess different sensitivities to artifact and detectable $T$ ranges as described in Vepuri et al. (2021). In this study, for each ambient sample, we evaluated 70 solution droplets (3 μL each) placed on a hydrophobic Vaseline layer with a cooling rate of 1 °C min$^{-1}$. All droplets were prepared using filter rinse suspensions with high-performance liquid chromatography (HPLC)-grade water. The amount of HPLC water was determined based on the total amount of air sampled through the cross section of filter (**Table 1**), which limits the detection capability to 0.05 INP per L of air (standard $T$ and pressure, STP). As described in Vepuri et al. (2021), by optimizing the suspension water volume, the first frozen droplet observed was considered to have 0.05 INP L$^{-1}$ in this study. Each freezing event was determined optically based on the change in droplet brightness when the initially transparent liquid droplets became opaque upon freezing. If the freezing $T$ was not obvious for any droplets, the 8-bit grayscale images were assessed using ImageJ software to determine the $T$ of phase change. After the measurement, we calculated the frozen fraction and estimated the $n_{INP}$ per volume of air as a function of $T$, $n_{INP}(T)$, for every 0.5 °C following the parameterization described in Eqns. 1 – 2 of DeMott et al. (2017). As shown in Hiranuma et al. (2019, i.e., Table S2), the $T$ uncertainty in WT-CRAFT is ± 0.5 °C. The experimental uncertainty is typically represented by 95% binomial confidence intervals (CI95%). While the background freezing contribution of the field blank filter was negligible (< 3%) at -25 °C, we purposely limited our WT-CRAFT data analysis to the $T$ range between 0 °C and -25 °C to eliminate any possible artifacts in our WT-CRAFT data.

The INSEKT system is another offline immersion freezing technique used to assess the IN ability of surface OLLF samples collected on 47 mm polycarbonate filters at the AIDA facility. All filter samples were collected from the AIDA chamber prior to individual expansion experiments with a sampling flow rate of 10 L min$^{-1}$, and a total of ≈ 600 L of air was sampled through a cross section of each 47 mm polycarbonate filter (see **Table 3** for corresponding AIDA experiments). As described in Schiebel (2017), the design and concept of INSEKT is based on the CSU-IS instrument (Hill et al., 2014 and 2016). For INSEKT analysis, aerosol particles were washed off the filter and the resulting suspension is divided into volumes of 50 μL, which were placed in wells of a sterile PCR tray. It was then placed in an aluminum block thermostated with an ethanol cooling bath (LAUDA RP 890; Lauda), which was cooled down at a rate of 0.33°C min$^{-1}$. If a well froze upon the presence of an INP, a camera detected the brightness changes. The $T$ uncertainty of INSEKT was ± 0.5 °C, and the INP concentrations error was estimated by means of the binomial CI95% for each sample. The derivation of $n_{INP}$ based on Vali (1971) is described in **SI Sect. S4.** In this study, filter-collected aerosol particles were suspended in 8 ml filtered nanopure water, which has negligible contribution to background freezing, and used to characterize their IN efficiency (Schneider et al., 2021). Similar to WT-CRAFT, the amount of pure water to generate a stock suspension was adjusted for the first frozen aliquot-well observed to contain ≈ 0.015 INP L$^{-1}$ in this study, based on the total amount of air sampled through the cross section of filter. A series of diluted suspensions (×15 to ×225) was consistently analyzed for each sample to acquire an INP spectra covering a wide range of heterogeneous freezing $T$s (-7.5 °C to -25.5 °C). For the overlapping $T$s, we chose the data exhibiting the minimum CI95% as representative $n_{INP}$ for given $T$. In addition, **SI Sect.** S3 provides a comparison of our two immersion freezing techniques and results, which are reasonably comparable.

Condensation/immersion mode $n_{INP}$ were also measured at CNR-ISAC by means of DFPC (Santachiara et al., 2010). The DFPC chamber is a replica of the Langer dynamic developing chamber (Langer and Rogers, 1975). A systematic uncertainty in terms of $T$ in DFPC is within ± 0.1 °C (Table S1 in Hiranuma et al., 2019). With a water saturation error of ± 0.01, an ice detection error of ± 33%, and the experimental standard deviation, the overall IN efficiency uncertainties of DFPC are estimated to be less than ± 62% for this study. The application of DFPC for immersion freezing has been verified in previous inter-comparison studies (DeMott et al., 2018; Hiranuma et al., 2019). For the DFPC analyses, aerosol particles were collected on nitrocellulose black gridded membrane filters (0.45 μm porosity, Millipore) from the AIDA chamber prior to each expansion experiment (**Table 3**). Two parallel samplers employed in this study had an identical sampling flow rate of 2 L min$^{-1}$, and a total 100 L of air was sampled for each system. One sampling system collected the total aerosol particles, while another one was equipped with a cyclone impactor (MesaLabs, SCC0732, S/N 13864) to collect only submicron-sized aerosol particles. This impactor was characterized with a cut-off size around 1 μm in aerodynamic diameter (50% cut-off diameter at 0.9 μm) at 2 L min$^{-1}$ flow rate (Kenny, et al., 2000). Therefore, the latter line selectively collected particles smaller than 1 μm aerodynamic diameter. The cut-size efficiency of this cyclone impactor was tested in the lab against NaCl particles. Particle transmission efficiency along the total sampling line was taken into account by estimating gravitational losses in the horizontal tract of the sampling tube and inertial losses in the bend. At a particle size of 10 μm (larger than what was measured in the AIDA chamber), the overall particle transmission efficiency was higher than 86%. For a particle size of 2 μm, the particle loss is estimated to be ≈ 2.5%. Due to the small loss, we neglected any corrections for aerosol particle counts. After collection, the filters were safely kept in Petri dishes at room $T$ until the freezing experiments were initiated.

Prior to the DFPC measurement, the sampled filter was inserted onto a metal plate and covered with a smooth surface of paraffin in order to assure good thermal contact between the filter and the supporting substrate. Subsequently, the paraffin was

slightly heated and rapidly cooled in order to fill the filter pores. DFPC controlled the $T$s of the filter and the air, saturated with respect to finely-minced ice, with the flow continuously grazing the filter. IN measurements of total aerosol particles, $n_{INP,total}$, as well as, measurements of $PM_1$, $n_{INP,PM1}$, were performed at water supersaturation of 2%, and $T_{filter}$ of -18 °C and -22 °C. The supersaturation was calculated theoretically from vapor pressures over ice and water. The exposure time of the filter was 20 min to grow visible ice crystals on INPs at the considered $RH$ and $T$ condition. The use of the dynamic chamber is advantageous compared to other techniques as the supersaturation is maintained and less impacted by the effect of hygroscopic particles or ice crystal growth, which might lead to an incomplete activation of the INPs on the filter substrate.

**2.5. Extraction of total DNA from bulk and aerosolized dust samples**

Total DNA was extracted from Texas dust samples TXD01 and TXD05 prior to and after aerosolization in the AIDA cloud chamber. From bulk samples of dust, total DNA was extracted from 157.1 mg (TXD01) and 128.8 mg (TXD05). To sample aerosolized dust from the AIDA cloud chambers, stainless steel filter holders containing nucleopore filters (47mm diameter and 0.2 μm pore size) were used. These filters were previously sterilized in a standard vapor autoclave and fitted onto the AIDA cloud chamber for aerosol particle sampling prior to the expansion IN experiment. After the conclusion of the experiments, the holders were removed from the chamber to extract total DNA directly from the nucleopore filters. DNA extractions were performed using the FastDNA® Spin Kit for Soil (MP Biomedicals) as described in the manufacturer's protocol. Filters were aseptically removed from holders and placed in the Lysing Matrix E tube for mechanical cell disruption, which was carried out with the FastPrep® Instrument (MP Biomedicals). The concentration and purity of the extracted DNA was measured by using the Qubit™ 3.0 (Thermo Fisher Scientific). The volume of each sample was 50 – 100 μL.

Next, our metagenomics analysis method of total DNA is described. The amplification of phylogenetic marker genes and the metagenomics analysis of amplicons from each dust sample were performed by Eurofins Genomics Germany GmbH using the INVIEW Microbiome Profiling 3.0 protocol in order to identify and classify the microbial population (*Fungi*, *Bacteria*, and *Archaea*) of each sample. To achieve this, the hypervariable regions V1 – V3 and V3 – V5 of the bacterial 16SrRNA gene, the fungal internal transcribed spacer (ITS2) gene and part of the archaeal 16SrRNA gene were amplified by polymerase chain reaction from each sample using in-house primers. Amplicons were sequenced with the MiSeq next generation sequencing system with the $2 \times 300$ bp paired-end read module.

As the first step of the microbiome analysis, all reads with ambiguous bases ("N") were removed. Chimeric reads were identified and removed based on the de-novo algorithm of UCHIME (Edgar et al., 2011) as implemented in the VSEARCH package (Rognes et al., 2016). The remaining set of high-quality reads was processed using minimum entropy decomposition (MED; Eren et al., 2013 and 2015). MED provides a computationally efficient means to partition marker gene datasets into operational taxonomic units (OTUs). Each OTU represents a distinct cluster with significant sequence divergent from any other cluster. By employing Shannon entropy, MED uses only the information-rich nucleotide positions across reads and iteratively partitions large datasets while omitting stochastic variation. The MED procedure outperforms classical identity-based clustering algorithms. Sequences can be partitioned based on relevant single nucleotide differences without being susceptible to random sequencing errors. This allows a decomposition of sequence datasets with a single nucleotide resolution. Furthermore, the MED procedure identifies and filters random "noise" in the dataset, i.e., sequences with very low abundance (less than 0.02% of the average sample size).

To assign taxonomic information to each OTU, DC-MEGABLAST alignments of cluster-representative sequences to the sequence database were performed. The most specific taxonomic assignment for each OTU was then transferred from the set of best-matching reference sequences (lowest common taxonomic unit of all the best matches). A sequence identity of 70% across at least 80% of the representative sequence was the minimal requirement for considering reference sequences. Further processing of OTUs and taxonomic assignments was performed using the QIIME software package (version 1.9.1, http://qiime.org/). Abundances of bacterial taxonomic units were normalized using lineage-specific copy numbers of the relevant marker genes to improve estimates (Angly, 2014). Taxonomic assignments were performed using the NCBI_nt reference database (Release 2019-01-05).

**2.6. $n$INP estimation and IN parameterization method**

All IN data from AIDA, WT-CRAFT, INSEKT, and DFPC experiments were converted to and stored in $n$INP($T$), INP concentration per unit aerosol particle mass [$n_m(T)$], and INP concentration per unit aerosol particle surface as a function of $T$ [$n_{s,geo}(T)$] (DeMott et al., 2017; Ullrich et al., 2017; Hiranuma et al., 2015). The derivation process of these quantities are summarized in **SI Sect. 4**. These conversions required only scaling measured or estimated $n_{INP}(T)$ from each method to aerosol particle mass or surface area parameters provided in **Tables 1–3**. Niemand et al. (2012) infers that the application of $n_s$ is valid for small percentages of IN active fraction ($\leq 1\%$). From the numbers of $N_{total,0}$ given in **Table 3** (total number concentration of particles at the initial stage prior to expansion), we examined on average ~ 200,000 $L^{-1}$ aerosol particles in the immersion freezing mode in AIDA. Even assuming we evaluate INP up to 2,000 $L^{-1}$, our INP fraction is 1%. Thus, our $n_s$ parameterization is reasonable.

A consistent data interpolation method is important to systematically compare immersion freezing data from different IN measurement methodologies. In this study, we present $T$-binned-average IN data (i.e., 0.5 °C bins) for the lab and field IN data.

By following the inter-comparison method described in our previous studies (Hiranuma et al., 2015), all lab data were binned/interpolated in a consistent manner using a 0.5 °C resolution data.

## 3. Results and Discussion

### 3.1. Ambient INP spectra

To evaluate the immersion freezing efficiency of ambient aerosol particles collected at OLLFs, we converted our WT-CRAFT-based INP measurements to ice-nucleating efficiency metrics, such as $n_{INP}$, $n_m$, and $n_{s,geo}$ (**SI Sect. 4**). Individual values of cumulative mass (derived from DustTrak measurements), $n_{INP}$, and $n_m$ for each sampling date are provided in **Table 1**. On average, an extremely high cumulative $n_{INP}$ at -25 °C of 1,171.6 ± 691.6 L$^{-1}$ (standard error) L$^{-1}$ was found at the downwind site. **Figure 3a** shows the $n_{INP}$ comparison between downwind samples and upwind samples collected simultaneously at OLLF-1, 2, and 3 in 2017. Additionally, **Fig. 3b** summarizes the $n_{INP}$ diversity between downwind and upwind in $\log(n_{INP,downwind}/n_{INP,upwind})$, which represents the log-scaled ratio of individual measurements at each OLLF site at given $T$s. These $n_{INP}$ ratios are shown only for the $T$ range covered by both downwind and upwind data. As can be seen in these two panels, none of upwind spectra show $n_{INP}$ above -14 °C whereas we detected $n_{INP,downwind}$ at $T$s above -10.5 °C, suggesting that the INPs that are active at $T$s above -14 °C originate in OLLFs. In fact, across the examined freezing $T$s, the downwind spectra from all OLLFs exhibit higher $n_{INP}$ than the upwind spectra; therefore, the $\log(n_{INP,downwind}/n_{INP,upwind})$ values are above zero at $T$s below -14 °C. The source of upwind INPs is unknown. However, because the measured $n_{INP}$ is low at high $T$, the CI95% error of $n_{INP,upwind}$ at around -15 °C is relatively large as compared to that at a lower $T$ (**Fig. 3a**). Hence, the difference between $n_{INP,downwind}$ and $n_{INP,upwind}$ is not conclusive beyond the uncertainty at this $T$. Furthermore, since our polycarbonate filter samplers were deployed in the close proximity of livestock pens (< 80 m away as discussed in **Sect. 2.1**), the influence of soil dust even at an upwind site could not be ruled out depending on local meteorological conditions and livestock activities. Thus, it may be possible that a short episode of soil dust results in high $n_{INP}$ at a specific $T$ range for the upwind sample. Nonetheless, the downwind $n_{INP}$ values are indeed higher than $n_{INP,upwind}$ (beyond uncertainties) at $T$s below -20 °C. At -25 °C, all $n_{INP,downwind}$ values appear to be an order magnitude higher than the upwind ones without any exceptions, indicating that OLLF is a source of a notable amount of INPs across the examined $T$ range.

Shown in **Fig. 4** is a compilation of $n_{INP,downwind}$ based on the sampling season (i.e., summer, spring, and winter). Overall, we detected INPs at $T$s lower than -5 °C, and the range of $n_{INP,downwind}$ at -20 °C varied in different seasons in 2017 – 2019: summer (5.0 – 421.7 L$^{-1}$), spring (4.2 – 31.2 L$^{-1}$), and winter (0.9 – 20.4 L$^{-1}$). As inferred from **Fig. 4**, this seasonality holds true for all investigated $T$s. The observed seasonal variation in $n_{INP}$ corresponds to that in cumulative PM mass (**Table 1**). We observed a prominent linear relationship between aerosol particle mass and INP number concentration (at -25 °C: **Fig. 5a**). Further, the $n_{INP}$ values scaled to the mass ($n_m$: **Fig. 5b**) show a nearly constant value ($\approx 3 \times 10^9$ g$^{-1}$) at -25 °C (independent of particle mass concentration). These results imply the following: (1) ambient meteorological conditions, as summarized in **Table 1**, might not be determining factors for $n_{INP}$ for our study sites; (2) there is a predominance of supermicron INPs from the feedlot, which dominates particle mass.

**Figure 6** depicts the $n_{s,geo}$ spectra of aerosol particles from OLLF downwind ambient samples, color-coded with different sampling seasons. As seen in the figure, the seasonal diversity of $n_{s,geo,downwind}$ is less apparent as compared to that of $n_{INP,downwind}$ (**Fig. 4**). There is no systematic difference in the range of $n_{INP,downwind}$ in different seasons in 2017 – 2019 at -20 °C: summer (6.7 $\times 10^7$ – 2.7 $\times 10^9$ m$^{-2}$), spring (2.4 $\times 10^8$ – 2.3 $\times 10^9$ m$^{-2}$), and winter (1.2 $\times 10^8$ – 2.9 $\times 10^8$ m$^{-2}$). This observation is consistent with the prescribed dominance and importance of large particles as soil dust INPs.

Overall, our offline measurements of ambient $n_{INP}$ using field filter samples collected in OLLFs show more than several hundred INPs L$^{-1}$ at below -20 °C. More interestingly, there is a notable correlation between INP and ambient aerosol particle mass concentrations based on our 2017 – 2019 field study, which indicates the importance of large supermicron aerosol particles as INPs. This motivates the need for further characterization of our OLLF samples in a controlled-lab setting in order to identify what particulate size population (i.e., supermicron vs. submicron) and other properties trigger their IN in a controlled lab setting.

### 3.2. IN efficiencies of surface materials

As shown in **Table 3**, we conducted 10 AIDA experiments to measure IN efficiency of two surface materials; TXD01 and TXD05. Dry-heated samples of each type were also examined: TXD01H and TXD05H. All lab data associated with this study were archived according to the AIDA experiment number (i.e., TXDUST01_number), and we share these IDs with other associated measurements (e.g., INSEKT). As seen in **Table 3**, the mode diameters of TXD01 samples in AIDA were in general smaller than that of TXD05 samples, which is consistent with our *SSA* measurements (see **Table 2**). Shown in **Fig. 7** are expansion experiment profiles of these 10 experiments with different samples, including TXD01 (i) – (iii), TXD05 (iv) – (vi), TXD01H (vii – viii), and TXD05H (ix – x). These profiles represent data points measured in the chamber over a series of time, such as $T$ (a), pressure (b), relative humidity (*RH*, c), and aerosol particle and hydrometeor concentration (d) for each AIDA experiment. For a cloud formation experiment, the pressure within the chamber was reduced ($\Delta P \sim 180 – 290$ hPa), causing the $T$ to drop and a simulated adiabatic 'expansion' to occur. As can be seen, measurements were made by AIDA-simulated immersion freezing at water saturation (*RH* with respect to water around 100%). A droplet-ice threshold typically coincides with $\geq 20$ μm $D_{ve}$ (Hiranuma et al., 2016). Thus, the number concentration of > 20 μm $D_{ve}$ AIDA particles measured by a welas optical particle counter (Benz et al., 2005) primarily represents pristine ice crystals formed during the expansion (**Figs. 7d**). The *RH* dropped during some expansions at low $T$s (**Figs. 7c.iii and**

**7c.vi**). At these *T*s, ice crystal grow rather fast at the expense of available water vapor in the AIDA chamber, which causes the observed *RH* drop. Nevertheless, droplets were fully activated within ≈ 100 seconds of each expansion while reaching the peak *RH*, where we see the steep slope of $\Delta RH/\Delta t$ in **Fig. 7**. Further, as seen in **Fig. 7d**, particles of >20 µm $D_{ve}$ are not increasing and the total aerosol concentration measured by CPC also does not change after the *RH* peak. Thus, all predominant ice formation occurs at or before the *RH* peak through immersion freezing. Lastly, we made sure to only report our IN efficiency at *T*s higher than ~ -30 °C, corresponding to saturated condition in the AIDA vessel.

      **Figure 8** summarizes our $n_{s,geo}$ spectra of our surface material samples from the AIDA, INSEKT, and DFPC (total aerosol) experiments in comparison to six reference soil dust $n_{s,geo}$ spectra, O14, S16, S20, T14 (Wyoming), T14 (China), and U17, available in previously published studies (O'Sullivan et al., 2014; Steinke et al., 2016; 2020; Tobo et al., 2014; Ullrich et al., 2017), as well as our field data (**Fig. 3**). Untreated dry samples were assessed by all three techniques. Complementarily, INSEKT was also used to assess immersion freezing efficiency of wet-boiled (i.e., heated) filter samples. As explained in **Sect. 2.4**, a series of diluted samples were examined in INSEKT. We made sure to assess overlapping *T* intervals in a series of measurements to see if $n_{s,geo}$ values from multiple measurements agree within CL95% and, if so, to merge the results together. For each sample, the spectra nearly overlap each other at *T* ~ -25 °C, verifying their comparability and complementing features.

      As seen in **Fig. 8**, our OLLF spectra are comparable to the previous soil dust $n_{s,geo}$ parameterization at relatively low *T* (e.g., the $n_{s,geo}$ value range in orders of magnitude from $10^9$ to $10^{10}$ m$^{-2}$ at around -25 °C). At *T* above -20 °C, the INSEKT results suggest that the bulk TXD01 sample is more active than filter-collected samples beyond the $n_{s,geo}$ uncertainty (**Figs. 8a and 8c**). On the other hand, the INSEKT analyses of TXD05 (**Figs. 8b and 8d**) and all other filter samples did not find a notable difference amongst all samples. Furthermore, the lab-derived immersion spectra of both surface materials are reasonably comparable to the minimum – maximum boundaries of our field $n_{s,geo}$ spectra for *T* > -25 °C. While the variability of $n_{s,geo}$ at a single *T* could vary several orders of magnitude, similar variations are found for both lab and field results, implying the similarity of freezing efficiencies of our lab and field samples. Without scaling to the surface area, $n_{INP}$ spectra exhibited a wide range of INPs over three orders of magnitude; e.g., -25 °C (10.07 to > 10,000 L$^{-1}$). These results suggest that (1) there is a difference in the INP abundance between bulk (< 75 µm-sieved) and aerosolized/filtered-samples for TXD01 ($\lesssim$ 6.5 µm; **Table 3**) presumably due to different properties in particles of these two size subsets (6.5 – 75 µm and $\lesssim$ 6.5 µm) and/or different amount of IN-active soil organic matter (Tobo et al, 2014), (2) different physicochemical properties found for our TXD05 samples may not impact their INP propensities, and (3) TXD05 might be more representative of atmospherically relevant dust (see **Table 2** and **SI Sect. S2**).

      Our DFPC-derived $n_{s,geo}$ values in **Fig. 8** agreed reasonably well with the INSEKT results at the measured *T*s within our error ranges. This comparability suggests that freezing ability is similar for condensation and immersion for our surface samples. More importantly, **Table 4** summarizes the comparison of the submicron vs. supermicron INPs for a set of eight samples measured at -18 °C and -22 °C by DFPC. Due to limited range of *T*s and samples assessed by DFPC, we cannot provide any statistical variability of our individual data. But, on average, the supermicron INP fraction, given by $[(n_{INP,total} - n_{INP,PM1}) / n_{INP,total}] \times 100$, shows that this fraction contributed 49.7% ± 6.0% (average ± standard error) of total INP for TXD01 and TXD05 samples at the measured *T*s. This highlights the importance of the coarse fraction in the INP population. Note that we also compared the submicron vs. supermicron $n_{s,geo}$ values. Our PM$_1$ $n_{s,geo}$ and supermicron $n_{s,geo}$ were virtually identical, implying non-size dependent IN ability across the sizes evaluated in this study. Since DeMott et al. (2010) successfully demonstrated the correlation between immersion-mode $n_{INP}$ and the number concentration of aerosol particles larger than 0.5 µm diameter based on the compilation of field data for more than a decade, a number of studies have shown the evidence that supermicron aerosol particles dominate INPs across the world. For example, Mason et al. (2016) reported a substantial fraction of supermicron INPs through immersion freezing at relatively a high *T* (> 78% at -15 °C) measured at seven different sites over North America and Europe. Even at -20 °C, the author reported the fraction of supermicon INPs larger than 50%. Compared to these numbers, our laboratory data show lower fractions, but the INP sources are presumably different. Based on findings from recent study of size-resolved INPs vs. fluorescent biological particles, these INPs activated at −15 ℃ are typically thought to be biological (e.g., Huffman et al., 2013; Huang et al., 2021). While there has been more evidence that terrestrial and marine biological particles play an important role in immersion freezing of supermicron-sized particles (e.g., Ladino et al., 2019; Si et al., 2018; Creamean et al., 2018), the atmospheric implication of such rare aerosol species and the overall impact on aerosol-cloud interactions is still under debate. More recently, high IN efficiency by supermicron INPs derived from quartz-rich atmospheric mineral dusts have been reported from different locations, including East Asia (Chen et al., 2021) and eastern Mediterranean (Reicher et al., 2019). These mineral components usually contribute to IN at low *T*s. However, there has not been much discussion of large soil dust particles, especially organics, and their contribution to atmospheric ice nucleation in previous studies. Hence, direct implications of which components contribute to IN at different *T*s to the observed freezing properties of OLLF particles is still missing. Lastly, while we did not see a systematic increase of supermicron INP fraction as a function of *T* as shown in Mason et al. (2016; i.e., INP fraction at -15 °C larger than at -20 °C), our results in **Table 4** support that $n_{INP,total}$ is always higher than $n_{INP,PM1}$ for any type of samples used in this study.

Interestingly, our comparison between non-heated vs. heated samples indicated no substantial suppression in IN ability by heating, especially for dry-heated samples. This heat-resistant feature of OLLF samples may be due to their pre-exposure to dry, high ambient and soil *T* conditions (Cole et al., 2009). Further, our mass spectrometry analysis on these two subsets revealed no significant deviation in chemical compositions (**SI Sect. S2**). Additionally, our metagenomics analysis also found no deviation in terms of bacteria and fungi speciation between dry-heated and non-heat-treated samples as discussed below. A detailed comparison of the non-heat-treated sample to the heated-sample is discussed in **Sect. 3.6** and **SI Sect. S3**.

### 3.3. Metagenomics analysis

**Table 5** summarizes our results of metagenomics analysis. The diversity of the microbiome in the dust samples identified microorganisms common in soil, bovine manure, and inhabitants of the bovine rumen, as expected (detailed in **SI Sect. S5**). Interestingly, no known IN-active species of microorganisms (active at $T$s above -10ºC) were detected, although genera of *Bacteria* (*Pseudomonas*) and *Fungi* (*Fusarium*, *Mortierella*) known to include species with IN activity were detected, albeit in negligible numbers. This insignificance of IN-active microbiome and relatively high importance of non-biological supermicron particles as OLLF-INPs are deemed robust. Otherwise, the observed strong mass dependency of OLLF- $n_{INP}$ (**Fig. 5a**) cannot be explained as microorganisms typically contains small mass. We also found very little difference in the bacterial and eukaryotic metagenome in bulk and heat-treated dust samples (no data for *Archaea* were obtained from heat-treated dust samples). Heat treatment of dust samples at 100 °C for 12 hours apparently did not destroy the DNA in our samples, even though most microbial cells were killed. Thus, no notable difference after dry-heating was observed for both TXD01 and TXD05 (**Table 5**). This negative result is important because it agrees with our metagenomics analysis, where no known IN-active bacteria were detected. The diversity of the bacterial microbiome in both samples showed a considerable difference after aerosolization of dust in the AIDA cloud chamber and the subsequent IN experiments in simulated clouds. In aerosolized dust, a significant increase of desiccation-resistant *Actinobacteria* was observed in both samples. Further, we also identified a significant decrease of non-desiccation-resistant *Proteobacteria*, *Firmicutes*, and *Bacteroides* in aerosolized particles (**Table 5**). This result implies that aerosolization and microbial dispersion in the atmosphere may alter microbiome diversity and population, at least for our samples. This unique effect was not observed for *Fungi* and *Archaea* (see **SI Sect. S5** for more details).

### 3.4. Ice residual analysis

A total of 1,259 aerosol and residual particles in the diameter range of 0.2 to 3 µm were assessed through electron microscopy for their physicochemical properties. All of our single particle analyses were carried out with the following parameters: electron beam accelerating voltages of 15 keV, spot size of 50, and working distance of 10 mm. **Table 6** summarizes the size properties of analyzed particles. The number of measured particles was limited depending on the particle availability on each substrate. Nevertheless, we examined at least 100 particles for each sample type, as seen in the table. Out of these particles, the diameter of TXD01 (0.84 µm) particles was on average smaller than TXD05 (1.05 µm). This observation is consistent with our offline particle characterizations (**Table 2**) and the AIDA size measurements (**Table 3**). For the samples used in this study, we could not identify any systematic differences between aerosol particles and residuals in terms of size. Likewise, while we found substantial fractions of supermicron diameter particles in TXD01 (29.2%) and TXD05 (38.8%), there is no obvious enrichment in supermicron population in our ice crystal residuals from this study (**Table 6**).

Higher aspect ratios in residuals compared to aerosol particles were found for both TXD01 and TXD05 samples. This difference indicates a relative increase in non-spherical particles, that have a higher aspect ratio, in residuals. In short, Hiranuma et al. (2008) found that quasi-spherical OLLF particles were predominantly salt-rich hygroscopic particles, whereas non-spherical amorphous particles were found to be organic-dominant with negligible hygroscopicity. Thus, our results suggest the inclusion of non-hygroscopic particles as ice residuals.

Next, the elemental composition from energy dispersive X-ray spectroscopy analysis revealed some notable differences between aerosol particle samples and residual samples. In this study, we followed the H13 classification scheme to define particle types in the electron microscopy analysis (Hiranuma et al., 2013). Briefly, we semi-quantitatively assessed atomic weight percentage of organic (C, N, O), salt-rich (Na, Mg, K, P), mineral-rich (Al, Si, Ca), and other. We detected carbon in all particles exclusively, but a background signal from polycarbonate substrate film could not be separated and ruled out. **Table 7** shows the summary of particle types based on their elemental compositions for samples used in this study. It should be noted that the "rich" used in the names of particle classes only indicates intensive characteristic peaks in the energy dispersive X-ray spectra, and > 99.9% of particles (except a few aluminosilicate particles) examined in this study were predominantly composed of carbon elements as organics-mixed particles. As seen in the table, an increase in exclusively organic fractions as well as a substantial decrease in salt-rich particles in residuals persisted for both TXD01 and TXD05 samples. The organic type fraction in heated-aerosols is slightly smaller than that in non-heated aerosols. Nevertheless, the increase of organic type fraction for heated-ICRs implies an insignificant heating effect as well as the importance of heat-resistant organics for immersion freezing of OLLF materials. This observation supports the result in **Table 6**. The reduction in salt-rich particle percentage might be relevant to an increase in aspect ratio (Hiranuma et al., 2008). The observed relative increase in organic-including particles, which might be substantially less hygroscopic compared to salt-rich particles, is also indicative of the predominance of immersion freezing as an IN mechanism of OLLF particles (rather than condensation freezing; Belosi and Santachiara, 2019). Indeed, immersion is a dominant mechanism of IN in mixed-phase clouds (Hande and Hoose, 2017). Regardless, liquid cloud formation might be a prerequisite for activating OLLF particles as ice crystals in the atmosphere.

Finally, our attempts to analyze the size-resolved abundance of each composition class was not conclusive (not shown), possibly due to limitations in the small population examined. Nonetheless, finding no clear size-dependence of elemental compositions in both total aerosol and residual samples was an important negative result, which is consistent with findings through aerosol single particle spectrometry (**SI Sect. S2**).

### 3.5. Estimated INPs released from a OLLF

Upon confirmation of the comparability between field and lab $n_{s,geo}$ values, we proceeded with ambient $n_{INP}$ estimation based on our field mass concentration data, using the OLLF-1 TEOM $PM_{10}$ data. We elected to use the OLLF-1 data due to their reasonable spatiotemporal coverage (i.e., two identical model TEOMs deployed at the downwind and upwind sites for 2017 – 2019). A summary of TEOM mass concentration data in different seasons over 2017 – 2019 are available in **Table 8**. Frequently, the observed $PM_{10}$ concentration exceeded $10^{-7}$ g $L^{-1}$, which is consistent with previous studies (Bush et al., 2014; Hiranuma et al., 2011). On the other hand, the observed mass concentration at the upwind sites was typically substantially lower except for known/recorded interruptions (e.g., a tractor-trailer passing by), resulting in transient increase in mass concentration. As the upwind $n_{INP}$ can be considered non-negligible (see **Sect. 3.1**), we subtracted mass concentrations measured at a nominal upwind edge from the downwind TEOM mass concertation values to compute $PM_{10}$ from OLLF-1. The screened TEOM data were used as ambient particle concentration data to estimate $n_{INP}$ from an OLLF.

To estimate $n_{INP}$, we used the $n_{s,geo}$ parameterization given in **SI Sect. S6**. Due to the atmospheric relevance and $T$ coverage extending to -5 °C, we used a fit of ***Field_Median*** in **Table S3** to compute representative $n_{s,geo}$ relevant to OLLF. To convert $n_{s,geo}$ to $n_{INP}$, we have adapted Equations (1) – (3) in Hiranuma *et al.* (2015). Briefly, the measured mass concentration as well as field *SSA* were used to convert from $n_{s,geo}$ to $n_{INP}$:

$$n_{INP}(T)(L^{-1}) = n_{s,geo}(T)(m^{-2}) \times Geometric\ SSA \left(\frac{m^2}{g}\right) \times \text{Mass Conc.}\left(\frac{g}{L}\right). \tag{1}$$

where the geometric *SSA* value for field data, ~ 0.4 $m^2$ $g^{-1}$, is derived from particle size distribution measurements presented in Fig. 3 of Hiranuma et al. (2011).

**Table 8** summarizes the TEOM mass concentrations and estimated annual and seasonal $n_{INP}$ in different seasons over 2017 – 2019. In general, $PM_{10}$ mass concentrations from OLLF-1 (average ± standard errors) were high in meteorological summers ($3.9 \times 10^{-7} \pm 5.6 \times 10^{-8}$ g $L^{-1}$) and springs ($4.5 \times 10^{-7} \pm 2.4 \times 10^{-7}$ g $L^{-1}$) as compared to fall ($2.4 \times 10^{-7} \pm 4.4 \times 10^{-8}$ g $L^{-1}$) and winter ($1.5 \times 10^{-7} \pm 5.3 \times 10^{-8}$ g $L^{-1}$). A similar trend was found for the upwind $PM_{10}$ mass concentration: summer ($3.4 \times 10^{-8} \pm 9.0 \times 10^{-9}$ g $L^{-1}$) $\geq$ spring ($2.8 \times 10^{-8} \pm 9.3 \times 10^{-9}$ g $L^{-1}$) > fall ($1.8 \times 10^{-8} \pm 5.7 \times 10^{-9}$ g $L^{-1}$) $\geq$ winter ($1.4 \times 10^{-8} \pm 7.1 \times 10^{-10}$ g $L^{-1}$). But, the measured values at the upwind location are consistently an order magnitude lower than that from the downwind location.

On average, the estimated mean $n_{INP}$ values at -15, -20, and -25 °C in 2016 – 2019 were estimated as 46.8 (±25.3 seasonal standard deviation; same hereafter), 288.1 (± 156.1), and 5,250.9 (± 2,845.6) $L^{-1}$, respectively. In addition, the median $n_{INP}$ at -15, -20, and -25 °C in 2016 – 2019 were estimated as 14.7 (± 9.2), 90.9 (± 56.4), and 1,656.3 (± 1,028.1) $L^{-1}$, respectively. As our $n_{INP}$ is linearly scaled to mass concentration (Eqn. 1), estimated $n_{INP}$ showed a similar seasonal variability as seen in mass concentration. For instance, at -20 °C, the cumulative $n_{INP}$ averages for each meteorological season over three 2016 – 2019 were estimated as follows: spring ($315.4 \pm 164.9$ $L^{-1}$) $\geq$ summer ($270.4 \pm 39.0$ $L^{-1}$) > fall ($165.1 \pm 30.8$ $L^{-1}$) $\geq$ winter ($106.9 \pm 36.8$ $L^{-1}$). The observed high $n_{INP}$ values were expected for such a high $PM_{10}$ mass concentrations emitted from the cattle feedyard, which represent an important point source of agricultural aerosol particle emission. However, we reemphasize that the IN efficiency of OLLF aerosol particles is somehow similar to other agricultural aerosol particles found in previous studies as discussed in **Sect. 3.2** (**Fig. 8**).

**Figure 9** displays the TEOM mass concentration time series over 2017 – 2019 as well as cumulative $n_{INP}$ estimated at $T$s of -15 °C, -20 °C and -25 °C. The background mass concentration measured at the upwind location ($1.7 \times 10^{-8}$ to $2.6 \times 10^{-8}$ g $L^{-1}$) is shown with a red dashed line in **Fig. 9a** and subtracted from the downwind data. The resulting OLLF mass concentration was on average is $4.12 \times 10^{-7} \pm 2.96 \times 10^{-9}$ g $L^{-1}$ (or $411.57 \pm 2.96$ μg $m^{-3}$). Annual averages of OLLF mass concentrations are indicated with a blue dashed line in **Fig. 9a**. On average, the downwind concentration exhibited higher mass concentration by more than an order of magnitude. This result implies a constant high particle load from the OLLF, which was also seen by a previous study at the same OLLF (Hiranuma et al., 2011). Seasonal variation is also seen in **Fig. 9a,** as the annual peak of mass concentration (> $10^{-5}$ g $L^{-1}$) coincided with summer in each case.

**Figure 9b** shows associated $n_{INP}$ estimations. As seen in **Fig. 9b**, average estimated INPs at three different $T$s, -15 °C, -20 °C, and -25 °C, are shown as a gray dashed line, black dashed line and black solid line, respectively. Our results show that the aerosol particles downwind of a feedlot contain several thousand INPs $L^{-1}$ (median = 1,656 $L^{-1}$; average = 5,251 $L^{-1}$) at standard $T$ and pressure (STP) at -25 °C, which is three orders of magnitude higher than typical ambient $n_{INP}$ from continental sources as reported in DeMott et al. (2010). More discussion of OLLF $n_{INP}$ in comparison with previous studies is provided in **Sect. 3.6**.

Our lab and field measurements-based parameterizations open up further study opportunities to incorporate supermicron INPs from agricultural source in the atmospheric modeling simulation and may provide a hint to reveal the identity of INPs at relatively high $T$s (> -15 °C). Note that the existence of supermicron particles at cloud altitudes is especially non-negligible when we consider atmospheric immersion freezing, which initiates on the surface of a few in a million particles.

### 3.6. Comparison to previous soil dust IN studies

**Figure 10** summarizes our field measured $n_{INP}$ (**Fig. 4**) as well as estimated atmospheric $n_{INP}$ in the $T$ range between -5 °C and -25 °C (**Sect. 3.5**) in comparison to previously reported ambient $n_{INP}$ of soil dust and a compilation of other field-measured $n_{INP}$ from across the world. We purposely selected to display our estimated $n_{INP}$ with standard deviations and global reference field $n_{INP}$

data from Kanji et al. (2017) at their $T$ points (i.e., -15, -20, and -25 °C) to make all comparisons visible in this figure. It is clear that the estimated $n_{INP}$ from OLLF are within OLLF field-measured $n_{INP}$, implying that our $n_{INP}$ estimation is reasonable and atmospherically relevant. It is also apparent that the OLLF $n_{INP}$ spectra are consistently located above or overlapping with the upper bound of soil dust $n_{INP}$ spectra from previous studies across the $T$ range we examined in our field study (i.e., $T$ above -25 °C). Although our INP detection limit of 0.05 L$^{-1}$ in this study is not as good as Suski et al (2018; $\approx$ 0.002 L$^{-1}$), our data exceed their data from crop fields (soybean, sorghum, wheat, and corn) or are at least positioned towards the higher bound of the S18 data points. The observed consistent gap between our OLLF data and previous data holds true even when compared to the globally compiled $n_{INP}$ from multiple field campaigns at -15, -20, and -25 °C (Kanji et al., 2017), indicating that absolute INPs per unit volume at OLLF are much higher than previously investigated field INP sources. However, it is important to revisit our IN efficiency discussion included in **Sect. 3.2**. In short, our $n_{s,geo}$ values derived from surface materials as well as field OFFL samples are comparable to other reference soil and desert dust $n_{s,geo}$ (**Fig. 8**). Altogether, we conclude that OLLF soil dust is an important point-source of atmospheric INPs, which have comparable or higher IN efficiency compared to formally assessed soil dusts.

One unique aspect of our OLLF samples is their heat tolerance. Previously, Suski et al. (2018) found that heat-treatment (95 °C for 20 min) can suppress the $n_{INP}$ of wheat harvest soil dust sample from Kansas, USA by more than two orders of magnitude at -12 °C. The authors concluded that the decomposition of IN-active heat labile organics and bacteria is responsible for the observed $n_{INP}$ suppression. This result is consistent with the impact of heat treatment on the IN efficiency of soil dust samplew from different regions, such as the one from a lodgepole pine forest in Wyoming, USA (Hill et al., 2016; 105 °C for 20 min) and another from Central Yakutia (Conen et al., 2011; 100 °C for 10 min). Similarly, Tobo et al. (2014) found that the 300 °C combustion can reduce the IN fraction of Wyoming soil dust at -24 °C by the same orders of magnitude as Suski et al. (2018) observed. In contrast, Steinke et al (2016) found no notable effect of heat treatment (~ 110 °C) on the Argentinian soil dust IN efficiency at ~ -24 °C. This heat insensitive nature of Argentinian soil dust may have coincided with its lack of IN-active proteins and/or heat sensitive microbes, which aligns with the absence of known IN-active microbes in our OLLF samples (**Sect. 3.3**). Suppression of $n_{s, geo}$ for wet-boiled samples of TXD01H at $T$ above -20 °C can be found in **Table 5** and **Fig. 8c**. Nonetheless, the observed consistency in the spectral slopes suggests that lab and field measurements exhibit similar IN above examined $T$s. An example case of the negligible impact of the wet-boiling process on a field OLLF sample is discussed in **SI Sect. S3**. In total, our findings and the observation in Steinke et al. (2016) eliminate proteinaceous and biological ice-nucleating components as the primary source of IN abundance in air. The choice of 100-110 °C for heat treatment seems valid because proteinaceous structures will be destroyed below ~ 100 °C (Steinke et al., 2016). For example, Szyrmer and Zawadzki (1997) found some known cell-free IN-active microbes (e.g., Fusarial nuclei) are stable only up to 60 °C. Other than this study, ice nucleation activity by bacteria (Morris et al., 2004; Christner et al., 2008), fungi (Humphreys et al., 2001), and lichens (Henderson-Begg, et al., 2009) has been shown to be heat-sensitive irreversibly at 100 ºC or below. Other soil organic components can be decomposed at $T$ between 100 °C and 300 °C (Tobo et al., 2014).

**4. Conclusions**

This study was composed of two parts: (1) A multi-year field investigation of immersion-mode INPs from four commercial OLLFs in the Texas Panhandle in 2017 – 2019; (2) an AIDA laboratory campaign, which investigated the INP propensity and properties of two OLLF soil dust proxies. Our field and laboratory findings support that OLLFs are a substantial source of microbiome-enriched dust particles and soil dust INPs, which are estimated to exceed several hundred and several thousand INPs L$^{-1}$ at -20 °C and -25 °C, respectively.

From the first year of our field work, we found that OLLF is a source of INPs that can be active at $T$s below ~ -5 °C. Briefly, the analysis of log ratio of $n_{INP,downwind}$ to $n_{INP,upwind}$ from three different OLLFs consistently shows that the INP abundance at the downwind site of each OLLF is an order magnitude higher than at the nominal upwind edge across the examined $T$ range ($\geq$ -25 °C). This difference between downwind and upwind INPs clearly indicates that a vast majority of INPs found in our field sites (as high as 11,000 INP L$^{-1}$ cumulatively at -25 °C) are from OLLFs (**Table 1**). Over the three years of our field OLLF investigation, there was a clear seasonal variation in $n_{INP}$. Briefly, summer $n_{INP}$ at -20 °C from the downwind edge of OLLFs (5.0 – 421.7 L$^{-1}$) was notably higher than that of spring (4.2 – 31.2 L$^{-1}$) and winter (0.9 – 20.4 L$^{-1}$). The observed seasonal trend persisted for all heterogeneous freezing $T$s investigated in this study ($T \geq$ -25 °C). Interestingly, the observed $n_{INP}$ seasonality strongly correlated to that of PM$_{10}$ mass ($r = 0.94$). This relationship implies the importance of large particles, which dominate aerosol surface area and mass, on IN of OLLF dust. By scaling our $n_{INP}$ to the aerosol particle surface area, we are no longer able to see any clear seasonal variation in $n_{s,geo}$; thereby, we conclude that the abundance of INP from OLLFs depends on dust quantity at ground-level at given time, but its IN efficiency is consistent throughout the seasons at least for 2017 – 2019. These findings also suggest that future studies of soil dust INP might need to focus on statistically validating the link between large supermicron particles and INPs with longer observations from a multitude of regions, which might ultimately result in providing a simple IN parameterization for cloud and climate models.

The importance of large aerosol particles on immersion freezing, motivated by our field work, was verified in our controlled-AIDA laboratory study, using ground-collected samples from the OLLFs. The DFPC offline freezing instrument assessed IN abilities of OLLF dust surrogates with PM$_1$ and > PM$_1$ (total) size fractions, and revealed that on average $\approx$ 50% of OLLF $n_{INP}$ derived from supermicron aerosol particle population in the assessed $T$ range between -18 and -22 °C. Besides, several unique characteristics of OLLF INPs were disclosed. For instance, a comparability of results from our condensation freezing

instrument (DFPC) and immersion freezing assay (INSEKT) was found. A similar observation was previously made for another composition (mineral dust) in Wex et al. (2014); this similarity suggests that freezing ability is similar for condensation and immersion for our surface OLLF samples. Further, the comparability between immersion mode IN ability of ambient OLLF dust (sampled in the field and analyzed in the offline lab setting) and that of surface material samples aerosolized in the cloud simulation chamber sheds light on the representativeness of dried, pulverized surface materials as surrogates for ambient dust particles in immersion freezing tests (Boose et al., 2016). In short, our AIDA-INSEKT results for OLLF proxies reasonably agree with the range of our field-derived $n_{s,geo}$ values, validating the atmospheric relevance of our lab results (especially TXD05 regardless of varied particle size distributions and sample types; see **Table 3**). Additionally, the observed consistency in the spectral slopes (i.e., **Table 5**) suggests that lab and field measurements exhibit similar IN ability at examined $T$s.

Insignificance of dry-heating (100 °C for 24 hours) was demonstrated for both types of OLLF proxies. Previously, Steinke et al (2016) found no notable effect of heat treatment (~ 110 °C) on the Argentinian soil dust IN efficiency at ~ -24 °C. This heat insensitive nature of Argentinian soil dust might have coincided with its lack of IN-active proteins, which align with our lack of known IN-active microbiomes in our OLLF samples (**Sect. 3.3**). While suppression of $n_{s, geo}$ for wet-boiled (100 °C for 20 min) samples at $T$ above -20 °C was found for both proxies (**Fig. 8**), it is not conclusive how OLLF soil dust is susceptible to heat. An example case of negligible impact of wet-boiling process on a field OLLF sample is discussed in **SI Sect. S3**. In total, our findings and the observation in Steinke et al. (2016) eliminate proteinaceous and biological ice-nucleating components to be considered as the primary source of superb IN abundance from OLLFs. The future sampling of more ambient filters from multiple seasons and systematic analysis of non-heated vs. wet-boiled treatment of ambient samples may provide more conclusive idea of heat resistivity of ambient OLLF-INPs.

The predominance of organics with salt contents (e.g., potassium) in OLLF particle composition is consistent with our previous study of OLLF soil dust particle composition analyses (Hiranuma et al., 2011). Based on findings from this study, ICR analysis revealed a relative increase in organic inclusion (and decrease in salt inclusion) in residuals, highlighting the importance of organic material in OLLF-derived INPs for atmospheric immersion. Even after dry heating treatment, the increase in organic fraction was found in the ICR of our OLLF samples. Therefore, the investigation of heat-insensitive organics is key to further understand the properties of soil dust INPs, and further research should focus on understanding how organic composition influences IN. Our previous work using Raman micro-spectroscopy revealed that ambient aerosol particles sampled at OLLFs are internally mixed with brown or black carbon, hydrophobic humic acid, water soluble organics, less soluble fatty acids, and carbonaceous materials mixed with salts and minerals. But, our current knowledge regarding IN-active organics is still limited.

While we could not rule out the possibility of IN from TXD01 and TXD05 samples triggered by biological INPs, our current results did not support it. In the future, we also need to carry out an identical metagenomic analysis for ICR samples collected at various $T$s. Extracting enough DNA out of ICR samples would be challenging and is currently not feasible at the AIDA facility. Facilitating a dynamic cooling expansion chamber, and collecting ICRs for a prolonged expansion experiment period would be a potential resolution. Moreover, our metagenomics analysis indicated that most microorganisms were alive, but it did not provide any quantitative percentage. Therefore, we must do metatranscriptomics (analysis of RNA) in the future to examine gene expression in the microbial population. More interdisciplinary, collaborative studies (e.g., how the diet of cattle, inclusion of antibiotics, probiotics etc. influence INP abundance in samples of feedlot surface materials) would also be useful.

*Data availability.* Original data created for the study will be available in the Supplement upon publication.

*Supplement.* The supplement related to this article is available online at: www.atmospheric-chemistry-and-physics.net

*Author contributions.* N.H., N.S.U., B.W.A., and O.M. designed research; N.H., B.W.A., F.B., J.B., K.C., D.G., K.H., Y.H., H.S., X.S., I.S., N.S.U., F.V., L.L., and O.M. performed research; N.H., F.B., R.F., D.G., K.H., Y.H., G.S., X.S., I.S., N.S.U., H.S.K.V., F.V., L.L., and O.M. analyzed data; and N.H., F.B., D.G., and X.S. wrote the paper. N.H. led the revision effort with support of all authors.

*Competing interests.* The authors declare no conflict of interest.

*Acknowledgments.* The authors wish to thank the IMK-AAF engineering and infrastructure group (Georg Scheurig, Rainer Buschbacher, Tomasz Chudy, Olga Dombrowski, Jens Nadolny, Frank Schwarz, and Steffen Vogt) for their continued support throughout the TxDUST01 campaign. We also acknowledge Prof. Dr. Thomas Schwartz and Dr.-Ing. Johannes Alexander, Department of Microbiology and Molecular Biology/Institute of Functional Interfaces/Karlsruhe Institute of Technology for valuable technical assistance.

*Financial support.* This project has received funding from the European Union's Horizon 2020 research and innovation programme through the EUROCHAMP-2020 Infrastructure Activity under grant agreement No 730997. This material is based upon work supported by the U.S. Department of Energy, Office of Science, Office of Biological and Environmental Research under Award Number DE-SC-0018979. Naruki Hiranuma and Yidi Hou thank the Killgore Faculty Research and President's Undergraduate

Student Research Grants (WT20-034). This work was supported by Alexander von Humboldt – Stiftung (grant No 1188375) through postdoctoral fellowship for Nsikanabasi S. Umo.

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

**Figures and Tables**

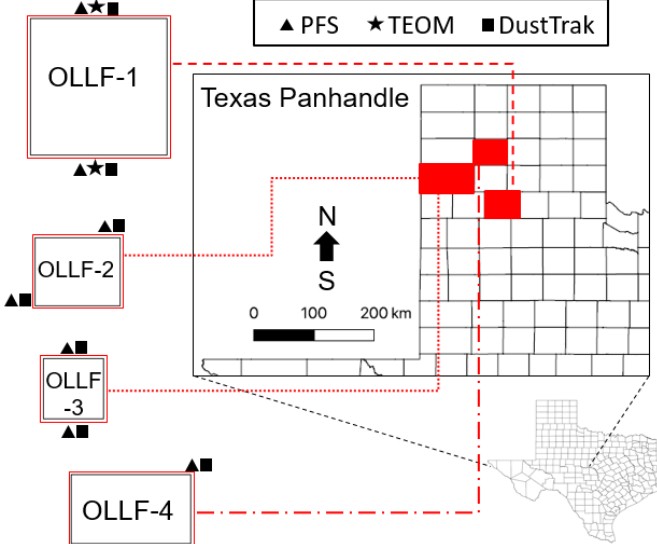

**Figure 1.** Schematic of the field sampling activity at individual sites (only the counties are shown). The dimension of each facility (east – west × north – south) is (1) 1.6 × 1.6 km, (2) 1.0 × 0.8 km, (3) 0.7 × 0.7 km, and (4) 0.8 × 1.4 km. A combination of polycarbonate filter samplers (PFSs) and DustTrak instruments was used at the nominally upwind and downwind edges of OLLF-1 to OLLF-3. Two tapered-element oscillating microbalances (TEOMs) were deployed at OLLF-1 alongside other instruments.

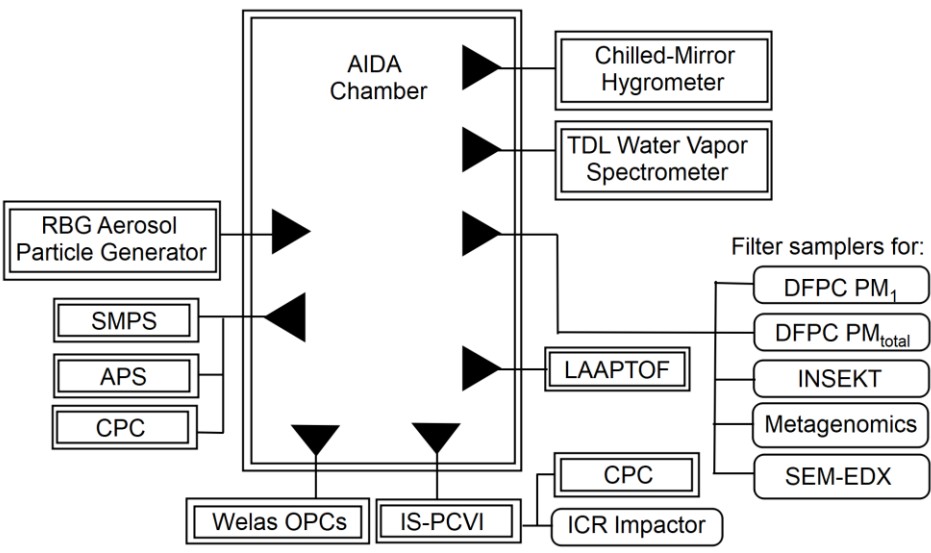

**Figure 2.** Lab experimental schematic of the AIDA facility. All samples were injected using a rotating brush generator (RBG) for aerosol particle generation. Multiple extramural instruments, welas optical particle counters (OPCs), an ice selective pumped counterflow virtual impactor (IS-PCVI), a hygrometer, a tunable diode laser (TDL) spectrometer, a laser ablation aerosol particle time-of-flight mass spectrometer (LAAPTOF; see SI), and aerosol particle counters/sizers (SMPS, APS, CPCs), are connected to the AIDA chamber. Downstream filters and an impactor collected aerosol particles and ice crystal residuals for multiple offline analyses.

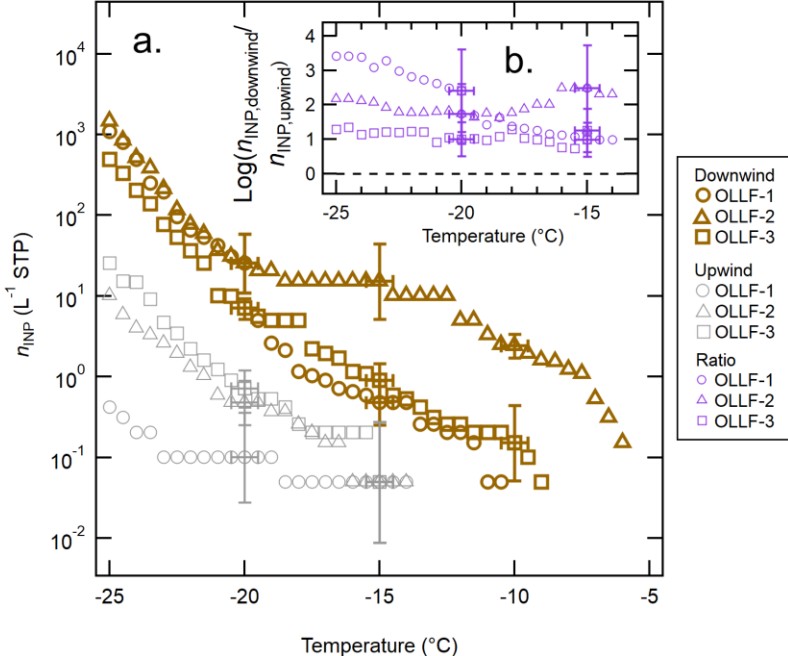

**Figure 3.** The $n_{INP}$ spectra of OLLF aerosol particles from field ambient samples: a comparison of the downwind $n_{INP}$ (brown) to the upwind $n_{INP}$ (grey) from Summer 2017 is shown in (a). Different symbol shapes correspond to individual OLLF sites as indicated in the legend. The uncertainties in $T$ and $n_{sNP}$ are ± 0.5 °C and ± CI95%, respectively. Error bars are shown at selected $T$s to make all data points visible. The log-scaled downwind-to-upwind $n_{INP}$ ratios, $\log(n_{INP,downwind}/n_{INP,upwind})$, for the overlapping $T$ ranges are shown in (b). Note that the uncertainty in this ratio is > 50% due to large CI95% errors for measured $n_{INP}$. The black dashed line represents the ratio of zero (i.e., no difference between $n_{INP,downwind}$ and $n_{INP,upwind}$).

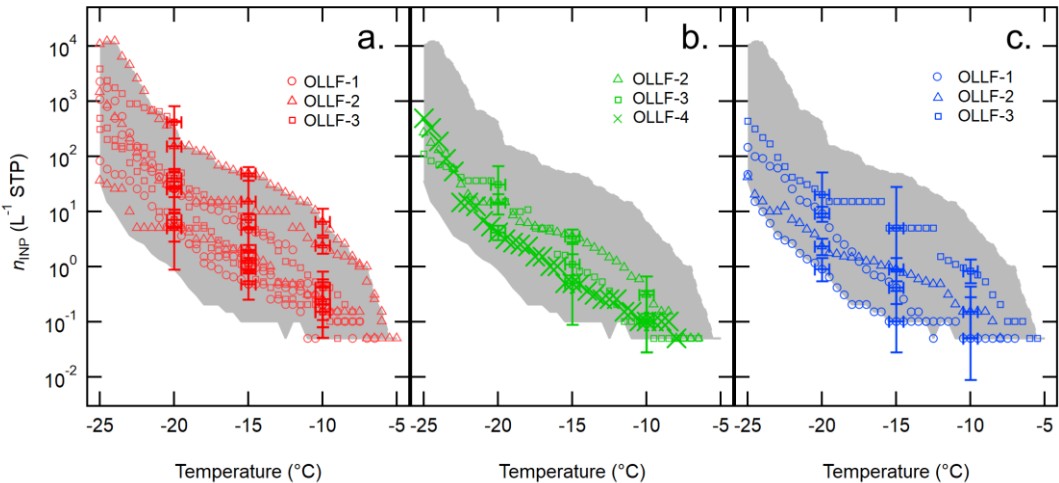

**Figure 4.** Downwind OLLF $n_{INP}$ spectra from 2017 – 2019 sorted based on meteorological seasons are shown; summer (a), spring (b), and winter (c). The uncertainties in $T$ and $n_{s,geo}$ are ± 0.5 °C and ± CI95%, respectively, and error bars are shown at -5, -10, and -15 °C. Shaded area represents minimum – maximum $n_{INP}$.

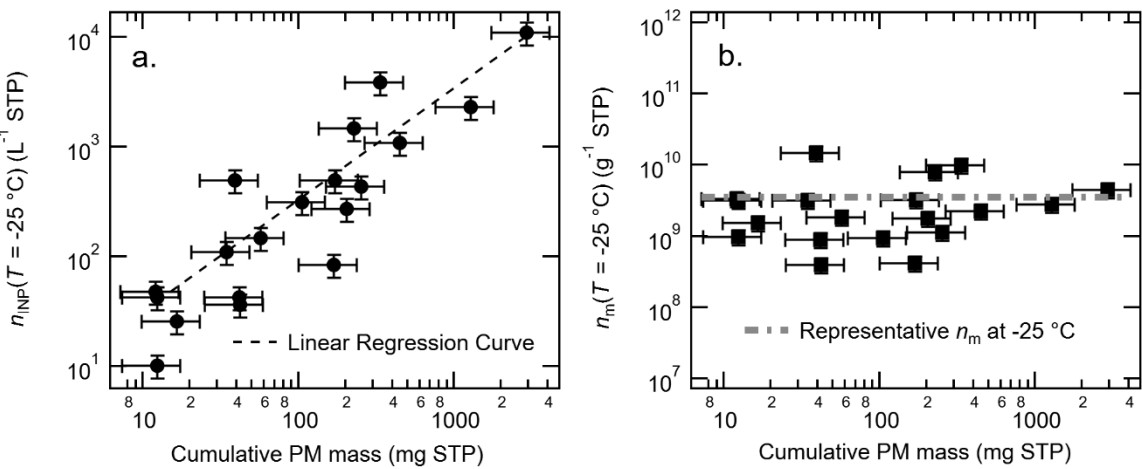

**Figure 5**. Correlation between cumulative PM mass vs. $n_{INP}$ (a) and vs. $n_m$ (b) at -25 °C; a linear regression curve in log scale ($n_{INP}$ = 3.51 × Cumulative PM Mass – 2.41; $r = 0.94$) is shown in (a), and the constant value of representative $n_m$ at the given $T$ (3.55 × $10^9$ g$^{-1}$), which is a median $n_m$ value of minimum – maximum, is shown in (b). Note the errors in cumulative PM mass are ± 40.4% as discussed in **Sect. 3.1**. The uncertainty in $n_{INP}$ and $n_m$ is ± 23.5%.

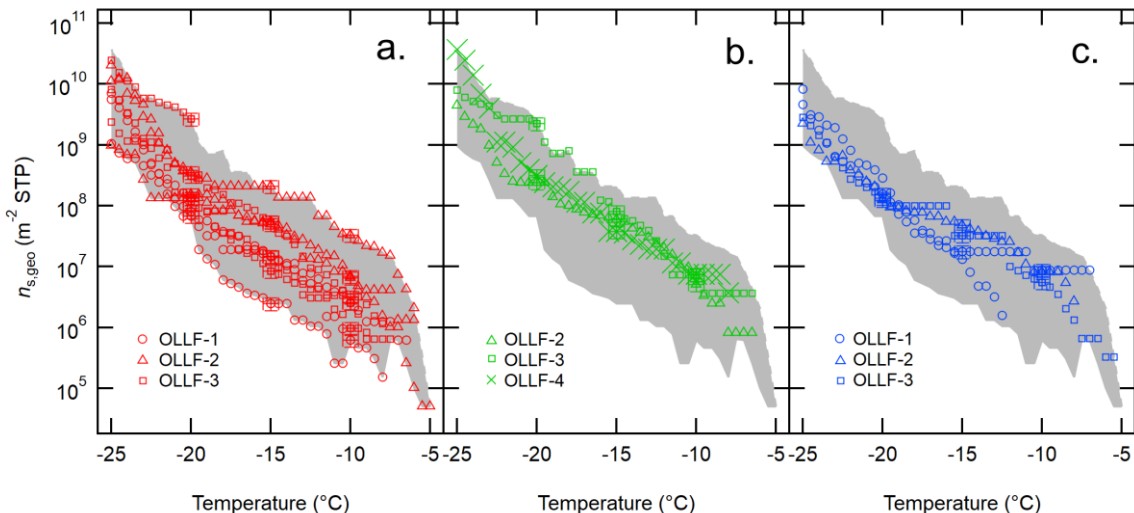

**Figure 6.** The $n_{s,geo}$ spectra of OLLF aerosol particles from field ambient samples collected in 2017 – 2019. All downwind $n_{s,geo}$ spectra from summer (a), spring (b), and winter (c) are shown. Different symbol shapes correspond to individual OLLF sites as indicated in the legend. The uncertainties in $T$ and $n_{s,geo}$ are ± 0.5 °C and ± 23.5%, respectively, and representing error bars are shown at -5, -10, and -15 °C. Shaded area represents minimum – maximum $n_{s,geo}$.

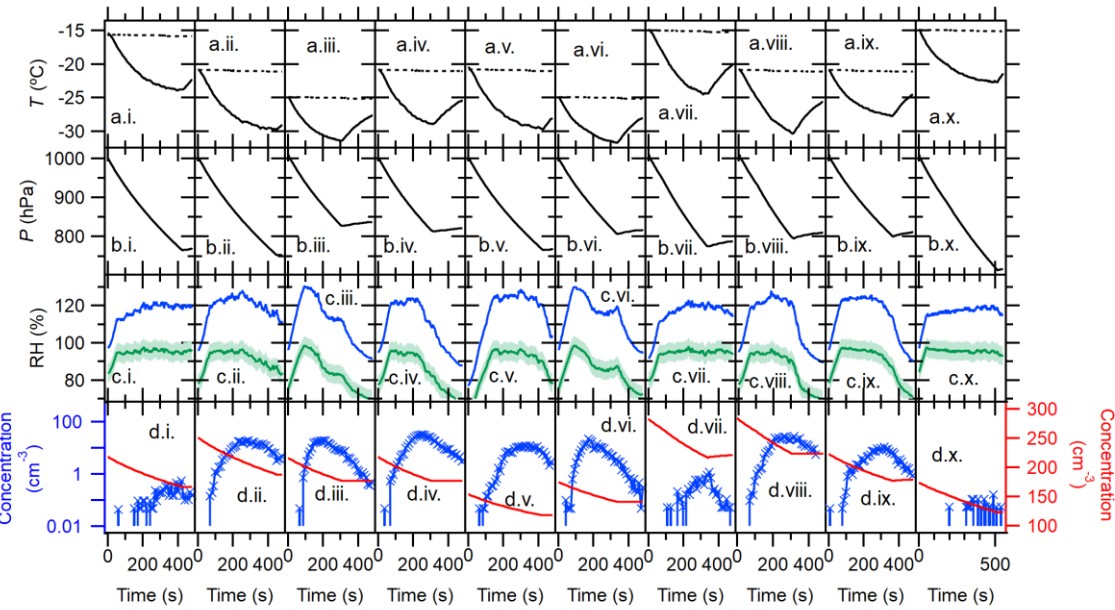

**Figure 7.** Temporal profiles of the AIDA immersion freezing experiment [TXDUST01_07 (i), _08 (ii), _30 (iii), _12 (iv), _13 (v), _32 (vi), _3 (vii), _4 (viii), _16 (ix), _17 (x)]. Arrays of alphabetical panels represent the chamber gas $T$ (solid line) and the chamber wall $T$ (dashed line) (a), P in the AIDA chamber vessel (b), $RH$ with respect to water (green line) and ice (blue line) (c), and aerosol particle concentration initially measured by the CPC (red solid line) as well as number concentration of > 20 μm $D_{ve}$ AIDA particles measured by a welas optical particle counter (blue line) (d). Horizontal numerical panels represent different sample types and AIDA experiments, including TXD01 (i) – (iii), TXD05 (iv) – (vi), TXD01H (vii – viii), and TXD05H (ix – x). $RH$s were determined with an accuracy of ± 5%, represented as green shaded area in (c), using the mean gas $T$ and the mean water vapor concentration.

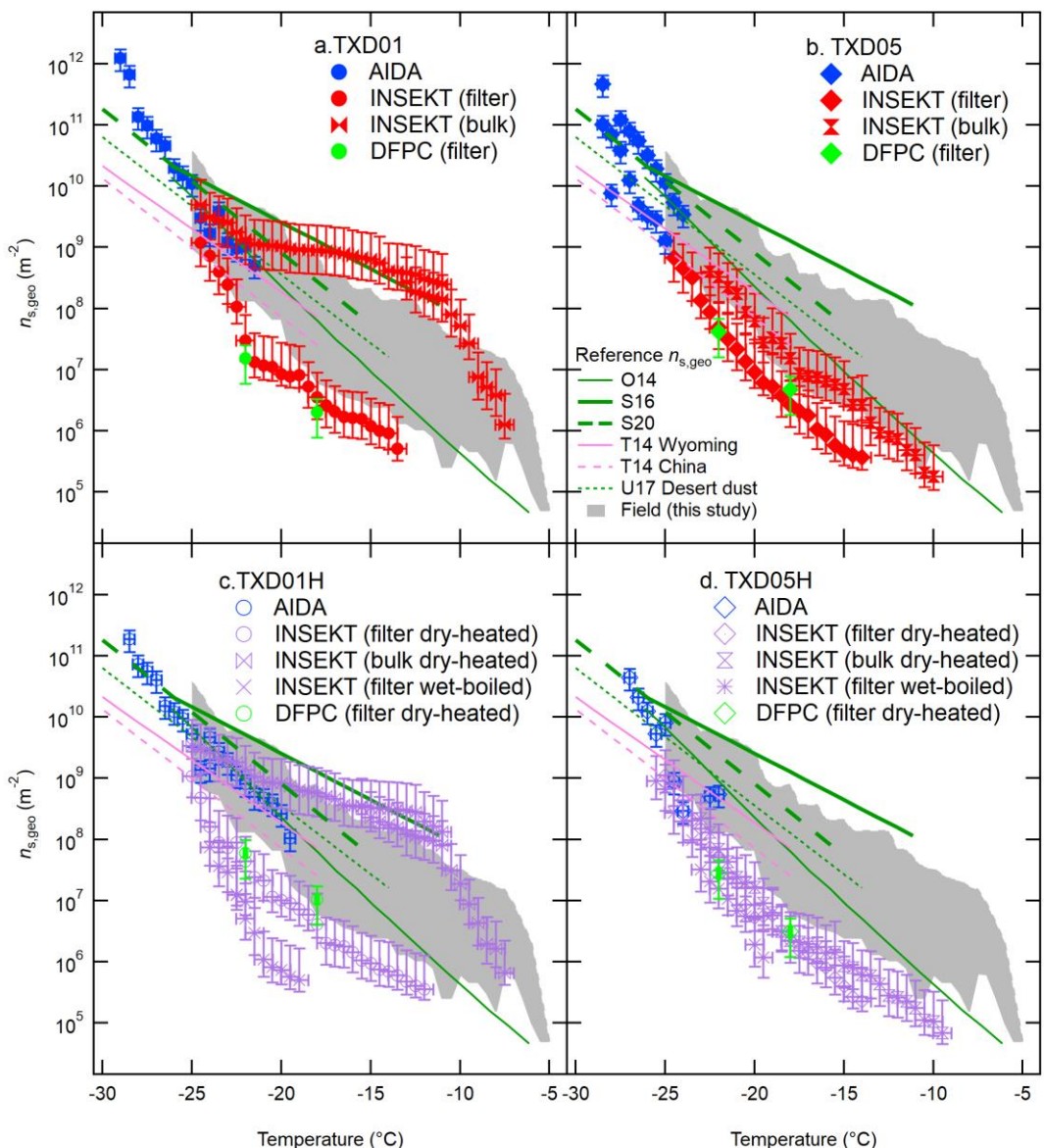

**Figure 8.** IN-active surface-site density, $n_{s,geo}$, of surface materials, TXD01 (a), TXD05 (b), TXD01H (c), and TXD05H (d), was assessed by AIDA, INSEKT, and DFPC (total aerosol particles) as a function of $T$. Six reference $n_{s,geo}$ curves for soil dust s and desert dust are adapted from O'Sullivan et al. (2014; O14), Steinke et al. (2016; S16), Steinke et al. (2020; S20), Ullrich et al. (2017; U17), and Tobo et al. (2014; T14). The grey-shaded area represents the range of our field $n_{s,geo}$ values at 0.5 °C interval for -5 °C > $T$ > -25 °C (**Fig. 6**).

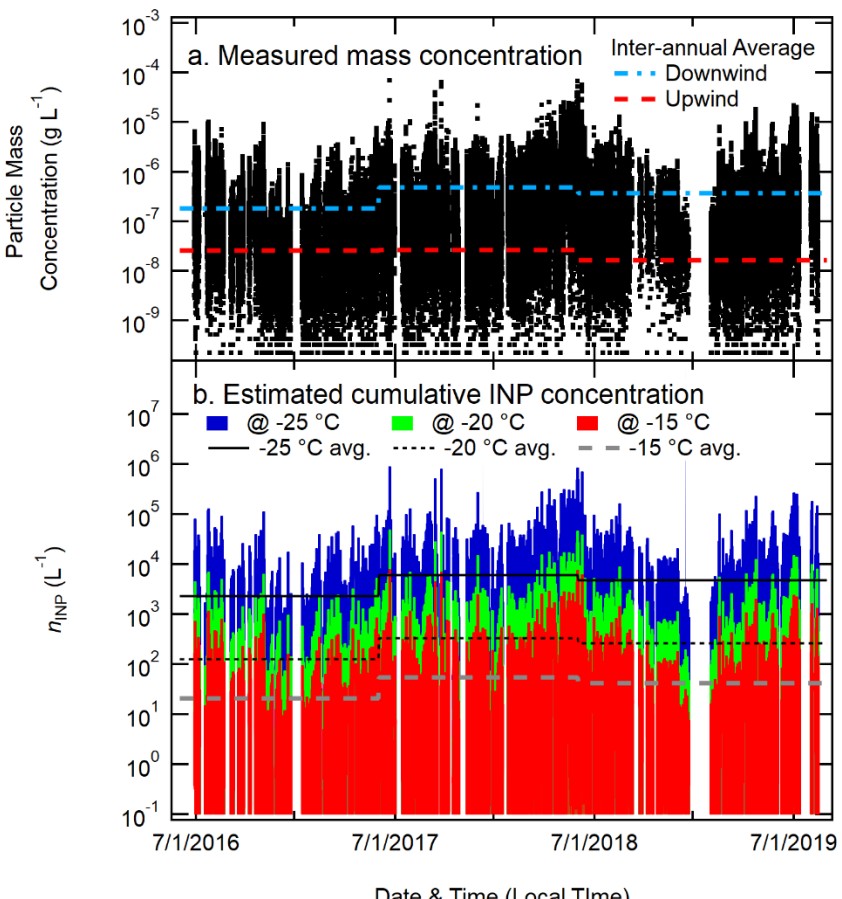

**Figure 9**. OLLF INP concentrations. Time-series plot of TEOM mass concentration measured at the downwind side of OLLF-1 (a) and cumulative $n_{INP}$ estimated at $T$s of -15 °C, -20 °C, and -25 °C (b). In Panel a, inter-annual average mass concentrations of aerosol particles from OLLF (blue dashed line) and upwind (red dashed line) are shown (numbers adapted from **Table 8**). In Panel b, likewise, inter-annual average $n_{INP}$ estimated at -15, -20, and -25 °C (reported in **Table 8**) are also shown. Meteorological summer in Texas is used for the beginning and ending time stamps of each year.

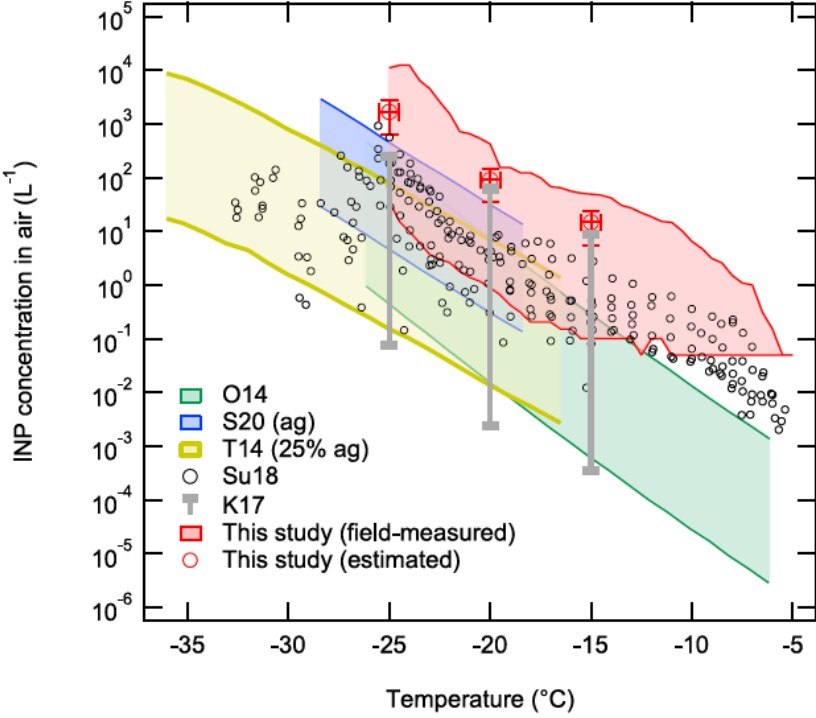

**Figure 10**. Ambient INP concentrations of soil dusts and aerosol particles as a function of *T*. The red-shaded area represents the range of our field $n_{INP}$ values at 0.5 °C interval for -5 °C > *T* > -25 °C from this study (**Fig. 4**). The red open symbols are our
estimated median (± standard deviation) at -15, -20, and -25 °C discussed in **Sect. 3.5**. Five reference data are adapted from O'Sullivan et al. (2014 Fig. 9; O14), Steinke et al. (2020 Fig. 3; S20), Tobo et al. (2014 Fig. 6b; T14), Suski et al. (2018 Fig. 1a-d; Su18), and Kanji et al. (2017 Fig. 1-10; K17). Note that we display the maximum and minimum at -15, -20, and -25 °C of K17 in comparison to our estimation.

**Table 1**. Summary of the ambient aerosol particle filter sampling conditions: UW denotes upwind.

| Year | Date | Location | Start Time (Local) | End Time (Local) | Sample Flow LPM)* | Total volume of sampled air (L STP) | $T$ (°C) | $P$ (mb) | *Relative Humidity* (%) |
|---|---|---|---|---|---|---|---|---|---|
| 2019 | 20190715 | OLLF-1 | 18:45:00 | 22:05:00 | 4.19 | 838.0 | 30.1 ± 3.2 | 1015.6 ± 0.2 | 42.0 ± 10.8 |
| | 20190716 | OLLF-2 | 18:45:00 | 20:29:00 | 4.30 | 447.2 | 34.0 ± 0.7 | 1016.0 ± 0.2 | 27.8 ± 1.7 |
| | 20190724 | OLLF-3 | 19:24:00 | 20:34:00 | 9.08 | 317.8 | 28.9 ± 0.8 | 1020.6 ± 0.1 | 31.6 ± 1.4 |
| | 20190226 | OLLF-1 | 16:08:00 | 19:09:00 | 3.95 | 715.0 | 20.5 ± 2.7 | 1014.8 ± 0.2 | 14.3 ± 2.9 |
| | 20190328 | OLLF-2 | 16:26:00 | 20:52:00 | 5.00 | 1330.0 | 19.4 ± 1.5 | 1012.8 ± 0.2 | 26.5 ± 6.8 |
| | 20190420 | OLLF-3 | 17:05:00 | 21:05:00 | 4.15 | 996.0 | 27.0 ± 2.9 | 1009.0 ± 0.4 | 16.6 ± 5.0 |
| | 20190116 | OLLF-1 | 16:03:00 | 19:33:00 | 3.97 | 832.7 | 16.5 ± 1.9 | 1014.7 ± 0.4 | 30.3 ± 3.1 |
| | 20190117 | OLLF-2 | 15:48:00 | 19:30:00 | 3.97 | 880.2 | 11.0 ± 0.2 | 1016.9 ± 3.5 | 30.2 ± 5.6 |
| | 20190118 | OLLF-3 | 15:40:00 | 18:40:00 | 3.62 | 651.6 | 11.5 ± 3.9 | 1005.3 ± 2.2 | 41.1 ± 21.8 |
| 2018 | 20180722 | OLLF-1 | 18:42:00 | 22:39:00 | 6.58 | 1560.0 | 33.4 ± 4.3 | 1015.7 ± 0.3 | 17.8 ± 5.8 |
| | 20180723 | OLLF-2 | 18:42:00 | 22:17:00 | 5.46 | 1173.8 | 28.4 ± 2.2 | 1022.5 ± 0.7 | 39 ± 5.1 |
| | 20180724 | OLLF-3 | 18:20:00 | 22:13:00 | 3.65 | 850.3 | 28.9 ± 1.4 | 1023.3 ± 0.6 | 38.1 ± 2.6 |
| | 20180416 | OLLF-4 | 16:53:30 | 20:06:40 | 5.99 | 1158.0 | 27.2 ± 1.3 | 1009.8 ± 8.0 | 5.6 ± 0.8 |
| 2017 | 20170709 | OLLF-1 | 19:32:45 | 22:26:00 | 5.28 | 915.6 | 27.9 ± 2.9 | 1017.0 ± 0.4 | 52.8 ± 13.1 |
| | 20170710 | OLLF-2 | 18:06:00 | 22:06:30 | 5.10 | 1227.2 | 30.5 ± 2.5 | 1015.5 ± 0.3 | 30.8 ± 5.1 |
| | 20170711 | OLLF-3 | 18:28:00 | 22:08:00 | 5.13 | 1128.0 | 29.9 ± 2.5 | 1015.2 ± 0.4 | 26.6 ± 6.0 |
| | 20170709 | OLLF-1-UW | 19:50:00 | 22:47:00 | 5.28 | 935.2 | 27.9 ± 2.9 | 1017.0 ± 0.4 | 52.8 ± 13.1 |
| | 20170710 | OLLF-2-UW | 18:28:00 | 22:24:00 | 5.10 | 1204.2 | 30.5 ± 2.5 | 1015.5 ± 0.3 | 30.8 ± 5.1 |
| | 20170711 | OLLF-3-UW | 18:41:45 | 21:54:00 | 5.12 | 983.5 | 29.9 ± 2.5 | 1015.2 ± 0.4 | 26.6 ± 6.0 |

| Year | Date | Location | Start Time (Local) | End Time (Local) | Cumulative PM mass (µg STP)† | $n_{INP}$@ -25°C (L⁻¹ STP) | $n_m$@ -25°C (g⁻¹ STP) |
|---|---|---|---|---|---|---|---|
| 2019 | 20190715 | OLLF-1 | 18:45:00 | 22:05:00 | 168.2 | 8.38E+01 | 4.18E+08 |
| | 20190716 | OLLF-2 | 18:45:00 | 20:29:00 | 41.9 | 3.66E+01 | 3.91E+08 |
| | 20190724 | OLLF-3 | 19:24:00 | 20:34:00 | 105.0 | 3.11E+02 | 9.42E+08 |
| | 20190226 | OLLF-1 | 16:08:00 | 19:09:00 | 57.2 | 1.48E+02 | 1.84E+09 |
| | 20190328 | OLLF-2 | 16:26:00 | 20:52:00 | 204.5 | 2.72E+02 | 1.77E+09 |
| | 20190420 | OLLF-3 | 17:05:00 | 21:05:00 | 34.5 | 1.10E+02 | 3.18E+09 |
| | 20190116 | OLLF-1 | 16:03:00 | 19:33:00 | 12.0 | 4.78E+01 | 3.31E+09 |
| | 20190117 | OLLF-2 | 15:48:00 | 19:30:00 | 41.5 | 4.22E+01 | 8.94E+08 |
| | 20190118 | OLLF-3 | 15:40:00 | 18:40:00 | 251.8 | 4.35E+02 | 1.13E+09 |
| 2018 | 20180722 | OLLF-1 | 18:42:00 | 22:39:00 | 1281.0 | 2.31E+03 | 2.81E+09 |
| | 20180723 | OLLF-2 | 18:42:00 | 22:17:00 | 2917.9 | 1.10E+04 | 4.43E+09 |
| | 20180724 | OLLF-3 | 18:20:00 | 22:13:00 | 334.1 | 3.87E+03 | 9.84E+09 |
| | 20180416 | OLLF-4 | 4:53:30 | 8:06:40 | 38.9 | 4.93E+02 | 1.47E+10 |
| 2017 | 20170709 | OLLF-1 | 19:32:45 | 22:26:00 | 445.3 | 1.09E+03 | 2.25E+09 |
| | 20170710 | OLLF-2 | 18:06:00 | 22:06:30 | 226.5 | 1.48E+03 | 8.00E+09 |
| | 20170711 | OLLF-3 | 18:28:00 | 22:08:00 | 171.5 | 4.92E+02 | 3.23E+09 |
| | 20170709 | OLLF-1-UW | 19:50:00 | 22:47:00 | 12.4 | 4.22E+01 | 3.18E+09 |
| | 20170710 | OLLF-2-UW | 18:28:00 | 22:24:00 | 12.4 | 1.01E+01 | 9.78E+08 |
| | 20170711 | OLLF-3-UW | 18:41:45 | 21:54:00 | 16.5 | 2.57E+01 | 1.53E+09 |

*A mass flow controller or a critical orifice was used to ensure a constant flow throughout each sampling activity. An air flow rate was measured with a flowmeter (TSI Inc., Model 4140). †Cumulative values of mass collected on a filter were estimated by integrating DustTrak mass data, sampling time, and flow rate.

**Table 2**. Properties of OLLF samples: non-heated (TXD01 & TXD05) and dry-heated (TXD01H & TXD05H).

| System | TXD01 | TXD05 | TXD01H | TXD05H |
|---|---|---|---|---|
| [1]Density, g cm⁻³ | 1.89 ± 0.06 | 2.05 ± 0.06 | 1.94 ± 0.06 | 2.00 ± 0.06 |
| Geometric *SSA*, m² g⁻¹ | 4.95 ± 0.82 | 3.97 ± 0.02 | 5.62 ± 0.16 | 4.04 ± 0.11 |
| [2]BET-based *SSA*, m² g⁻¹ | 3.23 ± 0.20 | 2.41 ± 0.20 | 3.23 ± 0.32 | 2.41 ± 0.24 |

[1]With a measurement standard deviation of ± 0.06, our system is capable of measuring densities of other powder samples, such as illite NX (2.91 g cm⁻³) and fibrous cellulose (1.62 g cm⁻³). Note that these values are similar to the density values reported by manufacturers for illite NX (2.65 g cm⁻³) and fibrous cellulose (1.5 g cm⁻³). [2]Brunauer et al., 1938.

**Table 3**. Characterization of particle properties: assessed prior to AIDA expansion experiments (H denotes dry-heated).

| | | | Aerosol Particle Measurements | | | |
|---|---|---|---|---|---|---|
| **Experiment ID** | Aerosol Particle Type | Mode (Min–Max) Diameter, μm** | $N_{total,0}$, $\times 10^3$ L$^{-1}$ | $S_{total,0}$, $\times 10^{-9}$ m$^2$ L$^{-1}$ | $M_{total,0}$, $\times 10^{-9}$ g L$^{-1}$ | Geometric *SSA*, m$^2$ g$^{-1}$ |
| TXDUST01_7 | TXD01 | 0.55 (0.10–3.16) | 213.7 | 98.8 | 18.4 | 5.38 |
| TXDUST01_8* | TXD01 | 0.54 (0.11–2.69) | 266.3 | 115.5 | 21.1 | 5.46 |
| TXDUST01_30 | TXD01 | 0.72 (0.08–6.44) | 210.6 | 119.0 | 29.7 | 4.01 |
| TXDUST01_12* | TXD05 | 0.67 (0.09–5.14) | 199.2 | 163.5 | 41.1 | 3.98 |
| TXDUST01_13 | TXD05 | 0.71 (0.10–4.71) | 155.0 | 117.2 | 29.6 | 3.95 |
| TXDUST01_32 | TXD05 | 0.84 (0.15–4.37) | 163.3 | 124.9 | 33.2 | 3.77 |
| TXDUST01_3* | TXD01H | 0.53 (0.10–2.69) | 301.1 | 130.5 | 23.7 | 5.51 |
| TXDUST01_4 | TXD01H | 0.52 (0.08–3.05) | 282.1 | 137.1 | 23.9 | 5.73 |
| TXDUST01_16* | TXD05H | 0.78 (0.12–4.95) | 227.4 | 195.1 | 49.3 | 3.96 |
| TXDUST01_17 | TXD05H | 0.74 (0.12–4.59) | 185.7 | 119.7 | 29.1 | 4.12 |

*INSEKT and DFPC samples were collected. **Based on the d$S$/dlog$D_{ve}$ fit; Min–Max values are estimated at 0.1 $\times$ 10$^{-9}$ m$^2$ L$^{-1}$; $N_{total,0}$ = total number concentration of particles at the initial stage (t = 0) prior to expansion; $S_{total,0}$ = total surface concentration of particles at the initial stage (t = 0) prior to expansion; $M_{total,0}$ = total mass concentration of particles at the initial stage (t = 0) prior to expansion; $D_{ve}$ = volume equivalent diameter.

**Table 4**. DFPC-estimated $n_{INP}$ for TXD01 and TXD05 samples: H denotes the dry-heated sample. The subscripts of Tot and PM$_1$ represent INP obtained from total aerosol particles and that from PM$_1$ size-segregated aerosol particles, respectively. Standard deviations were derived based on multiple measurements for each sample. Only PM$_{10}$ of TXD01 sample was examined due to the data limitation. This size limit is valid since we observed only < 6.44 μm aerosol particles in AIDA (**Table 3**). Supermicron INP fraction (%) is calculated by [() / $n_{INP,total}$] $\times$ 100. Note the change of supermicron INP fraction at two temperatures before and after the dry-heated treatment is due to different total aerosol particles available in AIDA (**Table 3**). A comparison of $n_{s,geo}$ shows a
reasonable agreement within the uncertainties reported in **Sect. 2.4**.

| Dust | $n_{INP}$ x10$^3$ (L$^{-1}$) ± standard dev. | | Supermicron INP fraction (%) | |
|---|---|---|---|---|
| | -18 °C | -22 °C | -18 °C | -22 °C |
| TXD01$_{Tot}$ | 340.0 ± 211.0 | 2580.0 ± 698.0 | 26.5 | 46.5 |
| TSD01$_{PM1}$ | 250.0 ± 90.0 | 1380.0 ± 219.0 | | |
| TXD01H$_{Tot}$ | 1266.7 ± 192.5 | 7141.7 ± 885.0 | 72.4 | 60.2 |
| TSD01H$_{PM1}$ | 350.0 ± 120.0 | 2841.7 ± 375.8 | | |
| TXD05$_{Tot}$ | 770.0 ± 110.0 | 6780.0 ± 426.0 | 58.4 | 48.4 |
| TSD05$_{PM1}$ | 320.0 ± 116.0 | 3500.0 ± 1066.0 | | |
| TXD05H$_{Tot}$ | 508.3 ± 100.0 | 4575.0 ± 1080.8 | 60.7 | 24.6 |
| TSD05H$_{PM1}$ | 200.0 ± 45.8 | 3450.0 ± 715.8 | | |

**Table 5**. Abundance of major bacterial phyla in dust samples TXD01 and TXD05. Numbers indicate percentage of the OTUs for each phylum in the total bacterial microbiome. The percentage of *Actinobacteria* in the microbiome is increased in aerosolized samples.

| Taxonomy | Bulk TXD01 | Bulk TXD01 (Dry-Heated) | Aerosolized TXD01 | Bulk TXD05 | Bulk TXD05 (Dry-Heated) | Aerosolized TXD05 |
|---|---|---|---|---|---|---|
| *Actinobacteria* | 40.1% | 42.2% | 60.1% | 51.7% | 67.1% | 92.9% |
| *Chloroflexi* | 4.4% | 4.9% | 4.0% | 9.3% | 4.0% | 1.1% |
| Unclassified | 3.8% | 4.0% | 4.0% | 8.1% | 9.9% | 2.6% |
| *Proteobacteria* | 19.9% | 16.8% | 11.4% | 13.0% | 10.3% | 0.6% |
| *Firmicutes* | 17.2% | 17.5% | 13.6% | 15.4% | 7.3% | 2.8% |
| *Bacteroidetes* | 12.6% | 13.1% | 6.5% | 2.3% | 1.4% | 0.0% |
| *Gemmatimonadetes* | 1.6% | 1.3% | 0.4% | 0.1% | 0.0% | 0.0% |
| *Cyanobacteria* | 0.2% | 0.1% | 0.0% | 0.0% | 0.0% | 0.0% |
| *Fibrobacteres* | 0.1% | 0.1% | 0.0% | 0.0% | 0.0% | 0.0% |
| *Nitrospinae* | 0.1% | 0.0% | 0.0% | 0.0% | 0.0% | 0.0% |
| *Planctomycetes* | 0.0% | 0.0% | 0.0% | 0.0% | 0.0% | 0.0% |
| *Rhodothermaeota* | 0.0% | 0.0% | 0.0% | 0.0% | 0.0% | 0.0% |
| *Spirochaetes* | 0.0% | 0.0% | 0.0% | 0.1% | 0.0% | 0.0% |

**Table 6**. Summary of particle size properties through electron microscopy.

| Sample Type | Measured Particles | *Diameter (µm) | | **Aspect Ratio | | Spermicron Size Fraction (%) |
|---|---|---|---|---|---|---|
| | | Average | Std. Error | Average | Std. Error | |
| TXD01 aerosol | 159 | 0.80 | 0.03 | 1.46 | 0.04 | 27.7% |
| TXD01 residual | 185 | 0.87 | 0.03 | 1.56↑ | 0.04 | 29.2% |
| TXD01H dry-heated aerosol | 162 | 0.82 | 0.03 | 1.42 | 0.03 | 26.5% |
| TXD01H dry-heated residual | 126 | 0.90 | 0.04 | 1.48↑ | 0.05 | 33.3% |
| TXD01 cumulative | 632 | **0.84** | 0.02 | 1.48 | 0.02 | 29.2% |
| TXD05 aerosol | 194 | 0.99 | 0.03 | 1.37 | 0.03 | 44.3% |
| TXD05 residual | 164 | 1.17 | 0.03 | 1.49↑ | 0.03 | 56.7% |
| TXD05H dry-heated aerosol | 100 | 1.23 | 0.04 | 1.41 | 0.05 | 64.0% |
| TXD05H dry-heated residual | 169 | 0.90 | 0.03 | 1.49↑ | 0.04 | 27.8% |
| TXD05 cumulative | 627 | **1.05** | 0.02 | 1.44 | 0.02 | 38.8% |

*Average of 2-D cross sections. **Ratio of cross sections (i.e., longer cross section/shorter cross section).

**Table 7**. Summary of particle composition types through energy dispersive X-ray spectroscopy.

| Particle Type | TXD01 Abundance (%) | | | | TXD05 Abundance (%) | | | |
|---|---|---|---|---|---|---|---|---|
| | Aerosol | Residual | Dry-heated Aerosol | Dry-heated Residual | Aerosol | Residual | Dry-heated Aerosol | Dry-heated Residual |
| Organic | 5.0 | 7.6↑ | 3.1 | 9.5↑ | 8.2 | 9.1↑ | 3.0 | 11.2↑ |
| Salt-rich | 34.6 | 10.3↓ | 35.8 | 4.0↓ | 22.2 | 4.9↓ | 15.0 | 10.1↓ |
| Mineral-rich | 57.2 | 77.8 | 56.2 | 70.6 | 68.0 | 82.9 | 79.0 | 74.6 |
| Other | 3.1 | 4.3 | 4.9 | 15.9 | 1.5 | 3.0 | 3.0 | 4.1 |

**Table 8.** Inter-annual and seasonal $PM_{10}$ mass concentrations from OLLF-1 as well as estimated $n_{INP}$.

| | $PM_{10}$ Mass Concentration (g L$^{-1}$) | | Estimated $n_{INP}(T)$ (L$^{-1}$) | | |
|---|---|---|---|---|---|
| | *OLLF | Upwind | $T$ = -15 °C | $T$ = -20 °C | $T$ = -25 °C |
| **2016 – 2017** | **1.8E-07** | **2.6E-08** | **20.7** | **127.5** | **2323.4** |
| Summer | 3.7E-07 | 5.2E-08 | 42.3 | 260.5 | 4747.7 |
| Fall | 1.6E-07 | 2.8E-08 | 18.1 | 111.7 | 2036.3 |
| Winter | 6.3E-08 | 1.5E-08 | 7.2 | 44.2 | 806.2 |
| Spring | 1.6E-07 | 2.1E-08 | 17.7 | 108.9 | 1985.5 |
| **2017 – 2018** | **4.8E-07** | **2.6E-08** | **54.6** | **336.4** | **6133.0** |
| Summer | 3.0E-07 | 2.3E-08 | 33.8 | 208.5 | 3801.1 |
| Fall | 3.1E-07 | 1.9E-08 | 35.4 | 218.2 | 3978.3 |
| Winter | 2.5E-07 | 1.3E-08 | 27.9 | 171.7 | 3129.6 |
| Spring | 9.2E-07 | 4.6E-08 | 104.1 | 641.3 | 11690.9 |
| **2018 – 2019** | **3.7E-07** | **1.7E-08** | **42.3** | **260.7** | **4752.5** |
| Summer | 4.9E-07 | 2.6E-08 | 55.6 | 342.3 | 6240.6 |
| Fall | 2.4E-07 | 7.9E-09 | 26.8 | 165.3 | 3013.0 |
| Winter | 1.5E-07 | 1.3E-08 | 17.0 | 104.8 | 1910.2 |
| Spring | 2.8E-07 | 1.6E-08 | 31.8 | 195.8 | 3570.0 |

*Upwind concentration is subtracted.