# Peer review of "Laboratory and field studies of ice-nucleating particles from open-lot livestock facilities in Texas"

_Atmospheric Chemistry and Physics, 2020_

## Referee Comment (RC1) · Anonymous Referee #1 · 24 Jan 2021

**Review of "Feedlot is a unique and constant source of atmospheric ice-nucleating particles" by Hiranuma et al.**

This study reports the immersion ice nucleation ability of ambient feedlot aerosol samples and surface materials. Several instruments were engaged to perform physical, chemical and biological characterization, as well as immersion freezing ability determination of these filter samples and proxies. The authors have done great job characterizing and the presented data is abundant. However, the writing quality is pretty poor and the authors did not seem to offer a consistent usage of terms and symbols throughout the paper, which in part is hard to follow. More importantly, there are a few aspects need to be sufficiently addressed before this paper can be reconsidered for acceptance: First, the writing quality does require substantial improvement. Second, the paper structure is imbalance, with about 50% pages on materials and methods yet I still feel difficult to get the logic connection and setup of these experiments. Third, the Results and Discussion part is basically an experiment report with little analysis and discussion.

**Major comments:**

1. Introduction: The logic of this part needs to be thoroughly improved. A much more detailed review of relevant previous work should be added. The following points should be explicitly addressed to give the audience an idea about the importance of this work: Why is feedlot dust relevant to mixed-phase clouds? How representative (e.g. size distribution, chemical composition, etc.) are the surface samples collected in this study compared to actual ambient aerosols? How representative are the sampling sites for the whole Texas? The title made a very broad and general statement without any solid support.

2. Materials and methods: The authors used 4 out of 9 pages to describe the materials and instruments used in this study. But it remains difficult to structure the experiments conducted. Please streamline this section and provide necessary experiment information.

3. As the main body of the manuscript, the results and discussion section is divided into 7 separate subsections without logical connection. Most of these subsection, simply report values and lack further analysis and discussion of the results. For instance, it is claimed that super-micron INP fraction contributed 49.7% ± 6.0% of total INP for TXD samples at -18°C and -22°C (P7L20-22). However, no analysis or interpretation of this conclusion is presented. In Table 4 (P23), the change of super-micron INP fraction at two temperatures before and after the dry-heated treatment is different, and what could be the reason for this deviation? Please revise this part in depth and add comparisons of the results obtained in this study to previous relevant work.

**Specific comments:**

- P1L18: Please capitalize each word.

- P1L21: Please add "the" before U.S. and check the usage of "the" throughout.

- P1L28: "thereby finding…" is not clear to me.

- P1L30: Please replace "improved" with "improving".

- P1L32: Please rephrase to "…explore the relationship between INP and supercooled cloud properties…".

- P1L38: 25% of total what?

- P1L46: Please rephrase.

- P1L48: "suggesting their regional scale impact" is not clear to me.

- P1L50: Please verify the usage of "impact in".

- P1L51: Please spell out "IN", as this is the first appearance of this term in the text. Please check the whole paper for first appearance of abbreviations.

- P1L54: Please add the classical textbook of Pruppacher and Klett (2010). Please also spell out "INP".

- P1L56: Please rephrase "focused in this study".

- P2L14: Please spell out "Ts" or specify the meaning of the subscript.

- P2L15: Please spell out "ICR".

- P2L39: Please add "as" before addressed.

- P2L43: Please spell out "AIDA".

- P2L47: Please spell out "TXD".

- P2L58: What instrument did the author use to measure BET SSA?

- P3L1: Does the author imply that TXD05 is soil dust while TXD01 contains K-feldspar here? If so, why is the density of TXD05 larger than that of TXD01?

- P3L3-8: Why did the author compare geometric and BET SSA directly and attribute the difference between these two totally different methods to sample particle size variance?

- P3L15: Please see the comment above.

- P3L17: What is the purpose of filter sampling? Can the authors provide a scheme of the experimental setup in this study?

- P3L24: Please add reference after "IS-PCVI".

- P3L28: Should be μm here.

- P3L31: Does the authors mean "individual ice crystal" here? If so, maybe it's better to delete "of individual particles".

- P3L41: Please add more AIDA references apart from the review paper.

- P3L42-45: According to Table 3, the super-micron particle size can reach 5-6 μm. What's the size detection limit of TSI SMPS and aerosol particle sizer?

- P3L45: What technique did the authors use to aerosolize particles?

- P4L2: Did the authors intend to say "array"?

- P4L7: Please spell out "HPLC".

- P4L8-9: How was the detection limit determined?

- P4L11-12: Can the authors provide an example for such processing in SI?

- P4L25-26: If it's "a series of diluted suspensions", should it be "×15 **to** ×225" here? Besides,

please replace "x" with "×" throughout the paper.

- P4L38: Remove the "of" after "100 L".

- P4L40: "was characterized".

- P4L42: Please specify what type of "diameter".

- Sect. 2.6: Please simplify the description of previous work.

- P5L31: What is the typical size range of droplets and ice crystals in this study?

- P5L36: Should be μm. Please check the units throughout.

- P5L37: During TXDUST01 what?

- P5L45: Please change the parenthesis to "(…Model 1400a; Patashnick…)".

- P5L58: Please enclose "<100 μg m$^{-3}$" into parenthesis.

- P6L3-9: What's the point of repeating previous results in this paper?

- P6L11: I suggest to place the symbol right after the corresponding definition. Do the author mean INP concentration per unit particle mass and INP concentration per unit particle surface? Please specify and avoid misunderstanding.

- P6L22: How were these quantities derived? Can the authors please provide the derivation or conversion process in SI apart from the references here?

- P6L23: Please verify the usage of "Regardless".

- P6L25: How did the author decide whether the data was "uncertain systematically erroneous data"?

- P6L27-28: The difference between field samples and lab samples in Fig. 3(a) can reach up to three orders of magnitude between -20 to -10 °C, which is not "comparable" and is not capable of "validating" from my point of view.

- P6L31: Is this linear relationship a linear correlation analysis or a linear regression fit? Please specify, and give the corresponding correlation coefficient and/or other parameters to indicate the goodness of this correlation.

- P6L31-32: Again, how was the "INP scaled to mass" conversion achieved?

- P6L33-34: The authors claimed that ambient meteorological conditions might not be determining factors for INP concentrations. What about the difference between different sampling sites? Is there any previous study drew the same conclusion?

- P6L34: Please add "as" before "summarized".

- P6L38: "several hundred INPs L$^{-1}$" seems very high. Is there any previous report of such superior IN activity of ambient dust? Besides, please specify the "notable correlation" with solid particle size distribution and IN activity data.

- P6L49: What does "they" refer to? Who/what agree with who/what?

- P6L52-53: The conclusion drawn here is vague and speculative. Why does different sample source affect the IN ability?

- P7L1-11: This paragraph should move to Sect. 2.

- P7L14-15: The difference between heated and unheated samples measured by INSEKT and DFPC exhibited different trend instead of "not apparent" as the authors stated, please explain.

- P7L12-16: The comparison of the instruments is irrelevant to the purpose of this section and should go to SI. In addition, why are the results of WT-CRAFT not added to compare the three instruments simultaneously?

- P7L28-30: Again, the result and interpretation seem very speculative. Is there any physical/chemical/biological evidence to support the statement here?

- P7L32-33: How representative are the samples tested in this study to "natural supermicron-dominant INPs"? Again, very broad and general statement without support.

- P7L33-35: Did the authors compare the immersion parameterization proposed in this work with other widely used parameterization models? How consistent and different they are? Whether the new immersion parameterization is universal? Also: Fitting an immersion parameterization solely as a function of a single parameter, temperature, might be misleading. The particle number and/or mass concentrations vary considerably around the world. Do the authors believe that the dataset of this parameterization is representative of the rest of the world?

- P7L44-45: Previous studies on bio-aerosol INP activity reported the importance OF ice active protein instead of DNA. Why didn't the authors conduct protein analysis?

- P7L59: The data is not consistent with Table 7.

- P7L6-9: Why do the authors refer to previous study instead of the single particle chemical analysis in this study to infer particle hygroscopicity?

- P7L17-18: Data presented in Table 8 does not support this statement. Why did the authors decide to omit the influence of carbonaceous content on particle IN activity?

- P7L38: Why did the authors use different units for PM mass loading here?

- P7L47: What is the significance of average estimated INP concentrations between different years?

- P7L490-50: Again, is there any reason for such high INP activity? Is there any previous paper on such efficient INP?

- P7L52-53: Reads like introduction.

- P8L4-35: Please rephrase and improve the writing quality.

- Please replace "lpm" with "LPM" in the paper and check unit usage throughout.

- Please check the space between the number and °C for consistency.

Fig. 1:

Panel (c):

● What does the green shading in panel (c) stand for?

● Why did the RH drop below water saturation in column (ii-iv), (vi), (viii) and (ix) during **immersion freezing** studies?

Panel (d):

- The total particle concentration line is hard to notice.

This figure add little to the paper, should it go to SI?

Fig. 2:

- The font size is too small and unclear.

- P14L13-15: Please rephrase.

- P14L13: What is "CF-to-IF" ratio?

This figure add little to the paper, should it go to SI?

Fig. 3:

- The font size is too small and unclear.

- Labels of panels a and b are missing.

- What does $n_s$ in the figure refer to? The authors mentioned $n_{s,geo}$ in P6L11, is there any relevance between the text and this figure? Why is the caption inconsistent with the figure? Is there any difference among ns,geo (STP), ns,geo, ns (STP), and (STP)?

Fig. 4:

- Please make it explicit that this figure is for -25°C.

- What caused such large error of cumulative PM mass?

- "The uncertainty in nINP and nm is ± 23.5%". However, the error bars of $n_{INP}$ in Fig. 4(a) seems not equal to this value. Also: the correlation coefficient and/or other parameters should be given in Fig. 4.

Fig. 5:

- The color of error bars is the same with the color of heated samples. Please make this figure clearer for readers.

- Why did the authors use different scales in the same figure?

- The meaning of $n_{s,geo}$ is different from the definition in P6L11. Please clarify if these are different quantities, and explain why the terms and symbols are so confusing in the paper.

- Please check the error bars in panel c. How come that the upper error bars are longer than the lower bars in a log-scale plot?

- Please check the labels of the figures for consistency.

- In Fig. 5 (a.ii), there is a significant difference between the filter and bulk samples at temperatures above -20 °C, but no such difference in Figure 5 (b.ii). What causes this difference?

Fig. 6:

- Is it worth wasting a figure and report just one set of DFPC data rather than include the data in Fig. 5?

Table 2:

- Why did the author report "relative standard deviation of ±3%" instead of one standard

deviation from the mean value?

- How does heating affect ATD sample density? Why did heating lead to increased density for TXD1? Why didn't the author measure the density of a specific sample before and after heating?

Table 3:

- What does the number in Experiment ID mean? It's very hard to understand together with Fig. 1.

- Why is the size distribution so wide? What does the size distribution look like?

- Why didn't the author report particle number size distribution? What quantities do the letters refer to?

Table 4:

- Why do the authors put a table here with little description and discussion in text?

Table 7:

- Why were the residual projected sizes consistently larger than those of aerosols except for TXD05H? What do the authors think happened here?

Reference

Pruppacher, H. R., and Klett, J. D.: Microphysics of Clouds and Precipitation, 2 ed., Atmospheric and Oceanographic Sciences Library, 18, Springer Netherlands, XXII, 954 pp., 2010.

---

## Referee Comment (RC2) · Anonymous Referee #2 · 25 Jan 2021

This paper is focused on the potential of domesticated animal feeding facilities as sources of atmospheric ice nucleating particles (INPs). Assessing anthropogenic sources of INPs is an important area of research. Hence, this is an area of research suitable for publication in ACP.

However, there are some significant problems with this paper. Most importantly, the paper is not well written. These authors, including the first author, have produced some excellent pieces of work in the past so I know they are capable of much better. I do not wish to spend a great deal of time going from line to line trying to edit the manuscript for them. Instead I will focus on several key areas, which I'll work through here:

1. 'Feedlot': it may be obvious what this is to a farmer in the USA, but it is not obvious

what this is to the wider community. I had to google the term to find out. An alternative title could be 'Cattle feeding facilities in the USA are a sources of ....'

2. The abstract needs to be rewritten. Tell the reader about the conclusions of your work, not the topics you cover without an indication of what the key results and conclusions are. E.g.: 'New data on the ice nucleation (IN) properties of agricultural dust at heterogeneous freezing temperatures (Ts > -29°C) were generated, providing statistical context.'; 'Overall, we successfully characterized physical, chemical, and biological properties of aerosol particles found at a cattle feedlot'; 'The relationship between these measured properties and atmospheric IN parameterization relevant to mixed-phase clouds is discussed.'; 'Our INP parameterization and ICR characterization are meaningful for improved understanding of INP emission and cloud microphysical processes in the supermicron-particle laden region'.

3. Intro: these paragraphs are far too long and confusing. Break up into topics and build a logical case for this study.

4. P2 ln 45. 'Milling and grinding'. It is inappropriate to mill natural samples of material where the larger sizes are likely made of different materials to the finer aerosolisable fraction. If, for example, the largest grains are ice active and they are milled down to sizes of atmospheric relevance then the ice nucleating ability you measure is likely to be simply a product of the milling process. Milled natural dusts like these samples are therefore not a meaningful proxy for the dust that may be aerosolised from this source, unless there is some justification for treating the sample in this way.

5. Ln 46: Dry heat tests: what precedent is there for 100 C being a suitable test for deactivation of INP proteins? Ideally we would be shown control experiments with a biological ice nucleator.

6. Ln 48: Wet heat tests: Clarify the procedure here. The normal practice is to place a sealed vial in a volume of boiling water. The way the text reads is that the sample itself was boiled for 20 minutes. If this was the case, then how was the loss of water from

the sample accounted for?

7. P7. Section 3.3/ The first two paragraphs have little to do with the heading of this section.

8. Section 3.4 and Table 5. Why is there a parameterisation for each sample? This seems excessive. It would be more useful to have a single parameterisation with an indication of variability.

9. P7, ln 34. 'this parameterization can be easily incorporated in many model platforms'. The authors need to be more specific how they envisage that this might happen. Do relevant models have this source of dust in them already with the emission already set up as an independent tracer? I suspect the answer is no. So, what else would we need to know in order to be able to represent this source of INP?

10. Ln 46. ' no notable difference after dry-heating was observed for both TXD01 and TXD05, representing an important negative result'. Why is this important? Is it significant that there is no effect of heating the sample dry to 100 C. It would only be significant if IN proteins are known to deactivate on heating to this temperature.

11. P 8 section 3.7: State which ns parameterisation was used in this calculation

12. P 8. Section 3.7: Equation 1 is only appropriate if a small fraction of INP at any one size is activated to ice. This equation does not take into account the number of dust particles. For example, if there were 10 dust particles per cm3, this parameterisation might predict 1000 dust cm-3, which would clearly be nonsense. To predict INP concentrations using an ns (or similar) parameterisation, you either need to be able to prove that only a small fraction of particles at one size over the whole size range activate to ice or integrate over the size distribution.

13. P8. Section 3.7: Why use ns. The data seems to spread out in ns substantially, but in nm, they collapse. Hence, nm seems to be a better way of doing this calculation.

14. Ln 50. 'which is three orders of magnitude higher than typical ambient INP concentration from continental sources'. This is a selective reading of the literature. The values are certainly high, but they are not 1000 times higher than recent literature values. For example, other studies also report high INP concs: Petters and Wright (2015) show values up to 1000 L-1 and O'Sullivan et al. (2018) report values approaching 100 L-1 and Suski et al. (2018) report values in excess of 100 L-1.

15. P 9 ln 4. What are 'controlled-experiments'?

16. Conclusions: I found this hard to follow. There is a lack of structure and several statements do not seem to follow on logically. E.g.: In ' The insignificance of dry-heating was demonstrated with the increase of organics found for the ICR of dry-heated samples' the second statement does not follow on from the first.

References

O'Sullivan, D., Adams, M. P., Tarn, M. D., Harrison, A. D., Vergara-Temprado, J., Porter, G. C. E., Holden, M. A., Sanchez-Marroquin, A., Carotenuto, F., Whale, T. F., McQuaid, J. B., Walshaw, R., Hedges, D. H. P., Burke, I. T., Cui, Z., and Murray, B. J.: Contributions of biogenic material to the atmospheric ice-nucleating particle population in North Western Europe, Scientific Reports, 8, 13821, 10.1038/s41598-018-31981-7, 2018.

Petters, M. D., and Wright, T. P.: Revisiting ice nucleation from precipitation samples, Geophys. Res. Lett., 42, 8758-8766, doi:10.1002/2015GL065733, 2015.

Suski, K. J., Hill, T. C. J., Levin, E. J. T., Miller, A., DeMott, P. J., and Kreidenweis, S. M.: Agricultural harvesting emissions of ice-nucleating particles, Atmos. Chem. Phys., 18, 13755-13771, 10.5194/acp-18-13755-2018, 2018.

---

## Author Comment (AC2) · 22 Apr 2021

**Response to Referee #1**

First of all, the authors thank the referee for submitting helpful and meaningful comments, which lead to improvements and clarifications within the manuscript.

Below, we provide our point-by-point responses. For clarity and easy visualization, the referee's comments (**RC**) are shown from here on in black. The authors' responses (**AR**) are in blue color below each of the referee's statement. In addition to the responses to referees' comments, we further modified the manuscript to increase its clarity and readability. The summary of other changes is included at the end of this document. We introduce the revised materials in green color along/below each one of your response (otherwise directed to the Track Changes version manuscript). All references are available in the end of this AR document.
* * *
**RC**: This study reports the immersion ice nucleation ability of ambient feedlot aerosol samples and surface materials. Several instruments were engaged to perform physical, chemical and biological characterization, as well as immersion freezing ability determination of these filter samples and proxies. The authors have done great job characterizing and the presented data is abundant. However, the writing quality is pretty poor and the authors did not seem to offer a consistent usage of terms and symbols throughout the paper, which in part is hard to follow. More importantly, there are a few aspects need to be sufficiently addressed before this paper can be reconsidered for acceptance: First, the writing quality does require substantial improvement. Second, the paper structure is imbalance, with about 50% pages on materials and methods yet I still feel difficult to get the logic connection and setup of these experiments. Third, the Results and Discussion part is basically an experiment report with little analysis and discussion.
**AR**: The authors appreciate these general remarks and constructive criticisms regarding our manuscript by referee #1. We found these as invaluable guidance. We believe that the analysis and discussion in the revised manuscript are robust. We have very good data from our three-year field survey and the lab setting. However, some insufficient discussions might have led some of our data interpretation and presentation in an original manuscript to be unclear. The authors appreciate the peer-review comments, which motivated further analyses and improved the overall presentation. To allay the reviewer's concerns and mitigate any misgivings, the authors decided to change the title of manuscript to "**Ice-nucleating particles from open-lot livestock facilities in Texas**", reflecting our changes and articulate what is truly presented in the revised version paper. We have also revised our abstract, the conclusion section, and overall structure to reflect all of our major revisions and to increase the readability of this paper with rigorous analysis and discussion (please read the Track Changes version paper). Further, a consistency of terms and symbols has been checked. Below, we provide our point-by-point responses in hopes of our manuscript being considered for another review by the reviewer.

Major comments:
**RC**: 1. Introduction: The logic of this part needs to be thoroughly improved. A much more detailed review of relevant previous work should be added.
**AR**: The logic of this section has been improved by introducing (1) the climatic impact of ice-nucleating particles (INPs), (2) previous fertile-and-agricultural soil dust-derived INP studies (Suski et al., 2018; Conen et al., 2011; Hill et al., 2016; Steinke et al., 2016; 2020; Tobo et al., 2014; O'Sullivan et al., 2014 – more discussion along with our results in Sect. 3.6), and (3) potential significance of soil dust INPs in the U.S. as well as Texas (and the reasons) in the first four paragraphs.

*RC*: The following points should be explicitly addressed to give the audience an idea about the importance of this work: Why is feedlot dust relevant to mixed-phase clouds?

*AR*: Our research hypothesis and objectives are now also clarified in the introduction section as, "Due to the potential to act as a prevalent point source of microbiome-enriched dust particles in the Southern High Plains region, where a convective cloud and updraft system persists (Li et al., 2017), we hypothesized that an OLLF can be a source of soil dust INPs. To verify this hypothesis, IN propensities of aerosol particles from OLLFs, IN efficiencies of OLLF proxies, and their physicochemical and biological properties were studied in both field and laboratory settings."

The authors also reformulated the third paragraph of our introduction section to point out the high loading of feedlot dust and potential impact of erosion by citing previous contributions and findings made by our team (i.e., Von Essen, S. G. and Auvermann, 2005; Bush et al., 2014; Hiranuma et al., 2011; Steinke et al., 2020) as follows; "In particular, the Texas Panhandle (northern most counties of Texas; also known as West Texas) is a major contributor to the U.S. cattle production, accounting for 42% of fed beef cattle in the U.S. and 30% of the total cattle population in Texas (> 11 million head). Annually, these cattle produce > 5 million tons of manure, which represents a complex microbial habitat containing bacteria and other microorganisms, on an as-collected basis (Von Essen and Auvermann, 2005). Agricultural dust particles observed at OLLFs have long been known to affect regional air quality because the dust emission flux and 24-hour averaged ground-level dust concentration can be as high as 23.5 $\mu$g m$^{-2}$ s$^{-1}$ and 1,200 $\mu$g m$^{-3}$ (Bush et al., 2014; Hiranuma et al., 2011). Furthermore, our previous study revealed a presence of OLLF-derived particles at 3.5 km downwind of the facility, suggesting their ability to be transported regionally (Hiranuma et al., 2011). Moreover, some recent studies suggest that aerosol particles emitted from agricultural activities might reach cloud heights due to wind erosion, scouring, and other relevant mechanisms (Steinke et al., 2020 and references therein; Duniway et al., 2019; Katra, 2020)."

*RC*: How representative (e.g. size distribution, chemical composition, etc.) are the surface samples collected in this study compared to actual ambient aerosols?

*AR*: **Composition:** Organic particles with small salt contents (e.g., potassium) are predominantly present in surface-derived OLLF particle composition from this study, which is consistent with our previous study of ambient OLLF soil dust particle composition analyses (Hiranuma et al., 2011). Thus, in terms of composition, the authors believe that our surface-derived samples represent atmospherically relevant OLLF dust. Further, our study revealed a relative increase in organic inclusions (and decrease in salt inclusion) in ice crystal residuals, highlighting the importance of organic material for atmospheric immersion to be OLLF-derived INPs. Our previous work using Raman micro-spectroscopy revealed that ambient aerosol particles sampled at OLLFs contains brown or black carbon, hydrophobic humic acid, water soluble organics, less soluble fatty acids, and carbonaceous materials mixed with salts and minerals. But, our current knowledge regarding IN active organics is still limited. In the future, more detailed research should focus on understanding IN active organics common in soil dust. These important remarks are now stated in the new conclusion section (please see the track change version manuscript).

**Size:** As shown in Fig. 5 of Hiranuma et al. (2011), the surface area distribution of ambient OLLF dust measured with a combination of a GRIMM sequential mobility particle sizer and a GRIMM portable aerosol spectrometer peaks in a mode at ~ 10 $\mu$m diameter. This mode diameter is somewhat larger than what we measured after aerosolizing our surface-derived samples in the AIDA chamber (summarized in Table 3). Note that we used cyclone impactors for the injection of OLLF surface sample into the AIDA to reduce the concentration of large aerosol particles to not misclassify them as ice crystals. But, it should be also noted that the ambient OLLF dust size distribution is not spatially uniform, and the emitting mechanism itself is not controllable as it highly depends on a unit of mobile livestock. The difference mentioned above and the demand of controlled investigation were exactly our motivation of analyzing IN properties of both bulk samples (< 75 $\mu$m-sieved) and aerosolized samples ($\lessgtr$ 6.5 $\mu$m). While we observed

a reasonable agreement between bulk and aerosolized IN propensity for the TXD05 sample within our experimental uncertainty is represented by 95% binomial confidence interval, there was a substantial difference of IN efficiency in between bulk and aerosolized one of the TXD01 sample. Overall, these results imply that TXD05 might be more atmospherically relevant. These points are now addressed in our new Sect. 3.2. To clarify our intention, we added the following sentence in the end of Sect. 2.2 – "As demonstrated in our previous study, the surface area distribution of ambient OLLF dust peaks in mode diameter at ~ 10 μm (i.e., Fig. 5 of Hiranuma et al., 2011). This mode diameter is larger than surface-derived samples aerosolized and examined in the AIDA chamber (**Table 3**). However, it is cautiously noted that the ambient OLLF dust size distribution is not spatially uniform, and the emitting mechanism itself is not controllable as it highly depends on a unit of mobile livestock. Granting the primacy of hoof action as the decisive emissions mechanism of OLLF dust as described in Bush et al. (2014), a more controlled laboratory experiment has been desired to characterize IN ability of OLLF soil dust. The difference mentioned above and the demand for controllable investigation motivated analyzing IN properties of both bulk samples (< 75 μm-sieved) and aerosolized samples ($\lesssim$ 6.5 μm). Further results and discussions about representativeness of the surface samples used in this study compared to ambient OLLF soil dust are provided in **Sect. 3**."

*RC*: How representative are the sampling sites for the whole Texas?
*AR*: This is a valid question. We now clarify the representativeness of our field study sites in Sect. 2.1 as follows; "Four commercial OLLFs, ranging from 0.5 to 2.6 km$^2$ (< 45,000 head capacity), located in the Texas Panhandle region were used as the ambient aerosol particle sampling sites. All four sites are located within a 53 km radius of West Texas A&M University in Canyon, Texas. Our experimental layouts at each site, denoted as OLLF-1 to OLLF-4, are shown in **Fig. 1** (no further specification is provided to protect location privacy). All sites have a capacity greater than 1,000 head, which are categorized as large concentrated animal feeding operation facilities for cattle under the U.S. Environmental Protection Agency's definition. These OLLFs were selected primarily for the east-west orientation of their feeding and working alleys, which were nearly orthogonal to prevailing south to southwest winds, allowing for downwind and upwind sampling. Our sampling sites represent typical OLLFs, as more than 75% of cattle are produced in large concentrated animal feeding operation facilities in the U.S. (Drouillard, 2018)."

*RC*: The title made a very broad and general statement without any solid support.
*AR*: The authors adapted a new title of "**Ice-nucleating particles from open-lot livestock facilities in Texas**" to reflect our changes and articulate what is truly presented in the revised version paper.

*RC*: 2. Materials and methods: The authors used 4 out of 9 pages to describe the materials and instruments used in this study. But it remains difficult to structure the experiments conducted. Please streamline this section and provide necessary experiment information.
*AR*: This is a valid point. The authors made a major revision in the manuscript structure, and the revised manuscript has the following structure and balanced page allocations; (1) Introduction: Page 1-2, (2) Methods: Page 3-6, (3) Results and Discussion: Page 6-10, and (4) Conclusion: Page 11.

*RC*: 3. As the main body of the manuscript, the results and discussion section is divided into 7 separate subsections without logical connection. Most of these subsection, simply report values and lack further analysis and discussion of the results. For instance, it is claimed that super-micron INP fraction contributed 49.7% ± 6.0% of total INP for TXD samples at -18°C and -22°C (P7L20-22). However, no analysis or interpretation of this conclusion is presented. In Table 4 (P23), the change of super-micron INP fraction at two temperatures before and after the dry-heated treatment is different, and what could be the reason

for this deviation? Please revise this part in depth and add comparisons of the results obtained in this study to previous relevant work.

*AR*: First, it is a good suggestion to add comparisons of the results obtained in this study to previous relevant work. The authors added the following comparative discussion in Sect. 3.2. The authors think that having this discussion in this section gives a smooth transition and also provides a link to Sects. 3.3. and 3.4. So, we thank the referee for this invaluable suggestion. The authors note that there has not been much discussion of large soil dust organics and their contribution to atmospheric ice nucleation in previous studies. Tobo et al. (2014) mentions the need for analyzing soil dust INPs using mono- vs. poly-disperse particles, and the authors agree with the necessity of a systematic study of size-segregated soil dust INPs in the future (beyond the scope of this work). Hence, direct implication of what composition contributes to IN at different $T$s of OLLF particles is still not available. Further, while we did not see a systematic increase of supermicron INP fraction as a function $T$ as shown in Mason et al. (2016; i.e., INP fraction at -15 °C > that of -20 °C), our results in **Table 4** support that $n_{INP,total}$ is always higher than $n_{INP,PM1}$ for any type of samples used in this study. These points are now addressed as follows; "Since DeMott et al. (2010) successfully demonstrated the correlation between immersion-mode $n_{INP}$ and the number concentration of aerosol particles larger than 0.5 µm diameter based on the compilation of field data for more than a decade, a number of studies have shown the evidence that supermicron aerosol particles dominate INPs across the world. For example, Mason et al. (2016) reported a substantial fraction of supermicron INPs through immersion freezing at relatively a high $T$ (> 78% at -15 °C) measured at seven different sites over North America and Europe. Even at -20 °C, the author reported the fraction of supermicon INPs larger than 50%. Compared to these numbers, our laboratory data show lower fractions, but the INP sources are presumably different. Based on findings from recent study of size-resolved INPs vs. fluorescent biological particles, these INPs activated at −15 °C are typically thought to be biological (e.g., Huffman et al., 2013; Huang et al., 2021). While there has been more evidence that terrestrial and marine biological particles play an important role in immersion freezing of supermicron-sized particles (e.g., Ladino et al., 2019; Si et al., 2018; Creamean et al., 2018), the atmospheric implication of such rare aerosol species and the overall impact on aerosol-cloud interactions is still under debate. More recently, high IN efficiency by supermicron INPs derived from quartz-rich atmospheric mineral dusts have been reported from different locations, including East Asia (Chen et al., 2021) and eastern Mediterranean (Reicher et al., 2019). These mineral components usually contribute to IN at low $T$s. However, there has not been much discussion of large soil dust particles, especially organics, and their contribution to atmospheric ice nucleation in previous studies. Hence, direct implications of which components contribute to IN at different $T$s to the observed freezing properties of OLLF particles is still missing. Lastly, while we did not see a systematic increase of supermicron INP fraction as a function of $T$ as shown in Mason et al. (2016; i.e., INP fraction at -15 °C larger than at -20 °C), our results in **Table 4** support that $n_{INP,total}$ is always higher than $n_{INP,PM1}$ for any type of samples used in this study."

Second, the letter-type manuscript content (remained for this particular section) is now revised to fit in the research manuscript format. The authors believe that the revised version provides sufficient analysis of the results and observations.

**Specific comments:**
*RC*: - P1L18: Please capitalize each word.
*AR*: Each keyword is now capitalized.

*RC*: - P1L21: Please add "the" before U.S. and check the usage of "the" throughout.
*AR*: Added and all consistent throughout now. The article usage in the main manuscript and SI has been re-checked with a native English speaker/PhD.

*RC*: - P1L28: "thereby finding…" is not clear to me.
*AR*: Rephrased to "leading to find…".

RC: - P1L30: Please replace "improved" with "improving".
AR: Replaced. Thank you.

*RC*: - P1L32: Please rephrase to "…explore the relationship between INP and supercooled cloud properties…".
*AR*: Rephrased as suggested.

*RC*: - P1L38: 25% of total what?
*AR*: We meant total "total global dust emission" as stated in Sect. 7 of Ginoux et al. (2012)

*RC*: - P1L46: Please rephrase.

*AR*: We rephrased this sentence now in L89 as "Agricultural dust particles observed at OLLFs have long been known to affect regional air quality because the dust emission  flux and 24-hour averaged ground-level dust concentration can be as high as 23.5 $\mu$g m$^{-2}$ s$^{-1}$ and 1,200 $\mu$g m$^{-3}$ (Bush et al., 2014; Hiranuma et al., 2011)."

*RC*: - P1L48: "suggesting their regional scale impact" is not clear to me.
*AR*: For clarity, we rephrased these words to "suggesting their ability to be transported regionally".

*RC*: - P1L50: Please verify the usage of "impact in".
*AR*: It should have been "influence on", but the authors decided to remove this sentence to improve the logical flow of this section with only the most relevant discussions and references.

*RC*: - P1L51: Please spell out "IN", as this is the first appearance of this term in the text. Please check the whole paper for first appearance of abbreviations.
*AR*: Corrected. The authors apologize for missing abbreviation definitions in the main text. Admittedly, we overlooked the guideline given in the ACP web - "They need to be defined in the abstract and then again at the first instance in the rest of the text."

*RC*: - P1L54: Please add the classical textbook of Pruppacher and Klett (2010). Please also spell out "INP".
*AR*: Pruppacher and Klett (2010) is added, and INP is defined at its first appearance.

RC: - P1L56: Please rephrase "focused in this study".
AR: The authors rephrased the sentence and now it read, "We focused on the immersion mode freezing because recent modeling simulation and remote sensing studies suggest that immersion freezing is the most prominent heterogeneous IN mechanism, accounting for 85 to 99%, through which ice crystals are formed in mixed-phase clouds (Hande and Hoose, 2017; Westbrook and Illingworth, 2011)."

*RC*: - P2L14: Please spell out "Ts" or specify the meaning of the subscript.
*AR*: Defined - temperatures (*Ts*).

*RC*: - P2L15: Please spell out "ICR".
*AR*: Defined - ice crystal residual (ICR) samples.

*RC*: - P2L39: Please add "as" before addressed.
*AR*: Added as suggested.

*RC*: - P2L43: Please spell out "AIDA".
*AR*: Defined.

*RC*: - P2L47: Please spell out "TXD".
*AR*: Defined - Texas-Dust-01 (TXD01) and Texas-Dust-05 (TXD05)

*RC*: - P2L58: What instrument did the author use to measure BET SSA?
*AR*: Autosorb iQ model 7 gas sorption system (Anton Paar, former Quantachrome Instruments) – we rephrased the sentence as follows, "The Autosorb iQ model 7 gas sorption system (Anton Paar, former Quantachrome Instruments) was used to measure BET SSAs in this study."

*RC*: - P3L1: Does the author imply that TXD05 is soil dust while TXD01 contains K-feldspar here? If so, why is the density of TXD05 larger than that of TXD01?
*AR*: No, the authors did not mean or imply an inclusion of K-feldspar. We simply meant that the increase in porosity can lead to higher BET SSA. To avoid any confusions, the authors decided to limit our BET SSA comparison to a previous soil dust study (O'Sullivan et al., 2014) and exclude two K-feldspar study references. Excluding these references does not change the story of this paper.

*RC*: - P3L3-8: Why did the author compare geometric and BET SSA directly and attribute the difference between these two totally different methods to sample particle size variance?
*AR*: Both surface parameters are important for ice nucleation active surface site density estimations – $n_{s,BET}$ and $n_{s,geo}$ (Hiranuma et al., 2015), which is cited within the relevant sentence. Since BET SSA is typically measured for 'bulk' sample, it may not represent the specific surface area of aerosolized particles, which was discussed in our previous study (Hiranuma et al., 2019).

*RC*: - P3L15: Please see the comment above.
*AR*: Pruppacher and Klett (2010) is added above. At the first appearance of immersion freezing (L54), we decided to give its definition (i.e., the freezing propensity of INP immersed in supercooled water) and cite relevant references of soil dust immersion freezing (Suski et al., 2018; Conen et al., 2011; Hill et al., 2016; Steinke et al., 2016).

*RC*: - P3L17: What is the purpose of filter sampling? Can the authors provide a scheme of the experimental setup in this study?
*AR*: These are all clarified in Sect. 2.3. The authors thank the referee for this suggestion to use experimental schematics. The authors believe that Fig. 2 clarifies and improves the readability of the section.
    "An overall AIDA experimental schematic is shown in **Fig. 2**. Our OLLF dust proxy sample was injected into the AIDA chamber in an aerosolized form through a rotating brush disperser (PALAS, RGB1000) followed by passing through a series of inertial cyclone impactor stages to limit particle size to < 10 µm in $D_{ve}$. Subsequently, the OLLF particle size distribution in the AIDA chamber was measured prior to each simulated adiabatic expansion experiment. Specifically, a combination of a scanning mobility particle sizer (SMPS, TSI Inc., Model 3080 differential mobility analyzer and Model 3010 condensation

particle counter), an aerosol particle sizer (APS, TSI Inc., Model 3321), and a condensation particle counter (CPC; TSI Inc., Model 3076) collectively measured the total number and size distribution of aerosol particles at a horizontally extended outlet of the AIDA chamber (Möhler et al., 2006). As seen in **Fig. 2**, a set of complementary filter samples of the aerosol particles directly from the AIDA chamber was also collected prior to expansion experiments for three purposes: (1) examining the condensation/immersion freezing ability of aerosol particle collected on nitrocellulose membrane filters (Millipore HABG04700, nominal porosity 0.45 μm) in the dynamic filter processing chamber (DFPC; Santachiara et al., 2010), (2) using them to perform measurements with the IN Spectrometer of the Karlsruhe Institute of Technology (INSEKT; Schiebel, 2017; Schneider et al., 2021), and (3) conducting metagenomics analyses to study biological components of the aerosolized samples. Afterwards, each particle type (i.e., TXD01 and TXD05) was individually examined for its immersion freezing ability during expansion experiments. To complement the AIDA chamber immersion results, INSEKT was used for aerosol particles collected on 47 mm Nuclepore filters (Whatman WHA10417012, pore size 0.2 μm) as well as for < 75 μm sieved-bulk samples collected. The DFPC technique was also used to measure the number concentration, ice-activated fraction, and nucleation efficiency of the INPs under different $T$ conditions and for different particle sizes (i.e., $PM_1$ vs. total) collected on filters. DNA sampling for metagenomics analysis to study biological components of the OLLF bulk samples was also conducted on aerosol particles collected on the Nuclepore filters through an independent inlet."

[Figure]

**Figure 2.** Lab experimental schematic of the AIDA facility. All samples were injected using a rotating brush generator (RBG) for aerosol particle generation. Multiple extramural instruments, welas optical particle counters (OPCs), an ice selective pumped counterflow virtual impactor (IS-PCVI), a hygrometer, a tunable diode laser (*TDL*) spectrometer, a laser ablation aerosol particle time-of-flight mass spectrometer (LAAPTOF; see SI), and aerosol particle counters/sizers (SMPS, APS, CPCs), are connected to the AIDA chamber. Downstream filters and an impactor collected aerosol particles and ice crystal residuals for multiple offline analyses.

In addition, we decided to provide a field experimental schematic as Fig. 1.

[Figure]

**Figure 1.** Schematic of the field sampling activity at individual sites. The dimension of each facility (east – west × north – south) is (1) 1.6 × 1.6 km, (2) 1.0 × 0.8 km, (3) 0.7 × 0.7 km, and (4) 0.8 × 1.4 km. A combination of polycarbonate filter samplers (PFSs) and DustTrak instruments was used at the nominally upwind and downwind edges of OLLF-1 to OLLF-3. Two tapered-element oscillating microbalances (TEOMs) were deployed at OLLF-1 alongside other instruments.

*RC*: - P3L24: Please add reference after "IS-PCVI".
*AR*: Hiranuma et al. (2016) is now added.

*RC*: - P3L28: Should be µm here.
*AR*: Corrected.

*RC*: - P3L31: Does the authors mean "individual ice crystal" here? If so, maybe it's better to delete "of individual particles".
*AR*: We do not analyze the composition of ice crystals, we analyze leftover residual particles after evaporating all moisture. So the authors decided to reword it to "individual residual particles"

*RC*: - P3L41: Please add more AIDA references apart from the review paper.

*AR*: Steinke et al. (2020), Ullrich et al. (2017), and Niemand et al. (2012) are added.

*RC*: - P3L42-45: According to Table 3, the super-micron particle size can reach 5-6 µm. What's the size detection limit of TSI SMPS and aerosol particle sizer?
*AR*: We used a combination of SMPS and APS to characterize the size distribution of particles injected into AIDA (addressed in Sect 2.3). The APS measures aerodynamic particle size from 0.5 to 20 µm. As stated in the manuscript, the SMPS and APS data were merged. The authors followed the method introduced in Möhler et al. (2006).

*RC*: - P3L45: What technique did the authors use to aerosolize particles?

*AR*: We used a rotating brush disperser (RGB 1000, Palas) as used in Möhler et al. (2006).

*RC*: - P4L2: Did the authors intend to say "array"?
*AR*: We meant "a set of", and this part has been updated accordingly.

*RC*: - P4L7: Please spell out "HPLC".
*AR*: Done - high-performance liquid chromatography (HPLC)

RC: - P4L8-9: How was the detection limit determined?
AR: Suspension water volume was optimized to assign first frozen droplet to correspond to 0.05 INP L$^{-1}$. A recently published paper describing the WT-CRAFT method (including the raised point) and its application (Vepuri et al., 2021) is added as a reference.

*RC*: - P4L11-12: Can the authors provide an example for such processing in SI?
*AR*: Our previous product provides a good example of visual separation of droplet-ice (Fig. 4 of Cory et al., 2019a; DOI: https://doi.org/10.1002/essoar.10500739.1). As other specifications of WT-CRAFT are sufficiently provided in Cory (2019b) and Vepuri et al. (2021), the authors would like to avoid including this in SI.

*RC*: - P4L25-26: If it's "a series of diluted suspensions", should it be "×15 **to** ×225" here? Besides, please replace "x" with "×" throughout the paper.
*AR*: All corrected in a consistent manner. Thank you.

*RC*: - P4L38: Remove the "of" after "100 L".
*AR*: Removed.

*RC*: - P4L40: "was characterized".
*AR*: Corrected.

*RC*: - P4L42: Please specify what type of "diameter".
*AR*: Aerodynamic

*RC*: - Sect. 2.6: Please simplify the description of previous work.
*AR*: The authors believe that we need to retain all information in this section (now Sect. 2.5).

*RC*: - P5L31: What is the typical size range of droplets and ice crystals in this study?
*AR*: Droplet were typically < 24 $\mu$m and ice particles were > 24 $\mu$m optical diameter. For clarity, the figure shown below displays temporal evolution of droplet and ice crystal size distributions during the four ICR extraction experiments. Normalized size distributions of simulated cloud particles (d$N$/dlog$D_p$) were derived from the measurements in the welas optical particle counter installed below the AIDA chamber. As shown, a reasonable critical cut-size (> 24 $\mu$m) of the IS-PCVI was used for each expansion experiment.

[Figure]

**Figure**. Temporal profiles of the welas OPC size distribution of droplets and ice crystals in optical diameter ($D_p$) during individual expansions of TXDUST01_08 (a), _04 (b), _12 (c) and _31 (d). The red dotted lines represent the critical cut size, $D_c$.

*RC*: - P5L36: Should be µm. Please check the units throughout.
*AR*: Sorry. Checked and all corrected.

*RC*: - P5L37: During TXDUST01 what?
*AR*: We rephrased the sentence as: "During the TXDUST01 campaign, the output flow of IS-PCVI was constant at 2.5 lpm." – it is now in SI Sect. S1.

*RC*: - P5L45: Please change the parenthesis to "(…Model 1400a; Patashnick…)".
*AR*: Changed to "TEOMs; Thermo Scientific Inc., Model 1400ab; Patashnick and Rupprecht, 1991."
Note: 1400ab is a correct Model no.

*RC*: - P5L58: Please enclose "<100 µg m-3" into parenthesis.
*AR*: Enclosed.

*RC*: - P6L3-9: What's the point of repeating previous results in this paper?
*AR*: It was to clarify the observed supermicron ambient dust originates from a feedlot. As we now discuss this point using our new data, we decided to remove this repetitive discussion.

*RC*: - P6L11: I suggest to place the symbol right after the corresponding definition. Do the author mean INP concentration per unit particle mass and INP concentration per unit particle surface? Please specify and avoid misunderstanding.
*AR*: This is a good suggestion. Corrected as suggested.

*RC*: - P6L22: How were these quantities derived? Can the authors please provide the derivation or conversion process in SI apart from the references here?

*AR*:
The following is now available in SI Sect. S4 – derivation of $n_{INP}$, $n_m$, and $n_{s,geo}$:

**S4.** Derivation of $n_{INP}$, $n_m$, and $n_{s,geo}$

Here we describe the conversion procedure used to derive ambient $n_{INP}$, $n_m$, and $n_{s,geo}$. Initially, we computed the $C_{INP}(T)$ value, which is the nucleus concentration in ultrapure water suspension ($L^{-1}$ water) at a given $T$ as described in Vali (1971). This $C_{INP}(T)$ value was calculated as a function of unfrozen fraction, $f_{unfrozen}(T)$ (i.e., the ratio of number of droplets unfrozen to the total number of droplets) as:

$$C_{INP}(T) = -\frac{\ln\left(f_{unfrozen}(T)\right)}{V_d} \tag{1}$$

in which, $V_d$ is the volume of the droplet (3 $\mu$L) for WT-CRAFT and sample in a well (50 $\mu$L) for INSEKT.
Next, we converted $C_{INP}(T)$ to $n_{INP}(T)$: INP in the unit volume of atmospheric air at standard $T$ and pressure (STP) conditions, which is 273.15 K and 1013 mbar. The cumulative $n_{INP}$ per unit volume of sample air, described in the previous study DeMott et al. (2017), was then estimated as:

$$n_{INP}(T) = C_{INP}(T) \times (DF) \times \frac{V_l}{V_{air}} \tag{2}$$

where DF is a serial dilution factor (e.g., DF = 1 or 10 or 100 and so on). The sampled air volume ($V_{air}$) is given in **Table 1**. The suspension volume ($V_l$) is optimized to achieve the detection limit of 0.05 INP $L^{-1}$ (corresponding to the first frozen droplet).
Finally, based on Eqn. 3 of Hiranuma et al. (2015), the $n_{s,geo}(T)$ and $n_m(T)$ values can be derived as:

$$n_m(T) = \frac{n_{INP}(T)}{M_{ve}} \approx \left(\frac{S_{total}}{M_{total}}\right) n_{s,geo}(T) \tag{3}$$

where $M_{ve}$ is the mass of a spherical particle of volume equivalent diameter (g), and $S_{total}/M_{total}$ is a geometric specific surface area (Hiranuma et al., 2015). The value used for converting field $n_m(T)$ to $n_{s,geo}(T)$ data, ~ 0.4 $m^2\ g^{-1}$, is derived from particle size distribution measurements presented in Fig. 3 of Hiranuma et al. (2011).

*RC*: - P6L23: Please verify the usage of "Regardless".
*AR*: Done. For clarity, the authors now simplify this sentence by combining with previous sentence as; "While the background freezing contribution of the field blank filter was negligible (< 3%) at -25 °C, we purposely limited our WT-CRAFT data analysis to the $T$ range between 0 °C and -25 °C to eliminate any possible artifacts in our WT-CRAFT data."

*RC*: - P6L25: How did the author decide whether the data was "uncertain systematically erroneous data"?
*AR*: This just means we are not confident with our WT-CRAFT data below -25 °C as artifacts become non-negligible (Vepuri et al., 2021), and we report our WT-CRAFT data only in the T range above -25 °C.
We have rephrased it to: to eliminate any possible artifacts in our WT-CRAFT data

*RC*: - P6L27-28: The difference between field samples and lab samples in Fig. 3(a) can reach up to three orders of magnitude between -20 to -10 °C, which is not "comparable" and is not capable of "validating" from my point of view.
*AR*: This is a valid question. The authors took the referee's word as a motivation of further analyzing our data for spatial variability (Fig. 3) and seasonal variability (Figs. 4-6) to explain the observed data variation and completely reformulated Sect. 3.1. Please see the track changes manuscript.

*RC*: - P6L31: Is this linear relationship a linear correlation analysis or a linear regression fit? Please specify, and give the corresponding correlation coefficient and/or other parameters to indicate the goodness of this correlation.
*AR*: The authors apologize for extending the assessment to the raised point. The following description now appears in the Fig. 5 caption.
- Yes. It is a linear regression curve in log scale ($n_{INP}$ = 3.51 x Cumulative PM Mass – 2.41; r = 0.94).
- This is the value of representative $n_m$ at the given $T$ (3.55 x $10^9$ $g^{-1}$), which is a median $n_m$ value of minimum to maximum in order to guide the reader's eye and to show an example of quasi-constant $n_m$ of cattle feedlot-derived INP's nucleation efficiency.

[Figure]

**Figure 5**. Correlation between cumulative PM mass vs. $n_{INP}$ (a) and vs. $n_m$ (b) at -25 °C; a linear regression curve in log scale ($n_{INP}$ = 3.51 × Cumulative PM Mass – 2.41; $r$ = 0.94) is shown in (a), and the constant value of representative $n_m$ at the given $T$ (3.55 × $10^9$ $g^{-1}$), which is a median $n_m$ value of minimum – maximum, is shown in (b). Note the errors in cumulative PM mass are ± 40.4% as discussed in **Sect. 3.1**. The uncertainty in $n_{INP}$ and $n_m$ is ± 23.5%.

*RC*: - P6L31-32: Again, how was the "INP scaled to mass" conversion achieved?
*AR*: Explained above with equations.

*RC*: - P6L33-34: The authors claimed that ambient meteorological conditions might not be determining factors for INP concentrations. What about the difference between different sampling sites? Is there any previous study drew the same conclusion?

*AR*:  As all of our study sites are located in the close proximity in the same Texas Panhandle region, we do not expect inter-spatial variabilities of local meteorology. As discussed above, the ambient OLLF dust properties are not spatially uniform, and the emitting mechanism itself is not controllable as it highly depends on a unit of mobile livestock and perhaps feeding practices etc. The authors are not aware of any previous studies conducted in a similar set up. We have clarified that this observation applies for our study sites only.

 "These results imply the following: (1) ambient meteorological conditions, summarized in **Table 1**, might not be determining factors for $n_{\mathrm{INP}}$ for our study sites;…"

*RC*: - P6L34: Please add "as" before "summarized".
*AR*: Added. Thanks.

*RC*: - P6L38: "several hundred INPs L-1" seems very high. Is there any previous report of such superior IN activity of ambient dust? Besides, please specify the "notable correlation" with solid particle size distribution and IN activity data.

*AR*: This is a good question. Now, the revised manuscript discusses this point in Sect. 3.6. Comparison to previous soil dust IN studies. Please see our revised manuscript.

*RC*: - P6L49: What does "they" refer to? Who/what agree with who/what?
*AR*: We have rephrased the sentence to clarify this as; "As explained in **Sect. 2.4**, a series of diluted samples were examined in INSEKT. We made sure to assess overlapping $T$ intervals in a series of measurements to see if $n_{\mathrm{s,geo}}$ values from multiple measurements  agree within CL95% and, if so, to merge the results together."

*RC*: - P6L52-53: The conclusion drawn here is vague and speculative. Why does different sample source affect the IN ability?
*AR*: We meant to address that the different samples possess different physicochemical properties. To clarify this point, we have rephrased the whole sentence to;

    "These results suggest that (1) there is a difference in the INP abundance between bulk (< 75 μm-sieved) and aerosolized/filtered-samples for TXD01 ($\lessapprox$ 6.5 μm; **Table 3**) presumably due to different properties in particles of these two size subsets (6.5 – 75 μm and $\lessapprox$ 6.5 μm) and/or different amount of IN-active soil organic matter (Tobo et al, 2014), (2) different physicochemical properties found for our TXD05 samples may not impact their INP propensies, and (3) TXD05 might be more representative of atmospherically relevant dust (see **Table 2** and **SI Sect. S2**)."

*RC*: - P7L1-11: This paragraph should move to Sect. 2.
*AR*: Moved to Sect. 2.4.

*RC*: - P7L14-15: The difference between heated and unheated samples measured by INSEKT and DFPC exhibited different trend instead of "not apparent" as the authors stated, please explain.
*AR*: The referee is right. The authors decided to detail and clarify the comparison of the non-heat-treated sample to the heated-sample is discussed in **Sect. 3.6** and **SI Sect. S3**. Please see the revised manuscript.

*RC*: - P7L12-16: The comparison of the instruments is irrelevant to the purpose of this section and should go to SI. In addition, why are the results of WT-CRAFT not added to compare the three instruments simultaneously?

*AR*: WT-CRAFT was primarily used to analyze the field samples and not used for systematic measurements of TXD01 and TXD05. The instrument comparison discussion is now in SI Sect. S3.

*RC*: - P7L28-30: Again, the result and interpretation seem very speculative. Is there any physical/chemical/biological evidence to support the statement here?

*AR*: The authors agree. This speculative part and associated reference have been removed from the manuscript.

*RC*: - P7L32-33: How representative are the samples tested in this study to "natural supermicron-dominant INPs"? Again, very broad and general statement without support.

*AR*: This is a valid point. To reflect what is presented. The authors rephrased; natural supermicron-dominant INPs to: supermicron-dominant INPs from cattle feedyard, which can act as an important point source of agricultural INPs

RC: - P7L33-35: Did the authors compare the immersion parameterization proposed in this work with other widely used parameterization models? How consistent and different they are? Whether the new immersion parameterization is universal? Also: Fitting an immersion parameterization solely as a function of a single parameter, temperature, might be misleading.

AR: Offering a universal or world representative parameterization for agricultural INPs is not the scope of this work. As each OLLF represents a point source of fresh agricultural dust, we expect that it would have a different ice nucleation efficiency compared to aged/weathered dusts. Nonetheless, our new Fig. 8 shows a comparison of our $n_{s,geo}$ data with six relevant IN parameterizations of soil/desert dust. Our parametrization is comparable to these past agricultural soil IN parameterization but with some deviation. The $n_{s,geo}(T)$ parameterization scales to the total particle surface – so it can be parameterized as a function of T (Niemand et al., 2012), assuming that the time dependence can be neglected.

*RC*: The particle number and/or mass concentrations vary considerably around the world. Do the authors believe that the dataset of this parameterization is representative of the rest of the world?

*AR*: This is a good question – as responded in the last question, we did not intend to offer our parameterizations to represent agricultural INP across the world. Below we address in what sense(s), and to what extent, the dataset reported here can/cannot represent particulate-matter emissions from OLLFs the world over.

Qualitatively, it is reasonable to suppose that our TEOM data directly collected at the site represents mass concentrations of the fugitive aerosol particles because those aerosols derived from livestock manure and correspond to the dominant feedstuffs that the livestock have consumed. In this case, those dominant feedstuffs are steam-flaked corn (50-85% dry basis of the mixed diet), dry distiller's by-products (0-30%), alfalfa hay (0-15%), and cereal silage (either corn or sorghum, 0-20%) (Asem-Hiablie et al., 2015), with various adjuncts, minerals, supplements, and fed pharmaceuticals present in relatively smaller amounts. To the extent that microbes of these aerosols are traceable to the microbial profile in the rumen, that behavior may be modulated by either the identity and relative amounts of the diet's macro-components, or the nature and activity of any antimicrobials administered to the animals during their feeding period, or both. The species (i.e., *Bos taurus* vs. *Bos indicus*) and breed of the cattle may also influence aerosol characteristics through variations in optimal feed composition. Feeding practices for

beef cattle thus vary widely throughout the world. For these reasons, the authors can provide only a qualitative answer for this particular question.

Bush et al. (2014) presented, in a generalized form, the mathematical basis and workflow associated with estimating emission fluxes from an extensive area source. Taken together, that approach – to which many including Seibert (1998) have attached the name "inverse dispersion modeling," or IDM – implicitly concedes several methodological criticisms that relegate the resulting emission-flux estimates to the status of *relative* quantities rather than *absolute* quantities. Those criticisms have been helpfully and succinctly summarized in Botlaguduru (2009). First, the emission-flux estimates are model-dependent; given a set of atmospheric conditions and measured mass concentrations as the model inputs, any two different dispersion models will yield two distinct estimates of the source strength. Second, because direct measurement of fugitive emissions fluxes from extensive area sources is not possible with current technology, it follows that an emission flux inferred via IDM with one dispersion model must not be used as the source-strength input to a different dispersion model operating in forward mode, for example, to predict downwind concentrations under other weather conditions. In other words, emission-flux estimates from IDM may be thought of as "correct" or "accurate" only in relative terms, and only with respect to the precise dispersion model used in the IDM workflow. Third, even the concept of an emission flux – an emission rate per unit area, having units of $M\ L^{-2}\ T^{-1}$ – is itself deeply problematic when used to quantify the source strength of an OLLF.

Importantly, a common computational approach in the refereed literature obscures the third deficiency. It has long been understood that hoof action on dry, uncompacted manure is the primary mechanism responsible for fugitive dust emissions from OLLFs. However, because dispersion models traditionally used with agricultural area sources are developed on the assumption of spatially uniform (or even piecewise spatially uniform) source strength, another attribute of OLLFs known as the stocking density – the average number of animals housed per unit area, with units of # (head) $L^{-2}$ – was added as a scaling factor once the emissions flux had been estimated through IDM. The resulting, scaled source strength was given the name, "emission factor," which according to the U. S. Environmental Protection Agency is "a representative value that attempts to relate the quantity of a pollutant released to the atmosphere with an activity associated with the release of that pollutant." The implicit idea is that if the extensive magnitude of a polluting activity (PA) is known, and if the emission factor (EF) associated with that activity is known, then the emission rate or flux (Q) associated with that activity can be straightforwardly computed (EPA, 1995) as

$$Q = EF \times PA \hspace{4cm} \text{Eqn. [A]}$$

In equation [A], which renders the EPA formula but omits the factor for control-method effectiveness, the units of both EF and PA are selected source-type by source-type (a) to represent the emission mechanism faithfully so that scaling by magnitude is methodologically valid and (b) to yield in their arithmetic product the desired units of the source strength, Q. In cattle feeding, given that fugitive emissions result from (and scale by, in approximate terms) animal activity rather than production area, the appropriate units for EF are $M\ (hd^{-1})\ T^{-1}$, as in "lbs/hd/d." Correspondingly, the polluting activity (PA) can be described by the stocking density, hd $L^{-2}$, as in "hd/ac." The product of those two quantities yields a flux, $M\ L^{-2}\ T^{-1}$. However, in practice, the computation works the other way: Q is inferred from inverse dispersion modeling, with all of the methodological problems pertaining to it, PA is computed from an OLLF's production records, and EF is computed from equation [A]. By now, it should be obvious that EF is itself subject to exactly the same criticisms as was Q; the scaling process through equation [A] was merely a fig leaf to present the emission factor, EF, as the fundamental quantity – having units appropriate to the emissions mechanism!– even though it was itself derived from the model-contingent quantity, Q.

The cumulative weight of those criticisms may leave the reader suspecting that the criticisms are overly tendentious and convenient. However, the history of the cattle feedlot emission factor in EPA's own guidance document confirms the criticisms' validity as a practical matter. That guidance document, cited above as "AP-42," includes a set of quality ratings for the emission factors contained in it as guidance for state air pollution regulatory authorities. For many years (EPA, 1985), the AP-42 emission factor for cattle feedlots, a subtype of OLLFs, was rated "Class E (poor)," which denoted that the emission factor "is developed from C- and D-rated test data, and there may be reason to suspect that the facilities tested do not represent a random sample of the industry…there also may be evidence of variability within the source category population." Quite so, as cited research abundantly confirms. Still, EPA retained its AP-42 guidance for particulate-matter emissions from cattle feedlots until 1995, after which even that Class E emission factor has been omitted.

*RC*: - P7L44-45: Previous studies on bio-aerosol INP activity reported the importance OF ice active protein instead of DNA. Why didn't the authors conduct protein analysis?
*AR*: The ice nucleation protein is a minor protein of the outer membrane of bacteria with ice nucleation activity, about 1% of outer membrane proteins only (Morris et al., 2004) and therefore hard to use as a target in bioaerosol studies. DNA-based detection offers greater sensitivity because, theoretically, if the DNA of even one cell in a sample is successfully extracted, it will be amplified and detected in a metagenomics study. Moreover, the research team which produced this manuscript does not possess the instrumentation for protein detection.

*RC*: - P7L59: The data is not consistent with Table 7.
*AR*: Corrected. Thanks for catching this.

*RC*: - P7L6-9: Why do the authors refer to previous study instead of the single particle chemical analysis in this study to infer particle hygroscopicity?
*AR*: That is because we previously found there is an inherent relationship between the composition and the hygroscopicity of OLLF particles. We clarified this sentence as, "Higher aspect ratios in residuals compared to aerosol particles were found for both TXD01 and TXD05 samples. This difference indicates a relative increase in non-spherical particles, that have a higher aspect ratio, in residuals. In short, Hiranuma et al. (2008) found that quasi-spherical OLLF particles were predominantly salt-rich hygroscopic particles, whereas non-spherical amorphous particles were found to be organic-dominant with negligible hygroscopicity. Thus, our results suggest the inclusion of non-hygroscopic particles as ice residuals."

*RC*: - P7L17-18: Data presented in Table 8 does not support this statement. Why did the authors decide to omit the influence of carbonaceous content on particle IN activity?
*AR*: We did not omit the influence of carbonaceous content. EDX provides only elemental composition (no molecular-level info), and almost all of our particles contained organics, which is consistent with our previous study with Raman (H11). We believe that we addressed in text corresponds to what we show in Table 8 with up and down arrow marks. In addition, the authors decided to address the importance of future work regarding soil dust-derived organic INPs in Sect. 4 as, "…Based on findings from this study, ICR analysis revealed a relative increase in organic inclusion (and decrease in salt inclusion) in residuals, highlighting the importance of organic material in OLLF-derived INPs for atmospheric immersion. Even after dry heating treatment, the increase in organic fraction was found in the ICR of our OLLF samples. Therefore, the investigation of heat-insensitive organics is key to further understand the properties of soil dust INPs, and further research should focus on understanding how organic composition influences IN. Our previous work using Raman micro-spectroscopy revealed that ambient aerosol particles sampled at OLLFs are internally mixed with brown or black carbon, hydrophobic humic acid, water soluble organics,

less soluble fatty acids, and carbonaceous materials mixed with salts and minerals. But, our current knowledge regarding IN-active organics is still limited."

*RC*: - P7L38: Why did the authors use different units for PM mass loading here?
*AR*: We would like to offer both units used in Eqn (1), g L$^{-1}$, and what TEOM offers as a preset unit, µg m$^{-3}$.

*RC*: - P7L47: What is the significance of average estimated INP concentrations between different years?
*AR*: This is an interesting question. We now provide the visualized inter-annual variability of average estimated INP concentrations in Fig. 9.

[Figure]

**Figure 9**. OLLF INP concentrations. Time-series plot of TEOM mass concentration measured at the downwind side of OLLF-1 (a) and cumulative $n_{INP}$ estimated at $T$s of -15 °C, -20 °C, and -25 °C (b). In Panel a, inter-annual average mass concentrations of aerosol particles from OLLF (blue dashed line) and upwind (red dashed line) are shown (numbers adapted from **Table 8**). In Panel b, likewise, inter-annual average $n_{INP}$ estimated at -15, -20, and -25 °C (reported in **Table 8**) are also shown. Meteorological summer in Texas is used for the beginning and ending time stamps of each year.

Furthermore, the authors decided to discuss the inter-seasonal variations in estimated PM and INP concentrations using Table 8 in Sect. 3.5.

**Table 8.** Inter-annual and seasonal PM$_{10}$ mass concentrations from OLLF-1 as well as estimated $n_{INP}$.

| | PM$_{10}$ Mass Concentration (g L$^{-1}$) | | Estimated $n_{INP}(T)$ (L$^{-1}$) | | |
| | *OLLF | Upwind | $T$ = -15 °C | $T$ = -20 °C | $T$ = -25 °C |
|---|---|---|---|---|---|
| 2016 – 2017 | 1.8E-07 | 2.6E-08 | 20.7 | 127.5 | 2323.4 |

| | | | | | |
|---|---|---|---|---|---|
| Summer | 3.7E-07 | 5.2E-08 | 42.3 | 260.5 | 4747.7 |
| Fall | 1.6E-07 | 2.8E-08 | 18.1 | 111.7 | 2036.3 |
| Winter | 6.3E-08 | 1.5E-08 | 7.2 | 44.2 | 806.2 |
| Spring | 1.6E-07 | 2.1E-08 | 17.7 | 108.9 | 1985.5 |
| **2017 – 2018** | **4.8E-07** | **2.6E-08** | **54.6** | **336.4** | **6133.0** |
| Summer | 3.0E-07 | 2.3E-08 | 33.8 | 208.5 | 3801.1 |
| Fall | 3.1E-07 | 1.9E-08 | 35.4 | 218.2 | 3978.3 |
| Winter | 2.5E-07 | 1.3E-08 | 27.9 | 171.7 | 3129.6 |
| Spring | 9.2E-07 | 4.6E-08 | 104.1 | 641.3 | 11690.9 |
| **2018 – 2019** | **3.7E-07** | **1.7E-08** | **42.3** | **260.7** | **4752.5** |
| Summer | 4.9E-07 | 2.6E-08 | 55.6 | 342.3 | 6240.6 |
| Fall | 2.4E-07 | 7.9E-09 | 26.8 | 165.3 | 3013.0 |
| Winter | 1.5E-07 | 1.3E-08 | 17.0 | 104.8 | 1910.2 |
| Spring | 2.8E-07 | 1.6E-08 | 31.8 | 195.8 | 3570.0 |

*Upwind concentration is subtracted.

The associated text in Sect. 3.5 has been rephrased as, "In general, $PM_{10}$ mass concentrations from OLLF-1 (average ± standard errors) were high in meteorological summers ($3.9 \times 10^{-7} \pm 5.6 \times 10^{-8}$ g $L^{-1}$) and springs ($4.5 \times 10^{-7} \pm 2.4 \times 10^{-7}$ g $L^{-1}$) as compared to fall ($2.4 \times 10^{-7} \pm 4.4 \times 10^{-8}$ g $L^{-1}$) and winter ($1.5 \times 10^{-7} \pm 5.3 \times 10^{-8}$ g $L^{-1}$). A similar trend was found for the upwind $PM_{10}$ mass concentration: summer ($3.4 \times 10^{-8} \pm 9.0 \times 10^{-9}$ g $L^{-1}$) ≥ spring ($2.8 \times 10^{-8} \pm 9.3 \times 10^{-9}$ g $L^{-1}$) > fall ($1.8 \times 10^{-8} \pm 5.7 \times 10^{-9}$ g $L^{-1}$) ≥ winter ($1.4 \times 10^{-8} \pm 7.1 \times 10^{-10}$ g $L^{-1}$). But, the measured values at the upwind location are consistently an order magnitude lower than that from the downwind location.

On average, the estimated mean $n_{INP}$ values at -15, -20, and -25 °C in 2016 – 2019 were estimated as 46.8 (±25.3 seasonal standard deviation; same hereafter), 288.1 (± 156.1), and 5,250.9 (± 2,845.6) $L^{-1}$, respectively. In addition, the median $n_{INP}$ at -15, -20, and -25 °C in 2016 – 2019 were estimated as 14.7 (± 9.2), 90.9 (± 56.4), and 1,656.3 (± 1,028.1) $L^{-1}$, respectively. As our $n_{INP}$ is linearly scaled to mass concentration (Eqn. 1), estimated $n_{INP}$ showed a similar seasonal variability as seen in mass concentration. For instance, at -20 °C, the cumulative $n_{INP}$ averages for each meteorological season over three 2016 – 2019 were estimated as follows: spring (315.4 ± 164.9 $L^{-1}$) ≥ summer (270.4 ± 39.0 $L^{-1}$) > fall (165.1 ± 30.8 $L^{-1}$) ≥ winter (106.9 ± 36.8 $L^{-1}$). The observed high $n_{INP}$ values were expected for such a high $PM_{10}$ mass concentrations emitted from the cattle feedyard, which represent an important point source of agricultural aerosol particle emission. However, we reemphasize that the IN efficiency of OLLF aerosol particles is somehow similar to other agricultural aerosol particles found in previous studies as discussed in **Sect. 3.2** (**Fig. 8**)."

*RC*: - P7L490-50: Again, is there any reason for such high INP activity? Is there any previous paper on such efficient INP?
*AR*: Addressed above.

*RC*: - P7L52-53: Reads like introduction.
*AR*: This sentence is removed.

*RC*: - P8L4-35: Please rephrase and improve the writing quality.
*AR*: Done.

*RC*: - Please replace "lpm" with "LPM" in the paper and check unit usage throughout.
*AR*: Replaced.

*RC*: - Please check the space between the number and °C for consistency.
*AR*: Done.

*RC*: Fig. 1:
Panel (c): What does the green shading in panel (c) stand for?
*AR*: RH uncertainty – "*RH*s were determined with an accuracy of ± 5%, represented as green shaded area in (c), using the mean gas *T* and the mean water vapor concentration."

Why did the RH drop below water saturation in column (ii-iv), (vi), (viii), and (ix) during immersion freezing studies?
*AR*: This is a valid question. At these *T*s, ice crystals grow rather fast at an expense of available water vapor in the AIDA chamber, which causes a drop in RH. Nevertheless, droplets should be fully activated within first 100 seconds of each expansion until the peak RH value is reached (Fig. 7). Moreover, as seen in panels (d), the number concentration of particles with >20 μm $D_{ve}$ is not increasing after the RH peak, and all predominant ice formation occurs at or before the RH peak through immersion freezing. Lastly, as seen in Fig. 8, we made sure to only report our IN efficiency at *T*s higher than ~ -30 °C, corresponding to water saturated condition in the AIDA vessel. This point is now clarified in Sect. 3.2 (please refer to the track changes version manuscript).

Panel (d): The total particle concentration line is hard to notice.
*AR*: We have modified all d panels with the right y-axis assigned for total particle concentration.

[Figure]

**Figure 7.** Temporal profiles of the AIDA immersion freezing experiment [TXDUST01_07 (i), _08 (ii), _30 (iii), _12 (iv), _13 (v), _32 (vi), _3 (vii), _4 (viii), _16 (ix), _17 (x)]. Arrays of alphabetical panels represent the chamber gas *T* (solid line) and the chamber wall *T* (dashed line) (a), P in the AIDA chamber vessel (b), *RH* with respect to water (green line) and ice (blue line) (c), and aerosol particle concentration initially measured by the CPC (red solid line) as well as number concentration of > 20 μm $D_{ve}$ AIDA particles measured by a welas optical particle counter (blue line) (d). Horizontal numerical panels represent different sample types and AIDA experiments, including TXD01 (i) – (iii), TXD05 (iv) – (vi), TXD01H (vii –

viii), and TXD05H (ix – x). *RH*s were determined with an accuracy of ± 5%, represented as green shaded area in (c), using the mean gas *T* and the mean water vapor concentration.

This figure add little to the paper, should it go to SI?
*AR*: It is a valid suggestion. While we put minimum text associated with this figure, the authors believe that this figure represents important result of AIDA experiments. We would like to keep this in the main manuscript.

*RC*: Fig. 2: The font size is too small and unclear.
*AR*: The former Fig. 2 is now moved to SI Sect. S1 (Fig. S1). The authors believe that all texts are visible in Fig. S1.

P14L13-15: Please rephrase.
 *AR*: Rephrased.

P14L13: What is "CF-to-IF" ratio?
*AR*: counterflow to input flow ratio

This figure add little to the paper, should it go to SI?
*AR*: The authors concur. Now, it is moved to SI Sect. 1.

*RC:* Fig. 3: The font size is too small and unclear. Labels of panels a and b are missing.
*AR*: As discussed above, the authors decided to further analyze our data for spatial variability (Fig. 3) and seasonal variability (Figs. 4-6) to explain the observed data variation and completely reformulated Sect. 3.1. Please see the revised figures in the track changes manuscript.

*RC:* What does ns in the figure refer to? The authors mentioned ns,geo in P6L11, is there any relevance between the text and this figure? Why is the caption inconsistent with the figure? Is there any difference among ns,geo (STP), ns,geo, ns (STP), and (STP)?
*AR*: We now strictly limited our terminologies to $n_{s,geo}$ and $n_{s,geo}(T)$, which is used for the $n_{s,geo}$ as a function of temperature, throughout the manuscript. They are all computed for STP.

*RC*: Fig. 4: Please make it explicit that this figure is for -25°C.
*AR*: Good point. Done.

*RC*: What caused such large error of cumulative PM mass?
*AR*: This is a valid question. The errors in cumulative PM mass are on average 40.4%, derived from calibration of two DustTrak instruments against TEOM in a side-by-side position. This error mainly stems from variable emission flux at OLLF rather than a systematic bias of DustTrak (<10%; Wallace et al., 2011). Briefly, the aerosol particle emission potential at OLLF is associated with highly pulverized, nearly single-grained surface material that develops on OLLF pen surfaces as manure accumulates, dries, and is crushed by animal hoof action, as contrasted with the small to medium-sized "clods" from which the single-grained material ultimately develops. Bush *et al.* (2014) provide a more extensive account of fugitive-dust emission dynamics and their interaction with boundary-layer stability to create transient peaks in ground-level mass concentrations of fugitive dust. Again, as addressed above, it is important to note that the emission "flux" at OLLF is not spatially uniform (not even in a *piecewise* sense), and the emitting mechanism itself is not a property of a source area *per se* but of a unit of mobile livestock.  Granting the

primacy of hoof action as the decisive emissions mechanism as described above and in Bush *et al.* (2014), a more accurate representation of the source strength of an OLLF would be e. g. an agent-based model in which the source is the time-varying aggregation of many mobile point sources, as described by Auvermann (2003). We also shorten the new Fig. 5 caption to "Note the errors in cumulative PM mass are ± 40.4% as discussed in **Sect. 3.1.**"

*RC*: "The uncertainty in nINP and nm is ± 23.5%". However, the error bars of nINP in Fig. 4(a) seems not equal to this value. Also: the correlation coefficient and/or other parameters should be given in Fig. 4.
*AR*: Double checked – they are good.

*RC*: Fig. 5: The color of error bars is the same with the color of heated samples. Please make this figure clearer for readers.
*AR*: Modified.

*RC*: Why did the authors use different scales in the same figure?
*AR*: Now, we incorporated all in the same scale.

*RC*: The meaning of ns,geo is different from the definition in P6L11. Please clarify if these are different quantities, and explain why the terms and symbols are so confusing in the paper.
*AR*: Clarified in the manuscript, and it is now consistent.

*RC*: Please check the error bars in panel c. How come that the upper error bars are longer than the lower bars in a log-scale plot?
*AR*: Checked.

*RC*: Please check the labels of the figures for consistency.
*AR*: Checked.

*RC*: In Fig. 5 (a.ii), there is a significant difference between the filter and bulk samples at temperatures above -20 °C, but no such difference in Figure 5 (b.ii). What causes this difference?
*AR*: Explained in Sect. 3.2.

*RC*: Fig. 6: Is it worth wasting a figure and report just one set of DFPC data rather than include the data in Fig. 5?
*AR*: The referee has the point. We now merged a subset of DFPC data in Fig. 8.

*RC*: Table 2: Why did the author report "relative standard deviation of ±3%" instead of one standard deviation from the mean value?
*AR*: No special intension, we thought % would be more straightforward to the reader. We now report standard deviation instead of relative one.

*RC*: How does heating affect ATD sample density?
*AR*: The authors do not discuss anything about ATD here.

*RC*: Why did heating lead to increased density for TXD1?
*AR*: For the given uncertainty, TXD1 and TXD1H show similar ice nucleation efficiencies. So it is not conclusive

*RC*: Why didn't the author measure the density of a specific sample before and after heating?
*AR*: The authors measured the density of all bulk samples before and after heating as shown in Table 2 and discussed in Sect. 2.2. We wanted to make sure that physical properties, like the density, are not impacted by heating.

*RC*: Table 3: What does the number in Experiment ID mean? It's very hard to understand together with Fig. 1.
*AR*: Unique code used in AIDA. In Sect. 3.2, now we clarify this code as, "All lab data associated with this study were archived according to the AIDA experiment number (i.e., TXDUST01_number), and we share these IDs for other associated measurements (e.g., INSEKT)."

*RC*: Why is the size distribution so wide? What does the size distribution look like?
*AR*: We did not select the size. It is polydisperse in nature.
We show the particle surface area distribution of TXDUST01_07, _12, _3, and _16 as for snapshot examples of each sample type (TXD01, TXD05, TXD01H, and TXD05H, respectively) here.

[Figure]

Why didn't the author report particle number size distribution? What quantities do the letters refer to?
*AR*: We use surface ($S$) to estimate ice nucleation efficiencies, so we report only surface-based distribution.
$N_{total,0}$ = total number concentration of particles at the initial stage (t = 0) prior to expansion.

$S_{total,0}$ = total surface concentration of particles at the initial stage (t = 0) prior to expansion.
$M_{total,0}$ = total mass concentration of particles at the initial stage (t = 0) prior to expansion.
$D_{ve}$ = volume equivalent diameter.

*RC*: Table 4: Why do the authors put a table here with little description and discussion in text?
*AR*: Now, the discussion is extended based on the referee's suggestion (i.e., comparison to previous studies), and we would like to keep this table in this main manuscript.

*RC*: Table 7: Why were the residual projected sizes consistently larger than those of aerosols except for TXD05H? What do the authors think happened here?
*AR*: This was expected as the larger the particles, the more surface is available for ice nucleation to take place. The reason for the TXD05H exception is not known. Fig. S2 shows >10% reduction of K-rich particles (only 2% for TXD01H) so, loss of condensation active INP may play a role. But as this is not conclusive, we will limit the discussion regarding TXD05H.

**Other revisions:**

West Texas → the Texas Panhandle
The authors recently learned that ACP does not accept "West Texas" as a proper noun. As the place name needs to have clearly defined boundaries and be internationally known, we decided to adapt "the Texas Panhandle" and replaced it with all West Texas in this paper. At its first appearance in the manuscript, we define it as "the Texas Panhandle (northern most counties of Texas; also known as West Texas)".

Dr. Larissa Lacher has been added as a coauthor because she has visited the Texas Panhandle to in part support the field sampling activities at OLLFs and was involved in analyzing some INSEKT samples, which led to improve SI Sect. S3.

We have a new acknowledgements section and separated financial support statements.

[revised manuscript text omitted]

---

## Author Comment (AC3) · 22 Apr 2021

**Response to Referee #2**

First of all, the authors thank the referee for submitting helpful and meaningful comments, which lead to improvements and clarifications within the manuscript.

Below, we provide our point-by-point responses. For clarity and easy visualization, the referee's comments (**RC**) are shown from here on in black. The authors' responses (**AR**) are in blue color below each of the referee's statement. In addition to the responses to referees' comments, we further modified the manuscript to increase its clarity and readability. The summary of other changes is included at the end of this document. We introduce the revised materials in green color along/below each one of your response (otherwise directed to the Track Changes version manuscript). All references are available in the end of this AR document.
* * *
**RC**: This paper is focused on the potential of domesticated animal feeding facilities as sources of atmospheric ice nucleating particles (INPs). Assessing anthropogenic sources of INPs is an important area of research. Hence, this is an area of research suitable for publication in ACP. However, there are some significant problems with this paper. Most importantly, the paper is not well written. These authors, including the first author, have produced some excellent pieces of work in the past so I know they are capable of much better. I do not wish to spend a great deal of time going from line to line trying to edit the manuscript for them. Instead I will focus on several key areas, which I'll work through here:

**AR**: The authors appreciate these general remarks and diplomatic criticisms regarding our manuscript by referee #2. We hope that with the changes made in the current version of the manuscript, the overall structure and readability improved such the quality of this paper (please read the Track Changes version paper). Further, a consistency of terms and symbols has been checked. Below, we provide our point-by-point responses in hopes of our manuscript being considered for another review by the reviewer.

**RC**: 1. 'Feedlot': it may be obvious what this is to a farmer in the USA, but it is not obvious what this is to the wider community. I had to google the term to find out. An alternative title could be 'Cattle feeding facilities in the USA are a sources of:

**AR**: This is a valid question. It is customary for authors to refer to "open-air feedlots" (OAFs), "animal-feeding operations" (AFOs) or even "concentrated [or confined] animal-feeding operations" (CAFOs), but those terms are not specific enough to distinguish the particular sort of production system we have in view as the referee mentioned. In the revised paper, for clarity, we have adopted the term "open-lot livestock facility" (OLLF) to denote a particular type of animal-feeding operation in which *livestock* (as distinct from poultry) is raised in *outdoor confinement* (as distinct from partially or totally enclosed housing, and also as distinct from pasture/range/"free-range" production systems). Open-lot livestock facilities are common in semi-arid and arid climates because, as contrasted with the alternative production systems typical of wetter and more temperate climates, they (a) are an intensified form of livestock production, generating more marketable product per unit land area with less built infrastructure, (b) make use of the elevated evaporative demand to reduce or eliminate precipitation-generated wastewater that must be controlled under water-quality regulations, and (c) capitalize on the nocturnal cooling characteristic of semi-arid and desert climates to avoid major investments in (and operating costs associated with) ventilation systems while still reducing the incidence and duration of livestock heat stress under most conditions. The authors now clarify our OLLF term in the third paragraph of the revised introduction as follows;

"Agricultural land use is in excess of 50% of total U.S. land use according to the U.S. Department of Agriculture, and there are > 26,000 "open-lot livestock facilities" (OLLFs) in the U.S. (Drouillard, 2018).

The term OLLF is adapted to denote a particular type of animal-feeding operation, in which cattle livestock is raised in outdoor confinement, as distinct from partially or totally enclosed housing, and also as distinct from pasture or free-range production systems (Auvermann et al., 2004). OLLFs are common in semi-arid and arid climates. Contrasted with the alternative production systems typical of wetter and more temperate climates, they (1) are an intensified form of livestock production, generating more marketable product per unit land area with less built infrastructure, (2) make use of the elevated evaporative demand to reduce or eliminate precipitation-generated wastewater that must be controlled under water-quality regulations, and (3) capitalize on the nocturnal cooling characteristic of semi-arid and desert climates to avoid major investments in (and operating costs associated with) ventilation systems while still reducing the incidence and duration of livestock heat stress under most conditions."

*RC*: 2. The abstract needs to be rewritten. Tell the reader about the conclusions of your work, not the topics you cover without an indication of what the key results and conclusions are. E.g.: 'New data on the ice nucleation (IN) properties of agricultural dust at heterogeneous freezing temperatures (Ts > -29_C) were generated, providing statistical context.'; 'Overall, we successfully characterized physical, chemical, and biological properties of aerosol particles found at a cattle feedlot'; 'The relationship between these measured properties and atmospheric IN parameterization relevant to mixed-phase clouds is discussed.' (This is deleted.); 'Our INP parameterization and ICR characterization are meaningful for improved understanding of INP emission and cloud microphysical processes in the supermicron-particle laden region'.
*AR*: We have re-written and re-formulated our abstract based on inputs from the reviewer.

*RC*: 3. Intro: these paragraphs are far too long and confusing. Break up into topics and build a logical case for this study.
*AR*: The logic of this section has been improved by introducing (1) the climatic impact of ice-nucleating particles (INPs), (2) previous fertile-and-agricultural soil dust-derived INP studies (Suski et al., 2018; Conen et al., 2011; Hill et al., 2016; Steinke et al., 2016; 2020; Tobo et al., 2014; O'Sullivan et al., 2014 – more discussion along with our results in Sect. 3.6), and (3) potential significance of soil dust INPs in the U.S. as well as Texas (and the reasons) in the first four paragraphs.

RC: 4. P2 ln 45. 'Milling and grinding'. It is inappropriate to mill natural samples of material where the larger sizes are likely made of different materials to the finer aerosolisable fraction. If, for example, the largest grains are ice active and they are milled down to sizes of atmospheric relevance then the ice nucleating ability you measure is likely to be simply a product of the milling process. Milled natural dusts like these samples are therefore not a meaningful proxy for the dust that may be aerosolised from this source, unless there is some justification for treating the sample in this way.
*AR:* This is a valid question, and the authors have a justification for this sample preparation procedure, which have been used for previous studies in this region. Physically pulverizing the manure samples by milling simulates in important ways the primary emissions mechanism at play in and characteristic of OLLFs. Although wind scouring is certainly one emissions mechanism that is occasionally responsible for fugitive aerosol emissions from OLLFs, by far the most significant emissions mechanism is the pulverization and airborne resuspension by animal hooves of the dry, uncompacted, friable manure that accumulates on pen surfaces in an OLLF in the southern High Plains and similar climates. In fact, the first known, bench-scale simulation of the emissions mechanism characteristic of OLLFs (Razote et al., 2006) featured an ultra-low-velocity "wind tunnel" with a test section in which both vertical and horizontal hoof action generated the aerosol. Further, a later evaluation tool designed for OLLF managers to conduct self-assessments of fugitive-dust potential (Bush et al., 2014; see Table 1, p. 818) is predicated on the greater

emissions potential associated with highly pulverized, nearly single-grained manure that develops on OLLF pen surfaces as manure accumulates, dries, and is crushed by animal hoof action, as contrasted with the small to medium-sized "clods" from which the single-grained material ultimately develops. Bush et al. (2014) also provide a more extensive account of fugitive-dust emission dynamics and their interaction with boundary-layer stability to create transient peaks in ground-level mass concentrations of fugitive dust. The short justification with citations is now provided in Sect. 2.2. as, "Physically pulverizing the surface samples simulates the primary emission mechanism and characteristic of OLLFs (Razote et al., 2006; Bush et al., 2014; von Holdt et al., 2021)."

The authors appreciate this question by the referee, and we cautiously note that there might be a bias between the field-sampled and lab-generated aerosol which might cause a difference in the ice nucleation ability (e.g. referencing Boose et al., 2016 c, who states "Furthermore, we find that under certain conditions milling can lead to a decrease in the ice nucleation ability of polymineral samples due to the different hardness and cleavage of individual mineral phases causing an increase of minerals with low ice nucleation ability in the atmospherically relevant size fraction."

*RC*: 5. Ln 46: Dry heat tests: what precedent is there for 100 C being a suitable test for deactivation of INP proteins? Ideally we would be shown control experiments with a biological ice nucleator.
*AR*: Ice nucleation activity by bacteria (Morris et al., 2004; Christner et al., 2008), fungi (Humphreys et al., 2001) and lichens (Henderson-Begg, et al., 2009) has been shown to be heat-sensitive irreversibly at 100 °C or below. This point is now addressed in Sect. 3.6 (please see the track changes manuscript).

*RC*: 6. Ln 48: Wet heat tests: Clarify the procedure here. The normal practice is to place a sealed vial in a volume of boiling water. The way the text reads is that the sample itself was boiled for 20 minutes. If this was the case, then how was the loss of water from the sample accounted for?
*AR*: What we did for our wet-heating is the normal practice that the referee mentioned. The falcon tube was closed and no water was lost. The sample tube was immersed in boiling water (~100 °C) for 20 minutes. This temperature was chosen to denature proteinaceous INPs. Thus, the subtraction of heated $n_{INP}$ from non-heated $n_{INP}$ might represent their contribution to immersion freezing. This procedure is adapted from Schiebel (2019). Briefly, the aerosol particle suspension (3 mL) from a non-treated stock was first transferred to a sterile 50ml falcon tube. The screw-cap is closed, such that no water is lost. Then the tube is placed together with a precisely fitting styrofoam ring in a water-filled glass beaker. The styrofoam ring ensures that the tube is floating and that all of the aerosol suspension is below the water surface for best heat transfer. The beaker is covered with aluminum foil and is placed on a stirring hot plate to boil the water. The sample tube remains in the boiling water for 20 min.

*RC*: 7. P7. Section 3.3/ The first two paragraphs have little to do with the heading of this section.
*AR*: The authors agree. Moved to Sect. 2.4.

*RC*: 8. Section 3.4 and Table 5. Why is there a parameterisation for each sample? This seems excessive. It would be more useful to have a single parameterisation with an indication of variability.
*AR*: Offering a universal or representative single parameterization for agricultural INPs is not the scope of this work. As OLLF represents a point source of fresh livestock-generated dust, we expect that it would have different ice nucleation efficiency than aged/weathered dusts. Individual parameterizations are useful to analyze spectra by comparing $\Delta\log(n_{s,geo})/\Delta T$ values etc. We would like to keep all individual parameterizations. As now seen in Sect. 3.5, we used the Field_Median parameterization as an example (and representative) one for its T coverage up to -5 °C. The summary table of all parameterizations and associated text are now moved to SI Sect. S6 to increase the readability of the main manuscript and to

focus on the main scientific discussion on verifying that OLLF can be an ecosystem acting as a source of soil dust INPs. This point is now clarified in SI Sect. S6 (please see the track changes version).

*RC*: 9. P7, ln 34. 'this parameterization can be easily incorporated in many model platforms'. The authors need to be more specific how they envisage that this might happen. Do relevant models have this source of dust in them already with the emission already set up as an independent tracer? I suspect the answer is no. So, what else would we need to know in order to be able to represent this source of INP?
*AR*: The reviewer is right. We admit that it was not written in a right tone. As this statement is too ambitious, we removed this sentence. Besides, our main objective of this study is to verify that OLLF can be an ecosystem acting as a source of soil dust INPs but not providing a computationally unexpansive universal soil dust IN parameterization. For these reasons, we decided to omit this paragraph completely from the manuscript. In general, incorporating IN in any cloud/atmospheric models is a complex effort of dealing emission flux, aerosol dynamics, cloud microphysics, IN parameterization, and cloud macrophysics etc. (Zhang et al., 2018). Thereby, we understand that it is not reasonable to use the word like "easily" etc.

*RC*: 10. Ln 46. ' no notable difference after dry-heating was observed for both TXD01 and TXD05, representing an important negative result'. Why is this important? Is it significant that there is no effect of heating the sample dry to 100 C. It would only be significant if IN proteins are known to deactivate on heating to this temperature.
*AR*: This negative result agrees with our metagenomics analysis, where no known ice nucleation active bacteria were detected. We clarified our point by rephrasing this sentence to, "Thus, no notable difference after dry-heating was observed for both TXD01 and TXD05 (**Table 5**). This negative result is important because it agrees with our metagenomics analysis, where no known IN-active bacteria were detected."

*RC*: 11. P 8 section 3.7: State which ns parameterisation was used in this calculation
*AR*: "Field_Median" as stated in the text. We now state, "Due to the atmospheric relevance and $T$ coverage extending to -5 °C, we used a fit of ***Field_Median*** in **Table S3** to compute representative $n_{s,geo}$ relevant to OLLF."

*RC*: 12. P 8. Section 3.7: Equation 1 is only appropriate if a small fraction of INP at any one size is activated to ice. This equation does not take into account the number of dust particles. For example, if there were 10 dust particles per cm3, this parameterization might predict 1000 dust cm-3, which would clearly be nonsense. To predict INP concentrations using an ns (or similar) parameterisation, you either need to be able to prove that only a small fraction of particles at one size over the whole size range activate to ice or integrate over the size distribution.
*AR*: The reviewer is right. Niemand et al. (2012) infers that the usage of $n_s$ is valid for small percentages of IN active fraction (~1%). From the numbers of $N_{total,0}$ given in our Table 3 (total number concentration of particles at the initial stage prior to expansion), we know we examined on average ~ 200,000 $L^{-1}$ of aerosol particles in the immersion freezing mode in ADIA. Even assuming we evaluate INP up to 2,000 $L^{-1}$, our INP fraction is 1%. Thus, our $n_s$ parameterization is reasonable. Now, this point is clarified in Sect. 2.6 (please see the track changes version).

*RC*: 13. P8. Section 3.7: Why use ns. The data seems to spread out in ns substantially, but in nm, they collapse. Hence, nm seems to be a better way of doing this calculation.
*AR*: We have presented a comparison of our $n_s$ results to six different $n_s$ parameterizations from previous (yet recent) soil dust IN studies to relatively assess the IN efficiency of OLLF dust to them. Such a rigorous comparison is invaluable, and we would like to keep using $n_s$ instead of $n_m$. If we were to analyze ice nucleation active macromolecules (INM) (i.e., the biological protein complexes), the number of INM

scaled with the known amount of mass of specimen in a droplet would make sense as IN is triggered by biological component, which may have a mass of ~ 150 kDa, rather than on insoluble surface (Wex et al., 2015 and references therein). However, as we did not find any notable amount of known IN active microbiome in our OLLF samples, we believe that our usage of $n_s$ is valid. Therefore, in the study presented here, the ice nucleation ability will be expressed per unit surface of OLLF sample/particle.

*RC*: 14. Ln 50. 'which is three orders of magnitude higher than typical ambient INP concentration from continental sources'. This is a selective reading of the literature. The values are certainly high, but they are not 1000 times higher than recent literature values. For example, other studies also report high INP concs: Petters and Wright (2015) show values up to 1000 L-1 and O'Sullivan et al. (2018) report values approaching 100 L-1 and Suski et al. (2018) report values in excess of 100 L-1.

*AR*: Now, the revised manuscript discuss this point in Sect. 3.6. Comparison to previous soil dust IN studies. Please see our revised manuscript. We also provide a summary of how high OLLF INPs could be as compared to other solid dust INPs in Fig. 10.

[Figure]

Figure 10. Ambient INP concentrations of soil dusts and aerosol particles as a function of T. The red-shaded area represents the range of our field $n_{INP}$ values at 0.5 °C interval for -5 °C > T > -25 °C from this study (Fig. 4). The red open symbols are our estimated median (± standard deviation) at -15, -20, and -25 °C discussed in Sect. 3.5. Five reference data are adapted from O'Sullivan et al. (2014 Fig. 9; O14), Steinke et al. (2020 Fig. 3; S20), Tobo et al. (2014 Fig. 6b; T14), Suski et al. (2018 Fig. 1a-d; Su18), and Kanji et al. (2017 Fig. 1-10; K17). Note that we display the maximum and minimum at -15, -20, and -25 °C of K17 in comparison to our estimation.

*RC*: 15. P 9 ln 4. What are 'controlled-experiments'?
*AR*: We meant to say the temperature-controlled laboratory experiments. We now revised our conclusion as suggested by referee #2, and this sentence is removed.

*RC*: 16. Conclusions: I found this hard to follow. There is a lack of structure and several statements do not seem to follow on logically. E.g.: In ' The insignificance of dry heating was demonstrated with the increase of organics found for the ICR of dry-heated samples' the second statement does not follow on from the first.

*AR*: This section has been revised. We have rephrased the sentence that the referee pointed out to; "Even after dry heating treatment, the increase of organics fraction was found for the ICR of our OLLF samples."

**Other revisions:**

West Texas → the Texas Panhandle
The authors recently learned that ACP does not accept "West Texas" as a proper noun. As the place name needs to have clearly defined boundaries and be internationally known, we decided to adapt "the Texas Panhandle" and replaced it with all West Texas in this paper. At its first appearance in the manuscript, we define it as "the Texas Panhandle (northern most counties of Texas; also known as West Texas)".

Dr. Larissa Lacher has been added as a coauthor because she has visited the Texas Panhandle to in part support the field sampling activities at OLLFs and was involved in analyzing some INSEKT samples, which led to improve SI Sect. S3.

We have a new acknowledgements section and separated financial support statements.

References
  O'Sullivan, D., Adams, M. P., Tarn, M. D., Harrison, A. D., Vergara-Temprado, J., Porter, G. C. E., Holden, M. A., Sanchez-Marroquin, A., Carotenuto, F., Whale, T. F., McQuaid, J. B., Walshaw, R., Hedges, D. H. P., Burke, I. T., Cui, Z., and Murray, B. J.: Contributions of biogenic material to the atmospheric ice-nucleating particle population in North Western Europe, Scientific Reports, 8, 13821, 10.1038/s41598-018-31981-7, 2018.
  Petters, M. D., and Wright, T. P.: Revisiting ice nucleation from precipitation samples, Geophys. Res. Lett., 42, 8758-8766, doi:10.1002/2015GL065733, 2015.
  Suski, K. J., Hill, T. C. J., Levin, E. J. T., Miller, A., DeMott, P. J., and Kreidenweis, S. M.: Agricultural harvesting emissions of ice-nucleating particles, Atmos. Chem. Phys., 18, 13755-13771, 10.5194/acp-18-13755-2018, 2018.

**References**

- Auvermann, B. W., Hiranuma, N., Heflin, K., and Marek, G.: 2004. Open-path transmissometry for measurement of visibility impairment by fugitive emissions from livestock facilities, American Society of Agricultural Engineers, 04-4010, https://doi.org/10.13031/2013.17090, https://www.researchgate.net/publication/228888957 (last accessed on March 21, 2021), 2004.
- Boose, Y., Welti, A., Atkinson, J., Ramelli, F., Danielczok, A., Bingemer, H. G., Plötze, M., Sierau, B., Kanji, Z. A., and Lohmann, U.: Heterogeneous ice nucleation on dust particles sourced from nine deserts worldwide – Part 1: Immersion freezing, Atmos. Chem. Phys., 16, 15075–15095, 2016.
- Bush, J., Heflin, K. R., Marek, G. W., Bryant, T. C., and Auvermann, B. W.: Increasing stocking density reduces emissions of fugitive dust from cattle feedyards, Applied Engineering in Agriculture, 30, 815–824, 2014.
- Christner, B. C., Morris, C. E., Foreman, C. M., Cai, R., and Sands, D. C.: Ubiquity of biological ice nucleators in snowfall, Science, 319, 1214. https://doi.org/10.1126/science.1149757, 2008.
- Drouillard, J. S.: Current situation and future trends for beef production in the United States of America - A review, Asian-Australas J Anim Sci., 31, 1007–1016, 2018.
- Henderson-Begg, S. K., Hill, T., Thyrhaug, R., Khan, M., and Moffett, B. F.: Terrestrial and airborne non-bacterial ice nuclei, Atmosph. Sci. Lett., 10: 215–219, https://doi.org/10.1002/asl.241, 2009.

- Humphreys, T. L., Castrillo, L. A., and Lee, M. R.: Sensitivity of Partially Purified Ice Nucleation Activity of Fusarium acuminatum SRSF 616, Curr. Microbiol., 42, 330–338, https://doi.org/10.1007/s002840010225, 2001.
- Morris, C. E., Georgakopoulos, D. G., and Sands, D. C.: Ice nucleation active bacteria and their potential role in precipitation, J. Phys. IV France, 121, 87–103, https://doi.org/10.1051/jp4:2004121004, 2004.
- Niemand, M., Moehler, O., Vogel, B., Vogel, H., Hoose, C., Connolly, P., Klein, H., Bingemer, H., DeMott, P., Skrotzki, J., and Leisner, T.: Parameterization of immersion freezing on mineral dust particles: An application in a regional scale model, J. Atmos. Sci., 69, 3077–3092, https://doi.org/10.1175/JAS-D-11-0249.1, 2012.
- Razote, E., Maghirang, R., Predicala, B., Murphy, J. P., Auvermann, B. W., Harner, J. P., and Hargrove, W. L.: Laboratory evaluation of the dust-emission potential of cattle feedlot surfaces, Transactions of the ASABE, 49, 1117–1124, https://doi.org/10.13031/2013.21729, 2006.
- von Holdt, J. R. C., Eckardt, F. D., Baddock, M. C., Hipondoka, M. H. T., and Wiggs, G. F. S.: Influence of sampling approaches on physical and geochemical analysis of aeolian dust in source regions, Aeolian Research, 50, 100684, https://doi.org/10.1016/j.aeolia.2021.100684, 2021.
- Wex, H., Augustin-Bauditz, S., Boose, Y., Budke, C., Curtius, J., Diehl, K., Dreyer, A., Frank, F., Hartmann, S., Hiranuma, N., Jantsch, E., Kanji, Z. A., Kiselev, A., Koop, T., Möhler, O., Niedermeier, D., Nillius, B., Rösch, M., Rose, D., Schmidt, C., Steinke, I., and Stratmann, F.: Intercomparing different devices for the investigation of ice nucleating particles using Snomax[®] as test substance, Atmos. Chem. Phys., 15, 1463–1485, https://doi.org/10.5194/acp-15-1463-2015, 2015.
- Zhang, K., Rasch, P. J., Taylor, M. A., Wan, H., Leung, R., Ma, P.-L., Golaz, J.-C., Wolfe, J., Lin, W., Singh, B., Burrows, S., Yoon, J.-H., Wang, H., Qian, Y., Tang, Q., Caldwell, P., and Xie, S.: Impact of numerical choices on water conservation in the E3SM Atmosphere Model version 1 (EAMv1), Geosci. Model Dev., 11, 1971–1988, https://doi.org/10.5194/gmd-11-1971-2018, 2018.

---

## Author Response (AR2)

**Response to Referee #3**

The authors would like to express our sincere gratitude for the referee and helpful comments. Below, we provide our point-by-point responses. The referee's comments (RC) are shown from here on in black. The authors' responses (AR) are in blue below each of the referee's statements. We introduce the revised materials in green color along/below each one of your responses (otherwise directed to the Track Changes version manuscript). All references are available at the end of this AR document.

General comment:

RC: This is a valuable contribution to the ice nucleation community and it has a good level of novelty. This is the first time that I reviewed this manuscript and given that this was the second round of revisions I was expecting to see a more readable, clean, and concise document. Although the incorporated changes improved the manuscript, it cannot be accepted in its current version. I invite the authors to take into account the comment by original reviewer #2 "I do not wish to spend a great deal of time going from line to line trying to edit the manuscript for them".
AR: The authors appreciate these general remarks. We reconsidered previous referee #2 comments and revised our manuscript, especially the Materials and Methods section, accordingly. Below, the authors provide our point-by-point responses.

Major Comments:

RC: The authors wanted to cover too much: i) field (annual trends, seasonal trends, downwind vs, upwind); ii) field vs. lab; iii) lab (heat vs. no heat, super micron vs. sub micron, bulk vs. filter, ); iv) intercomparison between 4 different instruments; and v) a parametrization. Although it is not bad at all to be ambitious and to report several measurements/observations, the authors need to combine the large amount of data into something that is easy to read and to follow.
AR: The authors agree with the referee. We realized that the structure of the method section and presenting various things could be overwhelming. The authors also acknowledge that retaining the format from a previous version would cause additional confusion. Therefore, the authors decided to restructure the main article to present the most invaluable scientific outcomes and limit the amount of Supplemental Information (SI) for the sake of readability. Our decision is based on considering all four referees' comments.

The authors revised the manuscript to feature the **abundance of supermicron aerosol particles acting as feedlot ice-nucleating particles (INPs) from lab and field studies** and did the following modifications:

(1) Separating **Sects. 2 and 3** based on laboratory study (sub-section 1) and field investigation (sub-section 2) to explain methods, materials, and results for each sub-section independently.
(2) Moving sample descriptions to the Results and Discussion section (**Sect. 3.1.1**) and leaving only concise technique explanation in the Materials and Methods section (**Sect. 2.1.1**) to increase the readability of the manuscript in an organized manner.
(3) Moving the heat treatment data, outcomes, and discussion from the main manuscript into a single **SI Sect. S4**. Keeping it over different sections in methods and results impaired the overall readability in our previous version, and the authors believe that this modification resolves the readability issue.
(4) Removing all bulk sample discussions from this manuscript and focusing on the filter-collected aerosolized samples – So there will be no bulk vs. aerosolized sample discussion. In the end, the

bulk is not our main outcome. The revised paper focuses on lab vs. field, we believe that the aerosolized sample is more relevant to what is in the field than the bulk sample.

(5) Removing the ice residual composition discussion from the main manuscript. We do not have statistically valid aerosol particle composition data of our 'field' samples from this study. As the referee is concerned, we cannot conduct the comparison of laboratory and filed sample compositions, and the former Sect. 3.4 contributes little to the main text. The authors agree that the composition analysis is not the main focus of this manuscript, and decided to exclude this part from the manuscript.

(6) Moving the discussion regarding the estimated INP concentrations to **SI Sect. S5**.

In addition, the authors also changed the title of our manuscript to "Laboratory and field studies of ice-nucleating particles from open-lot livestock facilities in Texas", which better represents our research.

RC: Given that the samples were found to be heat insensitive, the IN properties are likely coming from the mineral components. Why the mineralogical composition is not reported. In lines 110-112 it is stated "However, our knowledge regarding what particular features of OLLF dust trigger immersion freezing at heterogeneous freezing temperatures (Ts; i.e., size vs. composition) is still lacking". The authors focused on the size of the particles but in terms of composition the minerals are completely ignored.

AR: The authors conclude that investigating minerals is not relevant to understand feedlot-derived INPs in this study because of the following reasons:

- Our previous work using Raman micro-spectroscopy revealed that ≈ 96% of ambient aerosol particles sampled at the downwind edge of an open-lot livestock facility (OLLF) contain brown or black carbon, hydrophobic humic acid, water-soluble organics, less soluble fatty acids, and carbonaceous materials mixed with salts and minerals (Hiranuma et al., 2011). We have mineralogical understanding, but we miss the bulk composition information (e.g., X-ray fluorescence and X-ray diffraction).

- Our chemical composition analysis of laboratory samples (**SI Sect. S1**) indicates that our samples are exclusively organic in nature in terms of aerosol composition.

- Recently, organic acids (i.e., long-chain fatty acids) and heat-stable organics were found to act as efficient INPs (DeMott et al., 2018; Perkins et al., 2020). Thus, identifying heat-stable organic compounds and studying their physicochemical properties may be key to understand the properties of OLLF INPs.

- A comparison of our field ice nucleation active-surface site density ($n_{s,geo}$) data to ice nucleation (IN)-active minerals (e.g., K-feldspar, quartz) does not support the inclusion of IN-active minerals. Some of our field samples (from any season) contain more efficient INPs than K-feldspar at temperatures above approximately -15 °C, as seen in Figure 1 on the next page. The $n_{s,geo}$ spectra of our field data generally have gentler slopes than K-feldspar. We note that the immersion freezing efficiency of K-feldspar is higher than quartz (Atkinson et al., 2013; Harrison et al., 2019).

[Figure]

Figure 1. Comparison of our field $n_{s,geo}$ data (adapted from Fig. 8 in our revised manuscript) to K-feldspar. The $n_{s,geo}$ values of K-feldspar were derived using the laser diffraction-based surface-to-mass ratio, 0.89 m$^{-2}$ g$^{-1}$, and Brunauer-Emmett-Teller (BET) specific surface area, 3.2 m$^{-2}$ g$^{-1}$, reported in Atkinson et al. (2013).

In addition, we could not find any notable inclusions of known IN-active microbiomes in both laboratory and field samples (now discussed in **Sect. 3.3.1**). While we cannot rule out the possibility of IN from TXD01 and TXD05 samples triggered by biological INPs, our current results do not support it. Certainly, our study cannot conclude what particular features of OLLF dust trigger immersion freezing at heterogeneous freezing temperatures. However, this deficit is a good motivation to investigate OLLF-derived ice crystal residual samples in more detail in the future.

RC: This is the second round of revisions and the manuscript still needs to be edited to improve its readability and the language. I was expecting to see a cleaner version.
AR: The language is re-checked by an English native speaker and an external editorial service provider. Please see the track changed materials. With a revised paper structure, the authors believe that the readability is improved.

RC: Lines 367-368 it is mentioned that "These results imply the following: (1) ambient meteorological conditions, as summarized in Table 1, might not be determining factors for nINP for our study sites". This refers to RH, T and P but not to wind speed, one of the most important meteorological variables to resuspend dust particles. Why is wind speed and wind direction not reported?
AR: Wind direction was not reported because we arranged our sampling locations (i.e., downwind and upwind sites) according to the observed wind direction. For instance, when south wind prevailed (90° < wind direction < 270°), we used the Northern site as the downwind site. Likewise, the Southern site was used as the downwind site while the north wind was dominant (270° < wind direction < 90°). We include this point in the revised **Sect. 2.2.1**. The authors provide our observed wind properties, which include wind speed and direction, during our sampling activities in Table 1 on the next page.

Table 1. Average wind speed and direction (± standard deviation) for individual sampling activities.

| Year | Date | Location | Start Time (Local) | End Time (Local) | Average Wind Speed ± Standard Deviation (mile hr$^{-1}$) | | | Average Wind Direction ± Standard Deviation (degree) | | |
|------|------|----------|-----------|---------|------|---|-----|-------|---|------|
| 2019 | 20190715 | OLLF-1 | 18:45:00 | 22:05:00 | 3.6 | ± | 1.3 | 157.9 | ± | 13.9 |
| | 20190716 | OLLF-2 | 18:45:00 | 20:29:00 | 10.6 | ± | 1.7 | 186.4 | ± | 4.3 |
| | 20190724 | OLLF-3 | 19:24:00 | 20:34:00 | 10.1 | ± | 1.3 | 147.5 | ± | 6.6 |
| | 20190226 | OLLF-1 | 16:08:00 | 19:09:00 | 11.2 | ± | 4.3 | 207.9 | ± | 13.2 |
| | 20190328 | OLLF-2 | 16:26:00 | 20:52:00 | 8.7 | ± | 3.3 | 217.2 | ± | 6.7 |
| | 20190420 | OLLF-3 | 17:05:00 | 21:05:00 | 10.2 | ± | 2.9 | 197.2 | ± | 19.1 |
| | 20190116 | OLLF-1 | 16:03:00 | 19:33:00 | 16.6 | ± | 2.8 | 256.0 | ± | 6.8 |
| | 20190117 | OLLF-2 | 15:48:00 | 19:30:00 | 8.7 | ± | 1.8 | 188.3 | ± | 11.6 |
| | 20190118 | OLLF-3 | 15:40:00 | 18:40:00 | 23.3 | ± | 2.5 | 319.4 | ± | 33.1 |
| 2018 | 20180722 | OLLF-1 | 18:42:00 | 22:39:00 | 5.7 | ± | 1.6 | 170.7 | ± | 11.0 |
| | 20180723 | OLLF-2 | 18:42:00 | 22:17:00 | 5.1 | ± | 3.9 | 83.6 | ± | 21.1 |
| | 20180724 | OLLF-3 | 18:20:00 | 22:13:00 | 7.9 | ± | 1.9 | 136.6 | ± | 12.0 |
| | 20180416 | OLLF-4 | 4:53:30 | 8:06:40 | 12.1 | ± | 4.0 | 216.2 | ± | 8.3 |
| 2017 | 20170709 | OLLF-1 | 19:32:45 | 22:26:00 | 9.3 | ± | 2.9 | 160.5 | ± | 10.1 |
| | 20170710 | OLLF-2 | 18:06:00 | 22:06:30 | 10.3 | ± | 3.0 | 183.8 | ± | 9.0 |
| | 20170711 | OLLF-3 | 18:28:00 | 22:08:00 | 6.4 | ± | 1.7 | 172.0 | ± | 10.9 |

Resuspension of feedlot surface materials, the so-called hoof action, is not wind-driven. Cattle movement and hoof action are the decisive emissions mechanism of feedlot dust when the air is dry and hot as described in Auvermann (2001) and Bush et al. (2014). The authors performed linear regression analysis for wind speed vs. particulate matter (PM) concentration, and the resulting Pearson correlation coefficient (*r*) was -0.32 (see Table 2 on the next page). Concerning highly variable concentration, the authors also examined the relationship between wind speed and cumulative PM mass per unit time, and the resulting *r* was -0.35. Since these negative coefficients indicate an inverse relationship between wind speed and PM, the authors decided not to report it. Nevertheless, we now clarify this point in the revised **Sect. 3.2.2** and report both wind speed and direction data activities in **Table 7**.

Table 2. Summary of wind speed, average PM mass concentration, and cumulative PM mass for individual sampling activities.

| Year | Date | Location | Start Time (Local) | End Time (Local) | Wind Speed (mile hr$^{-1}$) | Average PM Mass Concentration (mg m$^{-3}$) | Cumulative PM Mass (µg hr$^{-1}$) |
|------|------|----------|-----------|----------|------|------|-------|
| 2019 | 20190715 | OLLF-1 | 18:45:00 | 22:05:00 | 3.63 | 0.21 | 50.46 |
|  | 20190716 | OLLF-2 | 18:45:00 | 20:29:00 | 10.58 | 0.09 | 24.17 |
|  | 20190724 | OLLF-3 | 19:24:00 | 20:34:00 | 10.07 | 0.33 | 90.00 |
|  | 20190226 | OLLF-1 | 16:08:00 | 19:09:00 | 11.24 | 0.13 | 18.96 |
|  | 20190328 | OLLF-2 | 16:26:00 | 20:52:00 | 8.66 | 0.15 | 46.13 |
|  | 20190420 | OLLF-3 | 17:05:00 | 21:05:00 | 10.22 | 0.05 | 8.63 |
|  | 20190116 | OLLF-1 | 16:03:00 | 19:33:00 | 16.63 | 0.01 | 3.43 |
|  | 20190117 | OLLF-2 | 15:48:00 | 19:30:00 | 8.74 | 0.05 | 11.22 |
|  | 20190118 | OLLF-3 | 15:40:00 | 18:40:00 | 23.30 | 0.35 | 83.93 |
| 2018 | 20180722 | OLLF-1 | 18:42:00 | 22:39:00 | 5.71 | 0.82 | 324.30 |
|  | 20180723 | OLLF-2 | 18:42:00 | 22:17:00 | 5.12 | 2.48 | 814.30 |
|  | 20180724 | OLLF-3 | 18:20:00 | 22:13:00 | 7.88 | 0.22 | 86.03 |
|  | 20180416 | OLLF-4 | 4:53:30 | 8:06:40 | 12.13 | 0.03 | 12.08 |
| 2017 | 20170709 | OLLF-1 | 19:32:45 | 22:26:00 | 9.26 | 0.49 | 154.22 |
|  | 20170710 | OLLF-2 | 18:06:00 | 22:06:30 | 10.31 | 0.18 | 56.51 |
|  | 20170711 | OLLF-3 | 18:28:00 | 22:08:00 | 6.42 | 0.15 | 46.77 |

RC: Lines 417-418 and Line 524: "For each sample, the spectra nearly overlap each other at T ~ -25 °C, verifying their comparability and complementing features." and "Upon confirmation of the comparability between field and lab ns,geo values". Why is this overlap only true at -25C and not at other temperatures? obtaining similar results at this temperature is enough to conclude that the systems are really comparable. Is it not the overlap at -25C suspicious? Why should they be comparable? In the field, aerosol particles are naturally aerosolized and this resuspension from the ground can favor specific aerosol sizes. On the other hand, the aerosolization in the laboratory is mechanical and the particles sizes, and hence, their composition may significantly differ from those found in natural environments.

AR: The authors understand the referee's concern. The authors intended to point out (1) our dynamic filter processing chamber (DFPC)-derived $n_{s,geo}$ values in **Fig. 4** agreed reasonably well with the ice nucleation spectrometer of the Karlsruhe Institute of Technology (INSEKT) results at the measured temperatures within our error ranges, and (2) both aerosol interaction and dynamics in the atmosphere (AIDA) and INSEKT show reasonably comparable results in the overlapping temperatures around -25 °C. These two ambiguous statements are now removed, and the potential source of discrepancy between laboratory and field results is now discussed in the revised **Sect. 3.3.1**.

RC: Lines 438-441: The authors need to show that the composition and particle size distribution of ambient and Laboratory particles are comparable.

AR: Upon the request, the authors conducted energy dispersive X-ray spectroscopy (EDX) on particles in the suspended field sample, collected from OLLF-1 on July 22, 2018. We used an electron microscope (JEOL, JSM-6010LA) equipped with an EDX function. We have looked at a total of 56 particles on an aluminum substrate. All particles had an area equivalent diameter smaller than 6.44 µm, which is the largest aerosol particle size found in the AIDA chamber (see **Table 1**). Then, we qualitatively assessed EDX signals of organic (C, N, O), salt-rich (Na, Mg, K, P), mineral-rich (Si, Ca), and others. We excluded a background signal of aluminum from a substrate. We detected carbon in all particles exclusively with the

inclusion of minerals for 20% of examined particles. Two representative electron microscopy images and EDX spectra are shown below in Fig. 2.

The observed predominance of carbonaceous particles is consistent with our previous field sample analysis from the same OLLF (Hiranuma et al., 2011). We also echo that our single particle mass spectrometry composition analysis of laboratory samples (**SI Sect. S1**) indicates that our samples are exclusively organic in terms of aerosol composition.

Note that, as stated in the Materials and Methods section, our field sample was consumed for immersion freezing analysis within a day after sampling concerning sensitivity to storage time. Thus, our composition result may not reflect the composition analyzed for the West Texas cryogenic refrigerator applied to freezing test system (WT-CRAFT) for immersion freezing. Moreover, our EDX analysis on a small number of particles from a single sample cannot provide any statistically valid conclusion and size distribution data. Concerning these deficits and the referee's comment, the authors decided to omit the discussion of particle composition from the main manuscript.

[Figure]

Figure 2. Typical electron microscopy images and EDX spectra of (a) organic dominant particle (ID# 12) and (b) mineral including particle found in the OLLF-1 sample (ID #1).

As demonstrated in our previous study, the surface area distribution of ambient OLLF dust peaks in mode diameter at ≤ 10 μm (Hiranuma et al., 2011). This mode diameter is larger than surface-derived samples aerosolized and examined in the AIDA chamber. However, it is cautiously noted that the ambient OLLF dust size distribution is not spatially uniform, and the emitting mechanism itself is not controllable as it highly depends on a unit of mobile livestock. Granting the primacy of hoof action as the decisive emissions mechanism of OLLF dust as described in Bush et al. (2014), a more controlled laboratory experiment has been desired to characterize IN ability of OLLF soil dust.

Minor Comments:

RC: The Abstract is extremely long. The authors need to shorten it focusing on the main results only. Similar to the Abstract, several parts in the Results section (and along the manuscript) are repetitive, longer than needed and not very concrete. The manuscript needs to be further edited to improve its readability.

AR: The abstract is revised, and the text count was reduced from 601 words (3952 characters) to 384 words (2600 characters). All repetitive parts are removed from the manuscript. The revised manuscript focus on the **abundance of supermicron aerosol particles acting as feedlot INPs from lab and field studies** in a concise and readable manner. Below is the revised abstract.

"In this work, an abundance of ice-nucleating particles (INPs) from livestock facilities was studied through laboratory measurements from cloud simulation chamber experiments and field investigation in the Texas Panhandle. Surface materials from two livestock facilities, one in the Texas Panhandle and another from McGregor, Texas, were selected as dust proxies for laboratory analyses. These two samples possessed different chemical and biological properties. A combination of aerosol interaction and dynamics in the atmosphere (AIDA) measurements and offline ice spectrometry was used to assess the immersion freezing mode ice nucleation ability and efficiency of these proxy samples at temperatures above -29 °C. A dynamic filter processing chamber was also used to complement the freezing efficiencies of submicron and supermicron particles collected from the AIDA chamber. For the field survey, periodic ambient particle sampling took place at four commercial livestock facilities from July 2017 to July 2019. INP concentrations of collected particles were measured using an offline freezing test system, and the data were acquired for temperatures between -5 °C and -25 °C.

Our AIDA laboratory results showed that the freezing spectra of two livestock dust proxies exhibited higher freezing efficiency than previously studied soil dust samples at temperatures below -25 °C. Despite their differences in composition, the freezing efficiencies of both proxy livestock dust samples were comparable to each other. Our dynamic filter processing chamber results showed on average approximately 50% supermicron size dominance in the INPs of both dust proxies. Thus, our laboratory findings suggest the importance of particle size in immersion freezing for these samples, and that the size might be a more important factor for immersion freezing of livestock dust than the composition. From a three-year field survey, we measured a high concentration of ambient INPs of 1,171.6 ± 691.6 $L^{-1}$ (average ± standard error) at -25 °C for aerosol particles collected at the downwind edges of livestock facilities. An obvious seasonal variation in INP concentration, peaking in summer, was observed with the maximum at the same temperature exceeding 10,000 $L^{-1}$ on July 23, 2018. The observed high INP concentrations suggest that a livestock facility is a substantial source of INPs. The INP concentration values from our field survey showed a strong correlation with measured particulate matter mass concentration, which supports the importance of size in ice nucleation of particles from livestock facilities."

RC: Avoid self citations. From my personal point of view this is excessive.

AR: By limiting our research focus as addressed above, self-citation has been reduced. However, the authors keep all essential, meaningful citations as-is.

RC: Move sections 3.3 and 3.4 to SI as they contribute little to the main text.

AR: The authors moved some technical details of metagenomics analysis to **SI Sect. S2** and discuss it in the revised **Sect. 3.1.4** concisely. The authors agree that the former Sect. 3.4 (ice residual analysis) contributes little to the overall outcome of this study. We decided to omit the discussion of ice residual from this manuscript. More sample analyses towards understanding molecular-level properties of soil dust ice crystal residual will be conducted for a separate publication.

RC: The use of "T" and "Ts" instead of "temperature" and "temperatures" add too much noise to the manuscript.

AR: All T and Ts are written out as temperature or temperatures in both the main manuscript and SI. The use of the abbreviation, $T$, is limited to the part of parameter expression (e.g., $n_{INP}(T)$ = INP concentration per unit standard air volume as a function of temperature), which is explained in the text.

RC: Lines 66-67: "(i.e., the freezing propensity of INP immersed in supercooled water)". Who freezes the INP or the droplet?

AR: It is the aerosol particle(s) immersed in a droplet. To increase the clarity, we rephrased the definition of immersion freezing as "the freezing of aerosol particle(s) immersed in a supercooled droplet".

RC: Lines 95-96: How about convection?

AR: Convection can certainly play a role, and it seems self-evident given the differential heating of a feedlot surface and surrounding vegetation. However, none of the authors, including experts in the OLLF research and agricultural engineering, are aware of any previous study showing it is the cause of vertical transport of OLLF dust.

In **SI Sect. S5**, the authors also added a note of "We note that our estimation of $n_{INP}$ is limited at the source location. Further understanding of OLLF-derived INPs in the atmosphere will require future research in the dust generation mechanisms in association with local dynamics and thermodynamics, vertical distribution of OLLF dust, and their fate in the atmosphere".

RC: Line 99: "where a convective cloud and updraft system persists". What does it mean? that you have such a system 365 days per year?

AR: We admit that "persistent" is not the right word. We meant to say that we frequently observe such systems in this region. We decided to rephrase this sentence to "Convection and updraft system may also help the vertical transport of aerosol particles in the Southern High Plains region (Li et al., 2017).".

RC: Lines 115-118: This is not the right place. Please add this text earlier.

AR: Moved to the 4th paragraph in the Introduction section, where we first mention "manure".

RC: Section 2.2. This Section is not easy to follow. It is longer than needed and it has to be rewritten to be more concise and clear.

AR: We moved sample analysis outcomes to the Results and Discussion section and leaving only a concise technical description in the Materials and Methods section. To improve the overall conciseness and readability of the methods section, the authors reorganized the section based on (2.1) laboratory study and (2.2) field investigation.

RC: Lines 154-155: Add the depth and how were they collected?

AR: We have clarified this information in the revised **Sect. 2.1.1**:

"Soil samples were collected on September 20, 2017. All samples were scooped from the loose dry surface layer of the pens (< 5 cm). Typically, the pen surface layer only extends to a depth of about 5 cm, which represents the depth of hoof penetration into the pen surface (Guo et al., 2011). This surface layer is rich in loose manure, which is a major source of ambient OLLF dust (Bush et al., 2014; von Holdt et al., 2021). All samples were ground and sieved for grain size < 75 μm. They were kept in chemically inert containers at room temperature until analyzed".

RC: Line 186: How about wind speed?

AR: Addressed above.

RC: Line 345: "Even assuming we evaluate INP up to 2,000 L-1, our INP fraction is 1%. Thus, our ns parameterization is reasonable." I don't get the message here.

AR: We reclarified this in the revised **Sect. 2.1.3**. The INSEKT system typically measures INP counts up to several hundred. For this study, the highest INP concentration measured by INSEKT was 135.9 INP $L^{-1}$ (95% confidence intervals, CI95% = 80.1 – 198.5 $L^{-1}$) from the TXDUST01_08 experiment. As seen in **Table 1**, for this particular experiment, the filter sampling activity for INSEKT was conducted with an aerosol concentration of 266.3 x $10^3$ $L^{-1}$. This simply translates to INP fraction of < 1%, which satisfies the prerequisite of $n_s$ application addressed in Niemand et al. (2102).

RC: Line2 354, 374 and 375: "cumulative mass", "cumulative PM mass", and "aerosol particle mass". What does it mean? In methods two different systems to measure PM10 were described. Do the authors refer to PM10?

AR: Discussed above – please see Table 2 in this author response document.

RC: Lines 363-364: "However, because the measured nINP is low at high T, the CI95% error of nINP,upwind at around -15 °C is relatively large as compared to that at a lower T (Fig. 3a)". I do not get it.

AR: For clarity, we rephrased it to "… At this temperature, the $n_{INP,upwind}$ (CI95%) error in a log scale spectrum is relatively large as compared to the lower temperature region, and the difference between $n_{INP,downwind}$ and $n_{INP,upwind}$ is not conclusive beyond the uncertainty around -15 °C".

RC: Line 366: "on local meteorological conditions". What was the typical wind speed?

AR: The measured wind speed was on average ≈ 10 miles per hour (min to max = 3.6 to 23.3 miles per hour). As explained above, the wind seems not to have much to do with the resuspension of feedlot surface materials, but certainly could contribute to instantaneous spikes of $PM_{10}$ as our field sampling activities were carried out in the proximity of livestock pens (**Sects. 2.2.2 and 3.2.1**).

RC: Line 367: "short episode of soil dust". What do the authors mean? Resuspended dust?

AR: Yes. We replaced it with "resuspended OLLF soil dust".

RC: Lines 388.389: "This motivates the need for further characterization of our OLLF samples in a controlled-lab setting in order to identify what particulate size population (i.e., supermicron vs. submicron) and other properties trigger their". This cannot be evaluated in the field? should not just simply change the cut-off of the filter sampling PM1.0 vs. PM10?

AR: It could have been. But we did not have proper apparatus during our field investigation and, thereby, did not and perform $PM_1$ and $PM_{10}$ sampling in our fields. Nevertheless, we show there is a notable correlation between INP and $PM_{10}$ based on our 2017 – 2019 field study, which indicates the importance of large supermicron aerosol particles as INPs. This result supports the DFPC characterization of our OLLF samples in a controlled-lab setting to identify what particulate size population (i.e., supermicron vs. submicron). The onsite measurements of size-segregated INPs with a combination of a size-selecting impactor inlet and an online INP monitor will be indeed meaningful to add insights on the importance of large INPs. We include this point in our conclusion.

RC: Lines 399-410: This belongs to methods. Most of this information is already known from previous experiments in the AIDA, and therefore, their contribution to the manuscript is little and should not be in this section.

AR: The authors would like to keep this information here. We consider this as specific 'results' during our TXDUST01 campaign. We have changed the sub-section tile to "**3.1.2. AIDA measurements and freezing efficiencies of surface materials**".

RC: Lines 420-421: "the INSEKT results suggest that the bulk TXD01 sample is more active than filter-collected samples". Is this not expected as the bulk likely contain larger particles?

AR: The authors are not sure if it could be expected or not. Previously, Boose et al. (2016) studied immersion freezing abilities of diverse natural dust samples from nine desert regions around the globe (4 airborne and 11 sieved/milled surface samples) and found that the surface-collected samples tend to contain more efficient INPs than the airborne samples. The authors suggested that mineralogy may play a significant role to explain the observed difference. On the other hand, Kaufmann et al. (2016) found a similar freezing behavior of multiple surface dust samples despite the variation in mineralogy. Both studies noted the necessity of investigating non-mineral compositions (i.e., biological and other organics). While our laboratory and field samples are different in nature, our organic predominant samples show a reduction in IN efficiency for surface-collected samples compared to airborne field samples. The observed offset motivates further research in organic INPs. This point is now discussed in the revised **Sect. 3.3.1**. Note that we omitted the discussion regarding bulk vs. aerosolized but kept the immersion freezing efficiency of laboratory vs. field sample, scaled to the aerosol particle surface area (so the size factor is incorporated).

RC: Lines 424-425: "the lab-derived immersion spectra of both surface materials are reasonably comparable to the minimum – maximum boundaries of our field ns,geo spectra for T > -25 °C". If it is the same samples analyzed by different setups why should they show the large variability observed on the field samples?

AR: The authors found this part was misleading. We have rephrased it to:

"The immersion spectra of both surface materials are located towards the minimum boundaries of our field $n_{s,geo}$ spectra for temperature > -25 °C. While the variability of $n_{s,geo}$ at a single temperature could vary by several orders of magnitude for our field data, smaller variations are found for both lab results, implying different properties of our lab and field samples. The difference between our laboratory results and field data is discussed in **Sect. 3.3.1** in more detail".

An important caveat is that we could not find any notable inclusions of known IN-active microbiomes in both sample subsets. While we cannot rule out the possibility of IN from our field and laboratory samples triggered by biological INPs, our current results do not support it. The authors think that identifying heat-stable organic compounds and studying their physicochemical properties may be key to understand the properties of OLLF INPs. Our chemical composition analysis of laboratory samples (**SI Sect. S1**) indicates that they are exclusively organic in nature in terms of aerosol composition. Further, airborne particles collected in OLLFs are generally known to include substantial amounts of organic materials. For example, our previous work using Raman micro-spectroscopy revealed that ≈ 96% of ambient aerosol particles sampled at the downwind edge of an OLLF contain brown or black carbon, hydrophobic humic acid, water-soluble organics, less soluble fatty acids, and carbonaceous materials mixed with salts and minerals (Hiranuma et al., 2011). Recently, organic acids (i.e., long-chain fatty acids) and heat-stable organics were found to act as efficient INPs (DeMott et al., 2018; Perkins et al., 2020). However, our knowledge regarding what particular organics from OLLFs trigger immersion freezing at heterogeneous freezing temperatures is still lacking. A more detailed follow-up study to investigate molecular compositions of OLLF organics in ice crystal residuals may be necessary to provide an answer for it. This discussion is now given in the revised **Sect. 3.3.1**.

RC: Lines 424-426: While the variability of ns,geo at a single T could vary several orders of magnitude, similar variations are found for both lab and field results, implying the similarity of freezing efficiencies of our lab and field samples". This is expected for field samples as they are not identical and may have different composition, but in the laboratory, the instruments collected exactly the same samples.

AR: Addressed above – RE: RC: Lines 417-418 and Line 524. As shown in **Fig. 4**, the laboratory-measured $n_{s,geo}$ values are in agreement for the overlapping temperatures as all measurements are made using the same aerosol particles in the AIDA chamber. An offset between AIDA and INSEKT at around -22 °C may derive from the online vs. offline instruments issue, which is previously reported by some of the authors (Hiranuma et al., 2015). This is beyond the scope of the current study, and we would like to avoid extensive discussion about this issue.

[Figure]

**Figure 4**. IN-active surface-site density, $n_{s,geo}$, of surface materials, TXD01 (a) andTXD05 (b), was assessed by AIDA, INSEKT, and DFPC (total aerosol particles) as a function of temperature. Six reference $n_{s,geo}$ curves for fertile and agricultural soil dust (FASD) and desert dust are adapted from O'Sullivan et al. (2014; O14), Steinke et al. (2016; S16), Steinke et al. (2020; S20), Ullrich et al. (2017; U17), and Tobo et al. (2014; T14). The grey-shaded area represents the range of our field $n_{s,geo}$ values at 0.5 °C interval for -5 °C > temperature > -25 °C (**Fig. 8**).

RC: Lines 427-430: "there is a difference in the INP abundance between bulk (< 75 µm-sieved) and aerosolized/filtered-samples for TXD01 ($\lesssim$ 6.5 µm; Table 3) presumably due to different properties in particles of these two size subsets (6.5 – 75 µm and $\lesssim$ 6.5 µm) and/or different amount of IN-active soil organic matter". Again, is this not expected?
AR: Addressed above. The discussion regarding bulk vs. aerosolized samples has been excluded.

RC: Line 431: "be more representative of atmospherically relevant dust". Based on what? This needs to be clearly discussed.
AR: The authors agree that it is confusing. We excluded this ambiguous statement and rephrazed the sentence to:
"Additionally, the similarity of our lab results between TXD01 and TXD05 suggests that different physicochemical properties found for our samples may not impact their INP propensities".

RC: Line 433: "This comparability suggests that freezing ability is similar for condensation and immersion for our surface samples". I am not sure if such a strong conclusion can be said from just 2 data points from the DFPC.

AR: The DFPC measurements were carried out within the optimal operating conditions of the DFPC chamber. We understand the referee's concern. We have excluded this sentence from the Conclusions section and rephrased this part as:

"Moreover, the importance of large aerosol particles on immersion freezing was verified in our AIDA-based laboratory study. The DFPC offline freezing instrument assessed IN abilities of OLLF dust surrogates with $PM_1$ and total (> $PM_1$) size fractions. Our assessment revealed that on average ≈ 50% of OLLF $n_{INP}$ derived from supermicron aerosol particle population in the assessed temperature range between -18 and -22 °C. Thus, our laboratory study showed the potential importance of supermicron aerosol particles from OLLFs as INPs. While our metagenomics analysis does not support the presence of known IN-active microbiomes, more research should be directed to reveal the compositional identities and associated IN abilities of various other animal feeding facility samples".

We have moved the associated discussion to the Results and Discussion section (**Sect. 3.1.3**):
"Besides, several unique characteristics of OLLF INPs were disclosed. For instance, comparability of results from our condensation freezing instrument (DFPC) and immersion freezing assay (INSEKT) was found for both sample types at the overlapped temperatures (18°C and -22°C). A similar observation was previously made for kaolinite particles in Wex et al.(2014). However, as the examined temperatures in our study are limited, the observed equivalence between immersion and condensation freezing for our surface OLLF samples should be cautiously interpreted and may not be conclusive".

RC: Line 569: "Our lab and field measurements-based parameterizations". It is not clear how the laboratory results were incorporated or used into the parametrizations. As stated in Major point #5 Lab and Field data should not be combined.
AR: Laboratory and field data were fitted independently (not combined). Please see our revised **Table S3**. We now only offer two lab parameterizations (for two laboratory samples, TXD01 and TXD05) and one field parametrization. In addition, the figure below shows our parametrization fits for those lab and field data.

[Figure]

Figure. OLLF-INP parameterizations and fit curves based on Table S3 compared to our measurements for (a) TXD01, (b) TXD02, and (c) Field_Median.
RC: Lines 643-544: "Additionally, the observed consistency in the spectral slopes (i.e., Table 5) suggests that lab and field measurements exhibit similar IN ability at examined Ts". This was true at -25C not for the whole temperature range.

AR: The referee is right. There is a slight difference between the Δlog ($n_{s,geo}$)/Δ$T$ values of laboratory results (~0.4) and that of the field (~0.5). The said sentence has been excluded from the manuscript. In general, our numbers are higher than what has been found in previous soil dust ice nucleation studies. This is now mentioned in the revised **Sect. 3.3.2**.

"Overall, the range of spectral slope deviations (0.41– 0.52) is higher than what we previously studied in soil dust samples in **Fig. 4** (0.15 – 0.27; S16 – O14), indicating a unique feature of the OLLF dust"

RC: Figure 5: In x-axis is it micro or milligrams? In Table 1 it is reported in micrograms
AR: Thank you for catching this. A "microgram" is right. The figure x-axis caption is corrected.

RC: Figure 7: it does not make sense to have a, b,c,d on each panel. The description of panels d are unclear. What is the concentration reported in red and blue?
AR: The panels (d) show two variables: (left axis) the number concentration of > 20 μm volume equivalent diameter particles measured by a welas optical particle counter, which is virtually equivalent to the number concentration of ice crystals measured during the AIDA expansion experiments; (right axis) the number concentration of aerosol particles in the AIDA chamber, measured by a condensation particle counter. It was clarified in L403-404 (now in the revised **Sect. 3.1.2**). We updated the figure axis texts accordingly. The authors would like to retain these panel IDs. We believe that retaining (a)-(d) offers a simple interpretation of panels.

RC: Figure 9: Why is it that noise? What is the time resolution? Would it not be better to do at a lower time resolution to avoid the noise?
AR: Time-resolution is 5-min. As part of our tapered-element oscillating microbalance (TEOM) data screening and evaluation protocol, all systematic errors (i.e., mass concentration outside of measurable limits, noise > 100%, 3.5 < main flow < 2.5, and 14 < sheath flow < 13) were excluded for our data analysis. The screened TEOM data were used as ambient particle emission data to estimate INP concentration from a feedlot. As stated in our manuscript, the resuspension/emissions mechanism of feedlot soil dust is not controllable as it highly depends on a unit of mobile livestock, which can be impulsive. Thus, these spikes are realistic (not any systematic errors). While time-averaging the data may eliminate some spikes in this figure, we would like to report processed individual data points in this figure. Please know that we offer seasonal time-averaged data of estimated $n_{INP}$ in **Table S2**.

Technical comments:

RC: Lines 28: what is the meaning of "3 × 10-7 g L-1" .
AR: It is the minimum TEOM-measured aerosol particle mass concentration. The sentence, which included this number, is now excluded as the relevant discussion is moved to SI.

RC: Lines 48-51: Delete them.
AR: Deleted.

RC: Line 54: delete "chapter 9".
AR: Deleted.

RC: Line 56: Add other references in addition to Storelvmo (2017)
AR: Bourcher et al. (2013) and Zelinka et al. (2020) added.

RC: Line 78: "Agricultural land use is in excess of 50% of total U.S. land use". Please rewrite it.

AR: Corrected to:
"Agricultural land use accounts for more than 50% of total U.S. land use according to the U.S. Department of Agriculture (Bigelow and Borchers, 2012),…"

RC: Line 87: Add references after "conditions".
AR: We added Auverman (2001) and Postoor et al. (2012).

RC: Line 90: Add references after "head".

AR: We added Annamalai et al. (2012) and USDA (2021).

RC: Line 101: "we examinedthe". Fix it.
AR: Fixed.

RC: Line 106: Add references after "materials".
AR: This sentence has been removed. So no reference is provided in the revised manuscript, but a relevant reference could have included: National Research Council (NRC): Air Emissions from Animal Feeding Operations: Current Knowledge, Future Needs, Ad Hoc Committee on Air Emissions from Animal Feeding Operations, Committee on Animal Nutrition, NRC, 2003.

RC: Line 125: Add references after "definition".
AR: We now provide the U.S. EPA's URL - https://www3.epa.gov/npdes/pubs/sector_table.pdf

RC: Line 161: "using an offline freezing technique". Which one?
AR: INSEKT

RC: Line 171: "those of previously measured". Fix it.
AR: The measured BET specific surface area ($SSA$) values of OLLF samples are slightly higher compared to those of previously measured agricultural soil dust samples (0.74 – 2.31 $m^2 g^{-1}$; O'Sullivan et al., 2014),
→
The measured BET $SSA$ values of OLLF samples are slightly higher compared to previously measured agricultural soil dust samples (0.74 – 2.31 $m^2 g^{-1}$; O'Sullivan et al., 2014),

RC: Line 184: Remove "Fig 5."
AR: Removed.

RC: Lines 196-198: "proxies. We chose the AIDA chamber as our study platform because it simulates ice formation in mixed-phase clouds in a controlled setting with respect to both T (± 0.3 °C) and humidity (± 5%; Fahey et al. 2014)." Delete.
AR: Deleted.

RC: Lines 201-203: "experiment. The AIDA has been applied for the analysis of both ambient and lab-generated INPs and has facilitated characterization of many INP species with the IN efficiency uncertainty of ± 39% (Steinke et al., 2020; Ullrich et al., 2017; Niemand et al., 2012; Hoose and Möhler, 2012)." Delete.
AR: Deleted.

RC: Lines 223-225: "Another motivation for using the AIDA facility is its ice-selecting pumped counterflow

virtual impactor (IS-PCVI; Hiranuma et al., 2016). As detailed in Supplemental Information (SI) Sect. S1, IS-PCVI separates ICRs from interstitial particles, including cloud droplets, at Ts below -20 °C." Delete.
AR:  Deleted.

RC: Line 225: "evaporation". Should it be sublimation?
AR: Could be both. The authors incorporated sublimation in the main text.

RC: Line 300: "Texas dust". Delete.
AR: Done.

RC: Line 310: "Next, our metagenomics analysis method of total DNA is described". Delete.
AR: Deleted.

RC: Line 386: "at below -20 °C". Fix it.
AR: below -20 °C.

RC: Lines 386-387: "ambient aerosol particle mass concentrations based". PM10?
AR: Yes.

RC: Line 389: "in a controlled lab setting". Delete.
AR: Deleted.

RC: Line 412: "O14, S16, S20". Add the origin/source of the samples.
AR: The authors added the followings: O14 (England), S16 (Mongolia, Argentina, and Germany), S20 (Northwestern Germany, Wyoming), T14 (Wyoming), T14 (China), and U17 (desert dust samples from Aisa, Canary Island, Israel, and Sahara).

RC: Line 472: Add references after "mass".
AR: The authors added Hoose et al. (2010) in **Sect. 3.1.4**.

RC: Lines 486-487: "properties. All of our single particle analyses were carried out with the following parameters: electron beam accelerating voltages of 15 keV, spot size of 50, and working distance of 10 mm". This belongs to Methods.
AR: Ok, but this part is now omitted.

RC: Line 525: "We elected to use the". Fix it.
AR: Fixed.

RC: Line 529: "typically substantially lower". Fix it.
AR: Fixed.

RC: Line 580: "atmospherically relevant". What do the authors mean?
AR: The authors realize that this is too ambitious to retain, so deleted it. The authors understand that it requires more investigation to state it this way.

RC: Line 593: "dust samplew". Fix it.
AR: Fixed.

RC: Table 6. Last column "Spermicron Size". Fix it.
AR: Fixed.

**References**

[revised manuscript text omitted]

**Response to Referee #4**

The authors would like to express our sincere gratitude for the referee and helpful comments. Below, we provide our point-by-point responses. The referee's comments (RC) are shown from here on in black. The authors' responses (AR) are in blue below each of the referee's statements. We introduce the revised materials in green color along/below each one of your responses.
* * *
RC: The reviewed manuscript presents IN measurements and particle characteristics from particles emitted from select feedlot sites in the Texas Panhandle. The field measurements are complemented by laboratory measurements and the study benefits from the various analyzed particle characteristics. However, I would strongly suggest the authors to further revise the manuscript for conciseness and thus increasing readability and clarity of most sections, but particularly the methods and results sections.

AR: The authors appreciate these general remarks. Following the referee's advice, we revised our manuscript structure and contents to improve the readability and conciseness of this paper. Below, we provide our point-by-point responses.

RC: Some additional comments for the authors:
- Line 214/215: What is the scientific motivation of point (2)?

AR: Our motivation is to complement the aerosol interaction and dynamics in the atmosphere (AIDA) immersion freezing data at relatively high temperatures. The authors now clarified our point is **Sect. 2.1.3** as:

"The INSEKT data are especially useful to complement the AIDA chamber immersion results at temperatures above -25 °C".

RC: - Line 218/219: INSEKT covers a different size range than AIDA? If this is the main point for using INSEKT then the authors should mention this here to make it more clear why certain methods are used alongside others.

AR: No. We consider that both AIDA and ice nucleation spectrometer of the Karlsruhe Institute of Technology (INSEKT) cover the same aerosol particle size range as the aerosol particles for INSEKT analysis were directly sampled from the AIDA chamber. The sampler for INSEKT employed a sampling flow rate of 10 L min$^{-1}$ to minimize in-line particle losses. For the ice nucleation efficiency estimation, both AIDA and ISEKT scaled ice-nucleating particle (INP) concentration to the same aerosol particle measurements (summarized in **Table 1**). We have added the following parts in our INSEKT methodology sub-section (**Sect. 2.1.3**) to clarify this point:

"…and $S_{total}/M_{total}$ is a geometric specific surface area. The $S_{total}/M_{total}$ value used for this study was derived from particle size distribution measurements from the AIDA chamber (presented in **Table 1**)".

RC: I would recommend to generally revise the methods section for conciseness and thus clarity with a focus on providing the reader with a clear overview of how methods complement each other.

AR: The authors agree with the referee. We realized that the structure of the method section and presenting various things could be overwhelming. The authors also acknowledge that retaining the format from a previous version would cause additional confusion. Therefore, the authors decided to restructure the main article to present the most invaluable scientific outcomes and limit the amount of Supplemental Information (SI) for the sake of readability. Our decision is based on considering all four referees' comments.

    The authors revised the manuscript to feature the **abundance of supermicron aerosol particles acting as feedlot INPs from lab and field studies** and did the following modifications:

(1) Separating **Sects. 2 and 3** based on laboratory study (sub-section 1) and field investigation (sub-section 2) to explain methods, materials, and results for each sub-section independently.

(2) Moving sample descriptions to the Results and Discussion section (**Sect. 3.1.1**) and leaving only concise technique explanation in the Materials and Methods section (**Sect. 2.1.1**) to increase the readability of the manuscript in an organized manner.

(3) Moving the heat treatment data, outcomes, and discussion from the main manuscript into a single **SI Sect. S4**. Keeping it over different sections in methods and results impaired the overall readability in our previous version, and the authors believe that this modification resolves the readability issue.

(4) Removing all bulk sample discussions from this manuscript and focusing on the filter-collected aerosolized samples – So there will be no bulk vs. aerosolized sample discussion. In the end, the bulk is not our main outcome. The revised paper focuses on lab vs. field, we believe that the aerosolized sample is more relevant to what is in the field than the bulk sample.

(5) Removing the ice residual composition discussion from the main manuscript. We do not have statistically valid aerosol particle composition data of our 'field' samples from this study. As referee 3 is concerned, we cannot conduct the comparison of laboratory and filed sample compositions, and the former Sect. 3.4 contributes little to the main text. The authors agree that the composition analysis is not the main focus of this manuscript, and decided to exclude this part from the manuscript.

(6) Moving the discussion regarding the estimated INP concentrations to **SI Sect. S5**.

In addition, the authors also changed the title of our manuscript to "Laboratory and field studies of ice-nucleating particles from open-lot livestock facilities in Texas", which better represents our research.

RC: - Method on DNA analysis: does sterilization remove DNA which might interfere with the DNA of interest here?
AR: The authors think it is not an issue. We made sure to clean the sampler itself and all fittings with volatile reagent alcohol well ahead of each sampling activity. Although sterilization may not completely remove DNA from filters and filter holders, assembly was done with great caution as to not contaminate filter holders and filters with non-sample DNA. Also, sterilization causes DNA fragmentation to small fragments which will not be amplifiable during metagenomics analysis.

RC: - Please ensure that acronyms are defined the first time they appear in the manuscript, this will help the reader to better follow along without having to search for definitions. E.g., definition of ICR in line 226 missing.
AR: Ice crystal residual (ICR) was defined in L113. We checked acronyms. We also provide a list of abbreviations in **SI Sect. S6**.

RC: - Figure 1: OLLF schematic is helpful but not very clear. Would suggest to adapt the shape and relative size of the study areas. Additionally, it is not clear from the map where the boundary between OLLF 2 and 3 is.
AR: It was intentionally made that way to protect the identity of commercial cattle feeders. The authors modified the figure to point out roughly where they are within county boundary lines but would not be able to provide any information beyond (e.g., exact coordinates). Please see the revised figure on the next page.

[Figure]

Figure 2. Schematic of the field sampling activity at individual sites (only the counties are shown). The dimension of each facility (east – west × north – south) is (1) 1.6 × 1.6 km, (2) 1.0 × 0.8 km, (3) 0.7 × 0.7 km, and (4) 0.8 × 1.4 km. A combination of polycarbonate filter samplers (PFSs) and DustTrak instruments was used at the nominally upwind and downwind edges of OLLF-1 to OLLF-3.

RC: - Comment 6. Ln 48 from reviewer 2, would suggest to also clarify in the manuscript
AR: To increase the readability and conciseness of the heat treatment part, all associated contents regarding heat treatment are now compiled in **SI Sect. S4**, and the clarification is provided in this section.

RC: - SI, particularly Figure S3: increase font size and/or image resolution for legibility.
AR: Done.

RC: - SI, Table S3: check for significant figures of the values provided in the table and be consistent.
AR: We now reduced significant figures as much as possible (please see the revised **Table S3**). We need to keep the reported significant figures to reproduce polynomial fits (We provide our fitting plots on the next page). It is susceptive to those decimals. As stated in the Table caption – "To reproduce the fitted curves, we needed to include all decimals". In addition, the authors have checked the consistency of significant figures and decimal digits for other tables, and we corrected them accordingly. For instance, in **Table 2**, we now report two decimal points. Air volume values and flow rates are scaled to the cross-section area of the filter examined.

[Figure]

Figure. Open-lot livestock facility (OLLF)-INP parameterizations and fit curves based on Table S3 compared to our measurements for (a) TXD01, (b) TXD02, and (c) Field_Median.

---

## Author Response (AR3)

Some additional comments for the authors:

The authors would like to thank the referee again for thoughtful comments, that are helpful to improve the quality of our manuscript. Below, we provide our point-by-point responses. The referee's comments (RC) are shown from here on in black. The authors' responses (AR) are in blue below each of the referee's statements.

RC: - L137: replace 'RGB1000' with 'brush dispenser'as it tells the reader more than an instrument ID . AR: Replaced.

RC: - Given this is a submission to ACP, thus I would suggest the authors to consider to word section 2.5.1 (Extraction of total DNA and metagenomics analysis of sample microbiomes) in a more accessible way to the atmospheric science community.

AR: We agree. We reword the subsection title to "2.1.5. Analysis of sample microbiomes". We also begin the section with the following phrase, which explains the method to the atmospheric science community:

"The microbiome of our samples was characterized by metagenomics analysis. With this approach, total DNA is extracted from environmental samples; this DNA is a mix from all microorganisms and macroorganisms present in a sample. The qualitative and quantitative identification of microorganisms is carried out by amplifying (by polymerase chain reaction) and sequencing (several methods are in use) specific DNA segments of phylogenetic markers (genes that are used for identifying an organism) from the extracted and purified total DNA. Bioinformatics analysis of sequences obtained determines the nature and abundance of microorganisms in this sample."

RC: - L 269: for the section 2.2.3 title consider adding the article ,The' at the beginning as otherwise it is a bit hard to read as title/instrument name rather than a sentence on the first read of the section heading. And/or add the instrument acronym in brackets to the section heading; this should clarify for the reader.

AR: We added both "The" and the instrument acronym in brackets for Sects. 2.2.3, 2.1.3, and 2.1.4.

RC: - L 278 to 280, something is a bit odd in the sentence structure – the order of information (particular of ,in this study') is confusing.

AR: Our apologies for this confusion. We realized that our detection limit of 0.05 INP per L air is stated and given in the previous sentence. As this sentence (L278-280) contains repetitive information, we simply decided to omit it as the paragraph makes sense without it.

RC: - L282: here you talk about a grayscale to detect freezing onset for the drop freezing setup WT-CRAFT. How is this for the INSEKT instrument you describe earlier, can you specify if grey scale is applicable for INSEKT to?

AR: Yes, it was applied for INSEKT. For this study, we set a threshold value for the greyscale on our LabView-based image analysis tool. Once the optimized threshold value was achieved, we considered the well was frozen. We now clarify this point in L162-163 as:

"If a well froze upon the presence of an INP, a camera detected the associated brightness change based on an optimized greyscale threshold value set on the LabView software for this study." RC: - L329: RH also decreases in case c.ii. Thus I would recommend rewording the sentence to "... in almost all cases RH drops/decreases...".

AR: Ok. We rephrased the sentence as "In almost all cases, the *RH* dropped during some expansions at low temperatures."

RC: -L405 (Table 6): How do the authors account for the different sampling duration for the listed samples? What are/might be the implications for the n(INP)?

AR: Different sampling durations result in different air volumes sampled for individual samples. As can be calculated from Table 2, the average air volume sampled for individual samples ( $\pm$  standard deviation) was 955.0  $\pm$  300.5 L (min – max = 317.8 – 1560.0 L). While the air volumes deviated (mainly limited by on-site feedlot activities), we conducted our sampling activities when cattle were active in the evening hours under relatively stable ambient conditions as seen in Table 7 (except the winter sampling activities). To respond to the reviewer's point, we checked a correlation between the air volume ( $V_{air}$ , Table 2) and  $n_{INP}$  @ -25 °C (Table 6). Our results show  $r^2$  of 0.069. Thus, our result implies that the different air volumes did not significantly impact to  $n_{INP}$  (otherwise, they might linearly correlated).

RC: - Figure 6: The order of the seasons (in panels a,b,c) is not intuitive. I would suggest the authors to change the order e.g. to (a) Winter, (b) Spring, (c) Summer and for easier reading of the graph to include the season names in the respective figure panel.

AR: Reordered as suggested. The figure captions as well as texts in Sect. 3.2.2 are also revised for consistency.

**Figure 6.** Downwind OLLF  $n_{\text{INP}}$  spectra from 2017 – 2019 sorted based on meteorological seasons are shown; winter (a), spring (b), and summer (c). The uncertainties in temperature and  $n_{\text{s,geo}}$  are  $\pm 0.5 \,^{\circ}\text{C}$  and  $\pm \text{CI95\%}$ , respectively, and error bars are shown at -5, -10, and -15 °C. The shaded area represents minimum – maximum  $n_{\text{INP}}$ .

**Figure 8.** The  $n_{s,geo}$  spectra of OLLF aerosol particles from field ambient samples collected in 2017 – 2019. All downwind  $n_{s,geo}$  spectra from winter (a), spring (b), and summer (c) are shown. Different symbol shapes correspond to individual OLLF sites as indicated in the legend. The uncertainties in temperature and  $n_{s,geo}$  are  $\pm 0.5$  °C and  $\pm 23.5$ %, respectively, and representing error bars are shown at - 5, -10, and -15 °C. The shaded area represents minimum – maximum  $n_{s,geo}$ .

RC: - SI, Figure S4: There is a typo in the caption.

AR: Corrected. filed  $\rightarrow$  field